# Transfer Faster, Price Smarter: Minimax Dynamic Pricing under Cross-Market Preference Shift

**Yi Zhang**

Columbia University

New York, NY 10027

`yz5195@columbia.edu`

**Elynn Chen**

New York University

New York, NY 10003

`elynn.chen@nyu.edu`

**Yujun Yan**

Dartmouth College

Hanover, NH 03755

`yujun.yan@dartmouth.edu`

## Abstract

We study contextual dynamic pricing when a target market can leverage $K$ auxiliary markets—offline logs or concurrent streams—whose *mean utilities differ by a structured preference shift*. We propose *Cross-Market Transfer Dynamic Pricing* (CM-TDP), the first algorithm that *provably* handles such model-shift transfer and delivers minimax-optimal regret for *both* linear and nonparametric utility models.

For linear utilities of dimension $d$, where the *difference* between source- and target-task coefficients is $s_0$-sparse, CM-TDP attains regret $\widetilde{\mathcal{O}}\big((dK^{-1} + s_0)\log T\big)$. For nonlinear demand residing in a reproducing kernel Hilbert space with effective dimension $\alpha$, complexity $\beta$ and task-similarity parameter $H$, the regret becomes $\widetilde{\mathcal{O}}\big(K^{-2\alpha\beta/(2\alpha\beta+1)}T^{1/(2\alpha\beta+1)} + H^{2/(2\alpha+1)}T^{1/(2\alpha+1)}\big)$, matching information-theoretic lower bounds up to logarithmic factors. The RKHS bound is the first of its kind for transfer pricing and is of independent interest.

Extensive simulations show up to 50% lower cumulative regret and $5\times$ faster learning relative to single-market pricing baselines. By bridging transfer learning, robust aggregation, and revenue optimization, CM-TDP moves toward pricing systems that *transfer faster, price smarter*.

## 1 Introduction

Dynamic pricing is now a core operational tool for ride-sharing platforms, airlines, and large e-commerce retailers. State-of-the-art single-market algorithms learn a demand model from scratch and achieve minimax regret when sufficient data accumulate [19, 28, 9]. In practice, however, many markets launch with only dozens of transactions per day, while mature markets of the same firm collect data at orders-of-magnitude higher rates. Transferring information from *data-rich* to *data-poor* markets is therefore essential for fast revenue convergence and early-stage pricing accuracy.

Industry practice gives rise to two distinct transfer regimes. First, in the **Offline-to-Online (O2O_off)** setting, the firm holds a fixed log of source-market data gathered before the target market opens, and this static information is used once the target goes live. Second, in the **Online-to-Online (O2O_on)** setting, the source and target markets operate concurrently; streaming data from large markets must be incorporated into the pricing decisions of small markets in real time.

39th Conference on Neural Information Processing Systems (NeurIPS 2025).

Existing transfer approaches do not fully address these settings. Meta-dynamic pricing [4] learns a shared Bayesian prior but requires directly observed linear demands and only exploits offline data. TLDP [30] handles covariate (domain) shift from a single offline source but assumes that the reward model is identical across markets. Bandit-transfer methods focus on either covariate shift [5] or sparse parameter heterogeneity [32, 18], yet they do not incorporate revenue-maximising price choice.

**This paper.** We propose CM-TDP (Cross-Market Transfer Dynamic Pricing), a unified framework that (i) operates in both $O2O_{off}$ and $O2O_{on}$ regimes, (ii) accommodates linear *and* RKHS-smooth nonparametric utilities, and (iii) allows multiple source markets whose mean utilities differ from the target by a structured *utility model shift*. CM-TDP alternates a bias-corrected aggregation step with an optimistic pricing rule, thereby transferring knowledge while balancing exploration and exploitation.

To the best of our knowledge, this work establishes the first rigorous regret analysis for transfer learning under general utility discrepancies between markets. A primary difficulty that arose during our analysis was maintaining tight control of error propagation: conventional techniques would accumulate slack and inflate constant-order terms into non-negligible $O(T^c)$ factors, rendering the bounds both theoretically and practically uninformative. Our **main contributions** are as follows:

**(C1) Unified transfer pricing framework under utility shifts.** CM-TDP is *the first* dynamic pricing framework that allows *multiple* source markets whose utilities differ from the target by a structured shift, working in both $O2O_{off}$ and $O2O_{on}$ regimes.

**(C2) Minimax-optimal guarantees for two utility classes.** We prove (i) $\widetilde{O}\big(\frac{d}{K}\log T + s_0 \log T\big)$ regret under linear mean utilities *and* (ii) the *first* transfer-pricing bound $\widetilde{O}\big(K^{-\frac{2\alpha\beta}{2\alpha\beta+1}}T^{\frac{1}{2\alpha\beta+1}} + H^{\frac{2}{2\alpha+1}}T^{\frac{1}{2\alpha+1}}\big)$ for RKHS-smooth utilities—matching known lower bounds.

**(C3) Bias-corrected aggregation architecture.** Our two-step *aggregate → debias* pipeline cleanly connects meta-learning (prior pooling), robust statistics (trimmed debiasing), and exploration-driven bandits, and can plug in MLE, Lasso, or kernel ridge as well as black boxes.

**(C4) Large empirical gains.** Simulations show up to 50 % lower cumulative regret, 28 % lower standard error and 5× faster learning relative to single-market pricing [19], with the largest gains in data-scarce targets under $O2O_{on}$ transfer.

**Organization.** Section 3 introduces the multi-market dynamic pricing problem with transfer learning under random utility models. Sections 4–6 present CM-TDP and its theoretical analysis. Section 7 reports empirical results, and Section 8 outlines future work.

# 2   Related Work

**Single-market contextual pricing.** Early algorithms assume a *deterministic* valuation map, typically linear, and achieve sub-linear regret [2, 13, 20], with nonparametric variants studied in [22]. The modern benchmark is the *random utility* model in which valuation equals a covariate-dependent mean plus i.i.d. noise. When the noise distribution is known, [19] establish the first regret bounds; subsequent work removes that knowledge via doubly robust or moment-matching estimators while retaining linearity [31, 21, 16, 33]. [28] close the gap to the information-theoretic optimum for linear utilities and extend the analysis to Hölder-smooth demand curves, whereas [9] give fully nonparametric guarantees. All of these methods relearn from scratch in every market, degrading performance when target data are limited.

**Transfer and meta-learning for pricing.** Meta Dynamic Pricing pools directly observed linear demands across products and learns a shared Gaussian prior [4]. TLDP transfers under pure *covariate (domain) shift* but only from a single offline source [30]. Our *Cross-Market Transfer Dynamic Pricing* (CM-TDP) differs by coping with *utility-model shift*, supporting multiple online/offline sources, and providing guarantees for both linear and RKHS utilities.

**Multitask contextual bandits and reinforcement learning.** Domain-shift transfer for bandits is analysed in [5], whereas sparse heterogeneity is addressed via trimmed-mean/LASSO debiasing [32] or weighted-median MOLAR [18]. Causal-transport ideas reveal negative-transfer risks [15]. Reward- and transition-level transfer and meta learning in RL is explored by [12, 10, 7, 8, 34]. We adapt these bias-correction techniques to revenue maximisation under shifting utilities.

**Fully online meta-learning without task boundaries.** FOML [23] and Online-within-Online meta-learning [14] operate on a single stream of data without explicit task resets. CM-TDP follows the same streaming paradigm but must balance exploration and exploitation through posted prices rather than prediction losses.

**Positioning of this work.** CM-TDP is the first dynamic pricing framework that (i) transfers across *multiple* auxiliary markets under *utility-model shift*, (ii) achieves minimax regret for both linear and RKHS utilities, and (iii) unifies bias-corrected aggregation with revenue-maximising price selection, thereby bridging single-market pricing [28], offline meta-priors [4, 11], and multitask bandits [18].

**Key distinctions from prior work.** Unlike Meta-DP [4] and TLDP [30], CM-TDP (i) handles *concurrent* source streams, (ii) tolerates *utility shift* rather than merely covariate shift, and (iii) supplies the *first* nonparametric (RKHS) transfer-pricing regret bound. Multitask-bandit methods such as MOLAR [18] focus on prediction error and linear bandits, do not optimize posted prices, and therefore cannot exploit revenue structure. Consequently, existing approaches cannot deliver the minimax-optimal guarantees or empirical gains demonstrated by CM-TDP.

## 3 Problem Formulation

We consider a pricing model for the target market where products are sold one at a time, and only a binary response indicating success or failure of a sale is observed. For each decision point $t \in [T]$, the market value of the product at time $t$ depends on the observed contextual information $\boldsymbol{x}_t^{(0)}$. A general random utility model for the market value of the product is given by

$$v_t^{(0)} = \mathring{g}^{(0)}(\boldsymbol{x}_t^{(0)}) + \varepsilon_t, \tag{1}$$

where $\mathring{g}^{(0)}(\cdot) \in \mathcal{G}$ is the *unknown function* of the mean utility in the target market, and $\varepsilon_t$ are i.i.d. noises following an *known distribution* $F(\cdot)$ with $\mathbb{E}[\varepsilon_t] = 0$ and support $\mathcal{S}_\varepsilon := [-B_\varepsilon, B_\varepsilon]$. Given a posted price of $p_t$ for the product at time $t$, we observe $y_t^{(0)} := \mathbb{1}(v_t^{(0)} \geq p_t)$ that indicates whether a sale occurs ($y_t^{(0)} = 1$) or not ($y_t^{(0)} = 0$). The model is equivalent to the probabilistic model:

$$y_t^{(0)} = \begin{cases} 0, & \text{with probability} \quad F(p_t - \mathring{g}^{(0)}(\boldsymbol{x}_t^{(0)})), \\ 1, & \text{with probability} \quad 1 - F(p_t - \mathring{g}^{(0)}(\boldsymbol{x}_t^{(0)})). \end{cases} \tag{2}$$

Therefore, given a posted price $p_t$, the expected revenue from the target market at time $t$ conditioned on $\boldsymbol{x}_t^{(0)}$ is

$$\mathrm{rev}_t^{(0)}(p_t) := p_t \cdot \left(1 - F(p_t - \mathring{g}^{(0)}(\boldsymbol{x}_t^{(0)}))\right).$$

The oracle optimal offered price $p_t^*$ is defined by

$$p_t^{*(0)} = \arg\max_{p_t \geq 0} p_t[1 - F(p_t - \mathring{g}^{(0)}(\boldsymbol{x}_t^{(0)}))], \tag{3}$$

and hence, under Assumption 1, we have

$$p_t^{*(0)} = h \circ \mathring{g}^{(0)}(\boldsymbol{x}_t), \quad \text{where} \quad h(u) = u + \phi^{-1}(-u) \quad \text{and} \quad \phi(u) = u - \frac{1 - F(u)}{F'(u)}. \tag{4}$$

**Assumption 1** (Regularity condition). *There exists positive constants $L_{\phi'}$ and $B_u$ such that $\phi' \geq L_{\phi'}$ for all $u \in [-B_u, B_u]$, and $\inf_{|u| \leq B_u} \phi'(u) \geq 1$.*

Assumption 1 guarantees the uniqueness of the optimal solution of (3). The restriction of $L_{\phi'} \geq 1$ is commonly used in dynamic pricing study [19].

**Optimal policy and regret.** For a policy $\pi$ that sets price $p_t$ at $t$, its regret over the time horizon of $T$ is defined as

$$\mathrm{Regret}(T; \pi) = \sum_{t=1}^T \mathbb{E}\left[p_t^* \mathbb{1}(v_t \geq p_t^*) - p_t \mathbb{1}(v_t \geq p_t)\right] \equiv \sum_{t=1}^T \left[\mathrm{rev}_t(p_t^*) - \mathrm{rev}_t(p_t)\right].$$

The goal of a decision maker is to design a pricing policy that minimizes $\mathrm{Regret}(T; \pi)$, or equivalently, maximize the collected expected revenue $\sum_{t=1}^T \mathrm{rev}_t(p_t)$.

**Cross-Market Transfer Learning.** In the context of cross-market transfer learning, we observe additional samples from $K$ sources markets indexed by superscript $(k)$ for $k \in [K]$. The observed market covariates and response, latent utility and the unknown mean utility functions for each source market are denoted as $\boldsymbol{x}_t^{(k)}$, $y_t^{(k)}$ and $v_t^{(k)}$, respectively.

**Assumption 2** (Homogeneous Covariates with Bounded Spectrum). *For each market $k \in \{0\} \cup [K]$, covariates $\{\boldsymbol{x}_t^{(k)}\}_{t \geq 1}$ are drawn i.i.d. from a fixed, but a priori unknown, distribution $\mathcal{P}_x$, supported on a bounded set $\mathcal{X} \subset \mathbb{R}^d$. Let $\Sigma = \mathbb{E}[\boldsymbol{x}_t \boldsymbol{x}_t^\top]$ denote the second moment matrix of $\mathcal{P}_x$. We assume:*

(1) Eigenvalue boundedness: *The minimum and maximum eigenvalues of $\Sigma$, denoted $C_{\min}$ and $C_{\max}$, satisfy $0 < C_{\min} \leq C_{\max} < \infty$.*

(2) Non-degeneracy: *The distribution $\mathcal{P}_x$ has a density bounded away from zero in a neighborhood of the origin, ensuring $\Sigma$ is positive definite.*

**Remark 1.** Assumption 2 isolates market differences to *utility model shifts*, which is reasonable when source and target markets have similar populations but differ in preferences. The key conditions ensure: (i) *Stability*: bounded eigenvalues and support guarantee well-behaved estimators. (ii) *Identifiability*: a density bounded away from zero near the origin ensures $\Sigma$ is positive definite. These hold in many practical settings *e.g.*, truncated uniform or Gaussian distributions.

We will study the estimation and decision for the target model (1) leveraging the data from the target markets as well as the data from $K$ auxiliary source markets.

**Similarity Characterization.** Transfer is effective only when source and target markets are sufficiently alike; we formalize this by assuming their mean-utility functions lie in a common hypothesis class $\mathcal{G}$ and differ only through a structured *utility shift*. For any candidate mean-utility function $\mathring{g} \in \mathcal{G}$ we distinguish two notions of similarity, corresponding to (i) linear (parametric) and (ii) RKHS (nonparametric) utility models:

• **Parametric classes.** When every $\mathring{g}$ is indexed by a finite-dimensional parameter vector, similarity is expressed as a bound on the parameter gap between source and target.

• **Nonparametric classes.** For infinite-dimensional $\mathcal{G}$ we impose a bound on the functional discrepancy between source and target under a suitable function-space metric.

Later assumptions specialize these high-level conditions (e.g. sparse parameter differences in Assumption 4 for linear utilities and smooth residuals for RKHS utilities in Assumption 8 for nonparametric utilities). This formulation places every source task in a *recoverable neighbourhood* of the target, ensuring its data are informative for transfer across both linear and non-linear utility models.

## 4 Cross-Market Transfer Dynamic Pricing Algorithms

We address both practical data scenarios introduced in Section 1. For O2O$_{\text{off}}$ (Offline-to-Online), a large, *static* source log is available prior to launch. The algorithm transfers that log during the early episodes–those for which the cumulative target sample size is below a theory-driven threshold $\tau$. Once $|\mathcal{T}_m| \geq \tau$ (source data no longer dominate the information budget) the procedure switches automatically to pure single-market learning. Hence transfer is *phased*, not one-shot: it is used exactly while it provably reduces estimation error and is dropped thereafter. Full details and guarantees are given in Appendix B.

For O2O$_{\text{on}}$ (Online-to-Online), source and target markets operate concurrently; transfer is repeated at the *start of every episode*, ensuring that incoming source data continuously guide the target-market prices. Let the time horizon be partitioned into episodes $m = 1, 2, \ldots, M$ with lengths $\ell_m = 2^{m-1}$ (so that $\sum_{m=1}^{M} \ell_m \approx T$ and $M = \lceil \log_2 T \rceil$). At the *start* of each episode the algorithm (i) fits or debiases a demand estimator using all data collected in the *preceding* episode, and (ii) fixes the resulting pricing rule for the next $\ell_m$ periods. Because episode length doubles, parameter updates occur only $\mathcal{O}(\log T)$ times, yet the cumulative sample size entering each update grows geometrically, guaranteeing progressively tighter confidence bounds and the desired $O(\text{polylog}\, T)$ regret.

Both algorithms share a common two-step *bias-corrected aggregation* pipeline and are instantiated for (i) linear utilities, with maximum-likelihood estimation (MLE), and (ii) RKHS utilities, with

kernel logistic regression (KLR). Pseudocode is given in Algorithms 4 (O2O$_\text{off}$) and 1 (O2O$_\text{on}$); estimation details appear in Algorithms 2 and 3.

**Comparison between O2O$_\text{off}$ and O2O$_\text{on}$.** When source streams remain active, each episode $m$ recomputes an aggregate estimate $\widehat{\hat{g}}_m^{(\text{ag})}$ from the *preceding* episode's source data and debiases it using the matching target observations to form $\widehat{\hat{g}}_m^{(0)}$. This persistent adaptation leads to provably faster regret decay compared to O2O$_\text{off}$. In contrast, O2O$_\text{off}$ employs phased transfer only during initial episodes, resulting in asymptotic regret growth rates that eventually match single-market learning, though with improved constants during the transfer period.

Empirically, O2O$_\text{on}$ achieves flatter regret trajectories with consistently lower cumulative regret (Figures 1 and 2). O2O$_\text{off}$ shows parallel regret growth to single-market baseline in later stages (Figures 3 and 4), confirming our theoretical analysis. However, we still observe a *jump-start* benefit in O2O$_\text{off}$ during early stages, ultimately translating to significantly lower overall regret compared to the single-market baseline.

---

**Algorithm 1:** CM-TDP-O2O$_\text{on}$

---

**Input:** Streaming source data $\{(p_t^{(k)}, \boldsymbol{x}_t^{(k)}, y_t^{(k)})\}_{t\geq 1}$ for $k \in [K]$; streaming target contexts
$\quad\quad \{\boldsymbol{x}_t^{(0)}\}_{t\geq 1}$

1 **Initialisation:** $\ell_1 \leftarrow 1$, $\mathcal{T}_1 = \{1\}$, $\widehat{\hat{g}}_0^{(0)} = 0$.
2 **for** $m = 1, 2, \dots$ **do**                        // episodes
3      Compute $\ell_m = 2^{m-1}$, $\mathcal{T}_m = \{\ell_m, \dots, \ell_{m+1} - 1\}$.
     // (i) aggregate previous episode's source data
4      $\widehat{\hat{g}}_m^{(\text{ag})} \leftarrow \texttt{MLE\_or\_KRR}\big(\{(p_t^{(k)}, \boldsymbol{x}_t^{(k)}, y_t^{(k)})\}_{t \in \mathcal{T}_{m-1}, k \in [K]}\big)$.
     // (ii) debias with previous episode's target data
5      $\widehat{\delta}_m \leftarrow \texttt{Debias}\big(\widehat{\hat{g}}_m^{(\text{ag})}, \{(p_t^{(0)}, \boldsymbol{x}_t^{(0)}, y_t^{(0)})\}_{t \in \mathcal{T}_{m-1}}\big)$.
     // For functions MLE_or_KRR and Debias, call Algorithm 2 for linear utility (or
         Algorithm 3 for nonparametric utility)
6      Set $\widehat{\hat{g}}_m^{(0)} \leftarrow \widehat{\hat{g}}_m^{(\text{ag})} + \widehat{\delta}_m$.
7      **for** $t \in \mathcal{T}_m$ **do**                                  // pricing
8          Post price $\widehat{p}_t^{(0)} = h\big(\widehat{\hat{g}}_m^{(0)}(\boldsymbol{x}_t^{(0)})\big)$; observe $y_t^{(0)}$ and store data.

---

# 5   Parametric Utility Models: Similarity, Transfer, and Guarantee

We start with the linear setting that allows us to isolate and rigorously characterize the *transfer mechanism* itself, before introducing the additional complexity of nonlinear effects.

**Linear Utility Models.** Consider a linear model for the mean utility:

$$v_t^{(0)} = \boldsymbol{x}_t^{(0)} \cdot \boldsymbol{\beta}^{(0)} + \varepsilon_t, \tag{5}$$

where $\boldsymbol{\beta}^{(0)} \in \mathbb{R}^d$ denotes the coefficient vector. For source market data, we have for $k \in [K]$, $v_t^{(k)} = \boldsymbol{x}_t^{(k)} \cdot \boldsymbol{\beta}^{(k)} + \varepsilon_t$. To simplify the presentation, we impose the following assumption on parameter space.

**Assumption 3** (Parameter Boundedness). *We assume that $||x_t||_\infty \leq 1, \forall x_t \in \mathcal{X}$, and $||\boldsymbol{\beta}^{(k)}||_1 \leq W$ for a known constant $W \geq 1, \forall k \in 0 \cup [K]$. We denote by $\Omega$ the set of feasible parameters, i.e.,*

$$\Omega = \big\{\boldsymbol{\beta} \in \mathbb{R}^{d+1} : ||\boldsymbol{\beta}||_1 \leq W\big\}, \quad \boldsymbol{\beta}^{(k)} \in \Omega, \forall k \in 0 \cup [K].$$

We formalize the notion of similarity between source and target markets using the sparsity of the difference between coefficients.

**Assumption 4** (Task Similarity in Linear Model). *The maximum $l_0$-norm of the difference between target and source coefficients is bounded:*

$$\max_{k \in [K]} \|\boldsymbol{\beta}^{(0)} - \boldsymbol{\beta}^{(k)}\|_0 \leq s_0.$$

While our linear utility model itself is not necessarily sparse, Assumption 4 specifically constrains the *cross-market parameter differences* to be sparse, implying that at most $s_0$ covariates have significantly different effects across markets, mirroring the economic intuition that only certain latent features drive market variations.

***Bias-corrected Aggregation* for Linear Utility.** Algorithm 2 serves as the dual-mode estimator in Algorithms 1 and 4, switching between: (i) source data aggregation (no prior input), or (ii) $\ell_1$-regularized debiasing (given aggregate estimate).

---

**Algorithm 2:** Maximum Likelihood Estimation for Linear Utility Model

---

**Input:** Data $\{(p_t, \boldsymbol{x}_t, y_t)\}_{t \in [n]}$, aggregate estimate $\widehat{\boldsymbol{\beta}}^{(ag)}$

1 **if** $\widehat{\boldsymbol{\beta}}^{(ag)}$ *is None* **then**    // compute aggregate term

$$\widehat{\boldsymbol{\beta}} = \operatorname*{argmin}_{\mathbf{b}} \left\{ \frac{1}{n} \sum_{t=1}^{n} L(\mathbf{b}; p_t, \boldsymbol{x}_t, y_t)) \right\}$$

3 **else**    // compute debiasing term

$$\widehat{\boldsymbol{\beta}} = \operatorname*{argmin}_{\mathbf{b}} \left\{ \frac{1}{n} \sum_{t=1}^{n} L(\mathbf{b} + \widehat{\boldsymbol{\beta}}^{(ag)}; p_t, \boldsymbol{x}_t, y_t)) + \lambda_{tf} \|\mathbf{b}\|_1 \right\}$$

5 where the function

$$L(\mathbf{b}; p, \boldsymbol{x}, y) := -\left\{ \mathbb{1}(y = 1) \log(1 - F(p - \mathbf{b} \cdot \boldsymbol{x})) + \mathbb{1}(y = 0) \log(F(p - \mathbf{b} \cdot \boldsymbol{x})) \right\}. \quad (6)$$

**Output:** $\widehat{\boldsymbol{\beta}}$

---

In Algorithm 2, the aggregation step employs unregularized MLE. This choice is motivated by the fact that source market data are typically abundant, so the aggregate estimate can be reliably learned without imposing sparsity or other high-dimensional penalties. By contrast, the debiasing step operates on target market samples, which are relatively scarce. Here, we incorporate regularization to stabilize estimation and exploit structured similarities across markets.

### 5.1 Theoretical Guarantee for CM-TDP-O2O$_{\text{on}}$ under Linear Utility

The following theorem bounds the regret of our O2O$_{\text{on}}$ Policy under linear utility model.

**Theorem 5** (Regret Upper Bound for O2O$_{\text{on}}$ under Linear Utility). *Consider linear utility model (5) with Assumptions 1 (revenue regularity), 2 (covariate property), 3 (parameter space), and 4 (market similarity) holding true, the cumulative regret of Algorithm 1 admits the following bound:*

$$\text{Regret}(T; \pi) = \mathcal{O}\big(\frac{d}{K} \log d \log T + s_0 \log d \log T\big). \quad (7)$$

*where $K$ and $T$ denote the number of source markets and time horizon, respectively.*

We defer the complete proof to Appendix E. Theorem 5 reveals crucial insights about the role of source market quantity $K$ in two distinct operational regimes. First, in the *source-constrained regime* $(K \ll d/s_0)$, the first term dominates, showing that each additional source market provides linear reduction in regret. Notably, the logarithmic dependence on $T$ is consistent with classical linear bandit results [1], though our bound strictly improves their $\mathcal{O}(d \log T)$ through transfer. Second, in the *source-saturation regime* $(K \gg d/s_0)$, the second term becomes pivotal, quantifying the price for cross-market heterogeneity.

As illustrated in Figure 1, the empirical scaling behavior of regret w.r.t. both the number of source markets $K$ and time horizon $T$ precisely matches the theoretical predictions derived from Theorem 5.

While our theoretical guarantee is established in asymptotic regimes, the finite-horizon empirical performance robustly validates the practical effectiveness. This alignment between theory and practice confirms that our asymptotic analysis yields operationally meaningful insights for real-world applications.

**Theorem 6** (Regret Lower Bound under Linear Utility). *Consider the linear utility model in* (5)*, for any pricing policy $\pi$, the worst-case target-market regret over horizon $T$ satisfies*

$$\inf_{\pi} \ \sup \ \mathrm{Regret}(T;\pi) \ \geq \ c_1\frac{d}{K}\log T + c_2\,s_0\log\frac{d}{s_0}\log T, \tag{8}$$

*where the two constants $c_1, c_2$ depends only on noise distribution $F$, second moment matrix $\Sigma$, and parameter space $W$.*

In particular, (8) matches the upper bound (7) up to polylogarithmic factors in $d$, hence CM-TDP is minimax-type optimal in its $T$- and $K$-scaling.

# 6 Nonparametric Utility Models: Similarity, Transfer, and Guarantee

In this section, we model market utilities in an RKHS [3], leveraging (i) the kernel trick for efficient nonlinear computation [24], and (ii) its universal approximation power to capture rich market responses [26].

**Nonparametric Utility Models.** Let $\mathcal{H}_k$ be an RKHS induced by a symmetric, positive and semi-definite kernel function $K : \mathcal{X} \times \mathcal{X} \to \mathbb{R}$, and we define its equipped norm as $\| \cdot \|_K^2 = \langle \cdot, \cdot \rangle_K$ with the endowed inner product $\langle \cdot, \cdot \rangle_K$. We also define $K_x := K(x, \cdot) \in \mathcal{H}_k$. An important property in $\mathcal{H}_k$ is called the reproducing property, stating that $\langle g, K_x \rangle_K = g(x)$. The utility function now follows:

$$v_t^{(k)} = g^{(k)}(\boldsymbol{x}_t^{(k)}) + \varepsilon_t, \quad g^{(k)} \in \mathcal{H}_k, \ k \in 0 \cup [K], \tag{9}$$

where $g^{(k)}$ is the unknown target function and $\{\varepsilon_t^{(k)}\}_{t\geq 1}$ are i.i.d. noise with known distribution $F$. To start with, we place the following regularity assumptions on $\mathcal{H}_k$, which requires the model to be well-specified, and the kernel to be bounded.

**Assumption 7** (Regularity Condition). *Assume that $\|g^{(0)}\|_{\mathcal{H}_k} \leq R$, for some $R > 0$, and there exists a positive constant $\kappa$, such that the feature map $\phi(\boldsymbol{x}) = K(\boldsymbol{x}, \cdot)$ satisfies $\|\phi(\boldsymbol{x})\|_{\mathcal{H}_k} \leq \kappa, \forall \boldsymbol{x} \in \mathcal{X}$.*

**Assumption 8** (Task Similarity in RKHS). *For all $k \in [K]$, the discrepancy between the target task $g^{(0)}$ and the $k$-th source task $g^{(k)}$ in the RKHS norm is uniformly bounded as*

$$\max_{k\in[K]} \|g^{(0)} - g^{(k)}\|_K \leq H.$$

**Remark 2.** Assumption 8 characterizes the similarity between the target and source tasks through the bound $H$. A smaller value of $H$ indicates that the source tasks are more similar to the target task, which enables more effective knowledge transfer and potentially improves the estimation accuracy by leveraging information from related sources.

**Assumption 9** (Complexity). *Define the effective dimension as $\mathcal{N}(\lambda) := Tr[\Sigma(\Sigma+\lambda\boldsymbol{I})^{-1}]$, a variant of what is typically used to characterize the complexity of RKHS [6]. We assume:*
(i) *There exist some constants $\alpha > 1/2$ such that $\mathcal{N}(\lambda) = Tr(\Sigma(\Sigma + \lambda I)^{-1}) \lesssim \lambda^{-1/(2\alpha)}$.*

(ii) *There exist some constants $\beta \in (0,1]$ such that for each $k \in 0 \cup [K]$, there holds $g^{(k)} = \Sigma^{\beta}\rho^{(k)}$, for some $\rho^{(k)} \in \mathcal{L}_2(\mathcal{X}, \mathcal{P}_x)$, where $\Sigma, \mathcal{P}_x$ are defined in Assumption 2.*

**Remark 3.** (i) controls the complexity of the considered $\mathcal{H}_k$. Smaller $\alpha$ means slower eigenvalue decay and higher intrinsic dimensionality. (ii) is a regularity condition on the source and target functions, and is also commonly assumed in literature[6, 25, 17]. Here $\beta > 0$ controls the degree of smoothness, and larger $\beta$ means $g^{(k)}$ is smoother and easier to estimate.

***Bias-corrected Aggregation*** **for Nonparametric Utility.** Algorithm 3 presents the aggregation and debiasing operations using regularized kernel regression.

## 6.1 Theoretical Guarantee for CM-TDP-O2O$_{\mathrm{on}}$ under Nonparametric Utility

The following theorem bounds the regret of our O2O$_{\mathrm{on}}$ Policy under RKHS utility model.

---

**Algorithm 3:** Kernel Regression for RKHS Utility Model

---

**Input:** Data $\{(p_t, \boldsymbol{x}_t, y_t)\}_{t \in [n]}$, Kernel $K(\boldsymbol{x}, \boldsymbol{x}') = \langle \phi(\boldsymbol{x}), \phi(\boldsymbol{x}') \rangle$, Aggregated estimator $\widehat{g}^{(ag)} \in \mathcal{H}_k$.

**1 if** $\widehat{g}^{(ag)}$ *is None* **then**                                                               // compute aggregation term

$$\widehat{g} = \underset{g \in \mathcal{H}_k}{\operatorname{argmin}} \left\{ \frac{1}{n} \sum_{t=1}^{n} L(g; p_t, \boldsymbol{x}_t, y_t) + \lambda_{ag} \|g\|_{\mathcal{H}_k}^2 \right\}$$

**3 else**                                                                                                          // compute debiasing term

$$\widehat{g} = \underset{g \in \mathcal{H}_k}{\operatorname{argmin}} \left\{ \sum_{t=1}^{n} L(g + \widehat{g}^{(ag)}; p_t, \boldsymbol{x}_t, y_t) + \lambda_{tf} \|g\|_{\mathcal{H}_k}^2 \right\}$$

**5 where the function**

$$L(g; p, \boldsymbol{x}, y) := -\Big\{ \mathbb{1}(y = 1) \log(1 - F(p - g(\boldsymbol{x}))) + \mathbb{1}(y = 0) \log(F(p - g(\boldsymbol{x}))) \Big\}.$$

**Output:** $\widehat{g}$

---

**Theorem 10** (Regret Upper Bound for O2O$_{\text{on}}$ under RKHS Utility). *Consider RKHS utility model (9) with Assumptions 1 (revenue regularity), 2 (covariate property), 7 (parameter space), 8 (market similarity), and 9 (complexity) holding true, the cumulative regret of Algorithm 1 admits the following bound:*

$$\text{Regret}(T; \pi) = \mathcal{O}\left( K^{-\frac{2\alpha\beta}{2\alpha\beta+1}} T^{\frac{1}{2\alpha\beta+1}} + H^{\frac{2}{2\alpha+1}} T^{\frac{1}{2\alpha+1}} \right) \tag{10}$$

*where $K$ and $T$ denote the number of source markets and time horizon, respectively.*

Theorem 10 again highlights the benefit of transfer learning, and finds clear empirical support in the experiments shown in Figure 2. The first term reflects the learning complexity of the target function, where $\beta$ quantifies its intrinsic complexity with larger $\beta$ indicating simpler functions and enabling faster transfer. The $\alpha$ parameter, as the kernel's effective dimension, modulates how $\beta$ impacts the exponent. The second term encodes cross-market disparity through $H$, which directly measures the worst-case RKHS distance between source and target markets. The $\alpha$-dependence shows that high-dimensional RKHS amplify heterogeneity costs. Compared to Theorem 5, we observe polynomial rather than logarithmic $T$-dependence, reflecting the fundamental difficulty shift from parametric to nonparametric estimation.

**Special Cases.** The following boundary cases demonstrate the degradation properties of our theoretical results, showing how the general bound naturally adapts to different simpler scenarios.

**Perfect Task Similarity.** When source and target domains are identical ($H = 0$), the bound becomes:

$$\text{Regret}(T; \pi) = \mathcal{O}\left( K^{-\frac{2\alpha\beta}{2\alpha\beta+1}} T^{\frac{1}{2\alpha\beta+1}} \right)$$

The regret depends on the total sample size from all sources. The rate improves with number of source market $K$.

**No Transfer Learning** ($K_{\text{eff}} = 1, H = 0$) In the absence of transfer learning, *i.e.*, when no source domain data is available ($K = 0, H = 0$), we evaluate the bound (10) with $K_{\text{eff}} = \max\{K, 1\} = 1$ (equivalently, a "self-aggregation" that ignores external sources). Our general framework reduces to conventional dynamic pricing with RKHS utility functions. The regret bound simplifies to:

$$\text{Regret}(T; \pi) = \mathcal{O}\left( T^{\frac{1}{2\alpha\beta+1}} \right).$$

While existing literature has not explicitly analyzed regret bounds for dynamic pricing with RKHS utility functions, we can establish an important connection to the well-studied linear case. By setting $\alpha \to 1/2^+$ and $\beta = 1$, which corresponds to linear utility functions, the general bound (10) reduces to $\mathcal{O}(\sqrt{T})$, matching the often-seen regret for online decision-making problems with linear structures [1, 19].

This $\mathcal{O}(\sqrt{T})$ rate should be contrasted with the $\mathcal{O}(\log T)$ regret we established for the linear transfer setting in (7). The difference stems from the underlying estimation complexity: in the parametric case, the generalized linear model is finite-dimensional, and abundant source data together with MLE-based updates enable nearly logarithmic regret growth. In contrast, the RKHS formulation

must estimate an infinite-dimensional function under binary feedback, where nonparametric learning is intrinsically harder. Thus, the gap reflects fundamental differences between parametric and nonparametric estimation, rather than looseness in the analysis.

**Theorem 11** (Regret Lower Bound under RKHS Utility). *Consider the RKHS utility model in* (9). *For any pricing policy $\pi$, then there exists a constant $c > 0$ depending only on $(F, P_x, K)$ via $(m_{\mathrm{rev}}, m_h, \kappa)$ and the Bernoulli KL smoothness constant (defined in Lemma 28) such that for all horizons $T \geq 1$,*

$$\inf_\pi \sup \mathrm{Regret}(T; \pi) \geq c \left\{ K^{-\frac{2\alpha\beta}{2\alpha\beta+1}} T^{\frac{1}{2\alpha\beta+1}} + H^{\frac{2}{2\alpha+1}} T^{\frac{1}{2\alpha+1}} \right\}. \tag{11}$$

In particular, (11) matches the upper bound (10) up to polylogarithmic factors, hence CM-TDP is minimax-type optimal in its $T$- and $K$-scaling.

# 7 Numerical Experiments

We evaluate CM-TDP through extensive simulations covering multiple market scenarios and dimensionalities:

*(1) Identical Markets*: an ideal baseline where $\boldsymbol{\beta}^{(0)} \equiv \boldsymbol{\beta}^{(k)}$ or $g^{(0)} \equiv g^{(k)}$ for all source markets;

*(2) Sparse difference Markets*: for linear utility, we implement Assumption 4 with $\|\boldsymbol{\beta}^{(0)} - \boldsymbol{\beta}^{(k)}\|_0 \leq 0.3 * d$; for RKHS utility, we implement Assumption 8 with $\|g^{(0)} - g^{(k)}\|_{\mathcal{H}_k} \leq 0.3$.

*(3) Dense difference Markets*: for linear utility, we implement Assumption 4 with $\|\boldsymbol{\beta}^{(0)} - \boldsymbol{\beta}^{(k)}\|_0 \leq 0.5 * d$; for RKHS utility, we implement Assumption 8 with $\|g^{(0)} - g^{(k)}\|_{\mathcal{H}_k} \leq 0.5$.

In each market scenario, we test $T = 2000$ periods with dimensions $d \in \{10, 15, 20, 100\}$ and $K \in \{1, 3, 5, 10\}$ source markets. RKHS function (RBF kernel with $\gamma = 0.5$) and kernel parameters ($\kappa = 0.5$, $R = 1.0$) remain consistent across all experiments. We evaluate O2O$_{\mathrm{on}}$ Policy against no transfer baseline [19], a standard dynamic pricing using only target market data. Market noise follows logistic distribution on $\mathbb{R}$. For numerical stability, we clip simulated valuations to $[-B_\varepsilon, B_\varepsilon]$ with $B_\varepsilon = 1$. The code is available at https://github.com/CS-SAIL/transfer_pricing_neurips2025.

**Simulation Results.** Figures 1 and 2 present the empirical cumulative regret for linear and RKHS utility models with $d = 10$ and $100$, respectively, demonstrating three key findings (For conciseness, results for $d = 15, 20$ are deferred to Appendix D.1).

**(1) Universal effectiveness over non-transfer scheme**. CM-TDP-O2O$_{\mathrm{on}}$ consistently outperforms single-market baseline in all scenarios. Significant improvements emerge even with minimal source markets ($K = 1$), with larger $K$ values enhance robustness in divergent market conditions. Against single-market learning, CM-TDP reduces cumulative regret by roughly 43–55% on average (peaking at 75%), reduces standard error by about 24–31% (up to 39 %), and attains the same estimation error level as much as $9 \times$ sooner (Table 2 in Appendix D.2.1).

**(2) Adaptivity**. The transfer mechanism automatically handles both identical markets and sparse-difference markets, without requiring manual adjustments, with identical market condition achieving faster convergence than sparse difference cases and dense difference cases.

**(3) Scalability**: Higher dimensions maintain stable performance, confirming the method's scalability.

These results collectively validate our framework as a versatile solution for real-world dynamic pricing, particularly in environments with varying market similarities.

# 8 Conclusion and Future Work

We introduce *Cross-Market Transfer Dynamic Pricing* (CM-TDP), the first framework that provably accelerates revenue learning by pooling information from multiple auxiliary markets in both Offline-to-Online and Online-to-Online regimes. CM-TDP achieves minimax-optimal regret for *both* linear and RKHS utilities, matching information-theoretic lower bounds, and delivers up to an average of 50% lower cumulative regret and $5 \times$ faster learning in extensive simulations. These results bridge single-market pricing, meta-learning, and multitask bandits, laying the groundwork for pricing systems that *"transfer faster, price smarter"*.

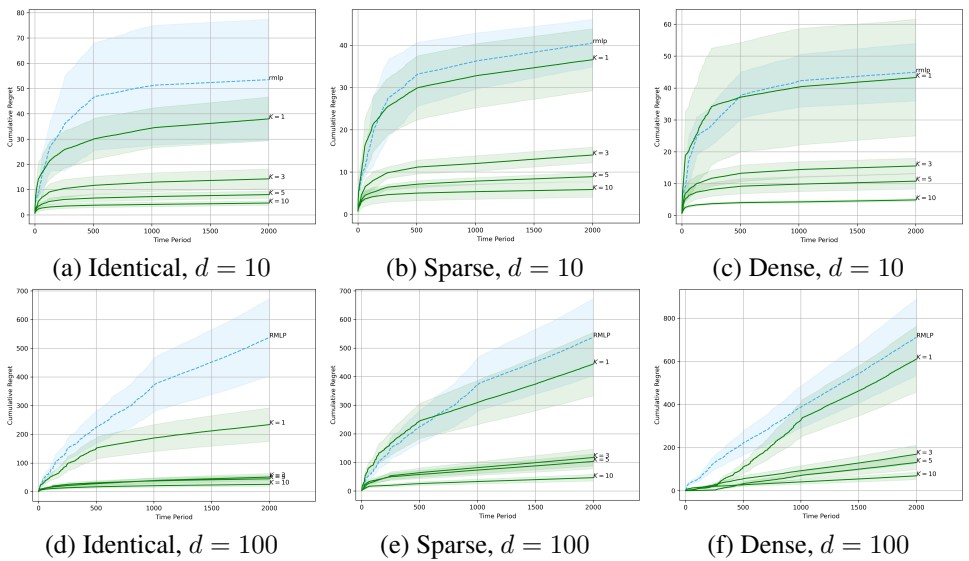

Figure 1: Cumulative regret across experimental conditions in O2O$_{\text{on}}$ with linear utility model.

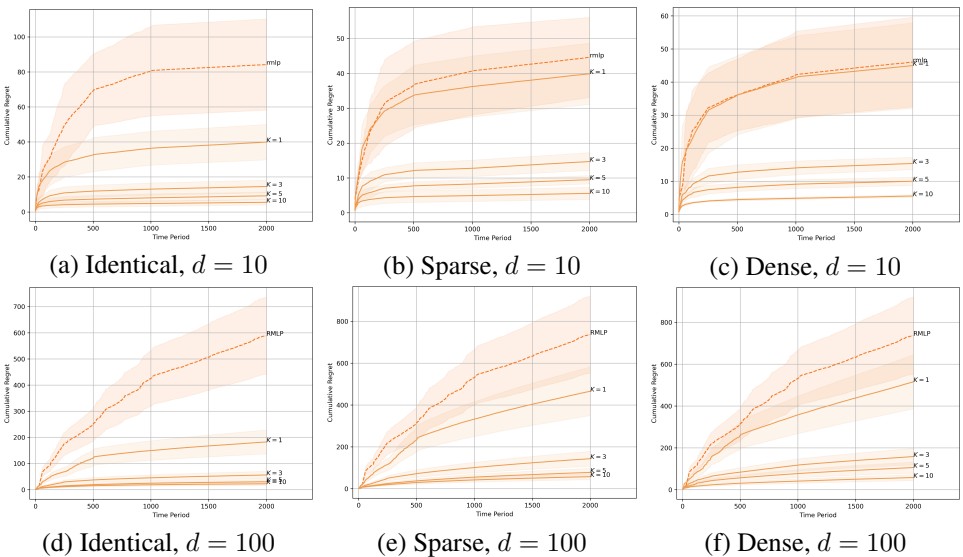

Figure 2: Cumulative regret across experimental conditions in O2O$_{\text{on}}$ with RKHS utility model.

Several extensions of CM-TDP provide fertile ground for future research. First, relaxing the assumption of homogeneous covariate distributions would allow the framework to handle domain shift, e.g., via reweighting, importance sampling, or domain-invariant representation learning. Second, while we focus on $\ell_0$-sparsity for interpretability, the aggregation framework naturally extends to richer similarity notions such as $\ell_q$-sparsity ($q \in [0, 1]$), smoothness metrics, or distributional divergences. Third, CM-TDP currently lacks mechanisms to down-weight or exclude adversarial source markets with large parameter gaps, and developing robust similarity detection and market-selection strategies remains an important direction.

## Acknowledgments and Disclosure of Funding

We thank the NeurIPS reviewers for their helpful comments. This work was supported by NSF Award No. 2412577 and the NYU Stern Research Fund.

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

# A   Notation

Let lowercase letter $x$, boldface letter $\boldsymbol{x}$, boldface capital letter $\boldsymbol{X}$, and blackboard-bold letter $\mathbb{X}$ represent scalar, vector, matrix, and tensor, respectively. The calligraphy letter $\mathcal{X}$ represents operator. We use the notation $[N]$ to refer to the positive integer set $\{1, \ldots, N\}$ for $N \in \mathbb{Z}_+$. Let $C, c, C_0, c_0, \ldots$ denote generic constants, where the uppercase and lowercase letters represent large and small constants, respectively. The actual values of these generic constants may vary from time to time.

All vectors are column vectors and row vectors are written as $\boldsymbol{x}^\top$ for any vector $\boldsymbol{x}$. For any vector $\boldsymbol{x} = (x_1, \ldots, x_p)^\top$, let $\|\boldsymbol{x}\| := \|\boldsymbol{x}\|_2 = (\sum_{i=1}^p x_i^2)^{1/2}$ be the $\ell_2$-norm, and let $\|\boldsymbol{x}\|_1 = \sum_{i=1}^p |x_i|$ be the $\ell_1$-norm.

For any matrix $\boldsymbol{X}$, we use $\boldsymbol{x}_{i\cdot}$, $\boldsymbol{x}_j$, and $x_{ij}$ to refer to its $i$-th row, $j$-th column, and $ij$-th entry, respectively. For two matrices $\boldsymbol{X}_1 \in \mathbb{R}^{m \times n}$ and $\boldsymbol{X}_2 \in \mathbb{R}^{p \times q}$, $\boldsymbol{X}_1 \otimes \boldsymbol{X}_2 \in \mathbb{R}^{pm \times qn}$ is the Kronecker product. When $\boldsymbol{X}$ is a square matrix, we denote by $\mathrm{Tr}\,(\boldsymbol{X})$, $\lambda_{max}\,(\boldsymbol{X})$, and $\lambda_{min}\,(\boldsymbol{X})$ the trace, maximum and minimum singular value of $\boldsymbol{X}$, respectively. For two matrices of the same dimension, define the inner product $\langle \boldsymbol{X}_1, \boldsymbol{X}_2 \rangle = \mathrm{Tr}(\boldsymbol{X}_1^\top \boldsymbol{X}_2)$.

We use $\mathcal{L}_2(\mathcal{X}, \mathcal{P}_x) = \left\{ f : \int_{\mathcal{X}} f^2(x) d\mathcal{P}_x < \infty \right\}$ to denote the space of square-integrable functions with respect to $\mathcal{P}_x$, equipped with the inner product $\langle f, g \rangle_{\mathcal{P}_x} = \int_{\mathcal{X}} f(x)g(x) d\mathcal{P}_x$ and squared norm $\|f\|_{\mathcal{P}_x}^2 = \int_{\mathcal{X}} f^2(x) d\mathcal{P}_x$.

| Notation | Definition |
|---|---|
| $t$ | Time index (period). |
| $T$ | Time horizon (total number of periods). |
| $m$ | Episode index (algorithmic episode). |
| $\ell_m$ | Length of episode $m$, $\ell_m = 2^{m-1}$. |
| $\mathcal{T}_m$ | Set of time indices in episode $m$. |
| $K$ | Number of source markets. |
| $k$ | Source market index, $k \in [K]$. |
| $n_0$ | Number of target samples. |
| $n_K$ | Total number of offline source samples (used in O2O$_{\mathrm{off}}$ analysis). |
| $x_t^{(0)}, p_t^{(0)}, y_t^{(0)}, v_t^{(0)}$ | Observed contextual covariates, posted price, observed binary sale indicator, latent market utility, for the target market at time $t$. |
| $x_t^{(k)}, p_t^{(k)}, y_t^{(k)}, v_t^{(k)}$ | Observed contextual covariates, posted price, observed binary sale indicator, latent market utility, for source market $k$ at time $t$. |
| $\mathring{g}^{(0)}(\cdot)$ | General unknown mean-utility function for target market. |
| $g^{(k)}(\cdot)$ | General unknown mean-utility function for source market $k$. |
| $\varepsilon_t$ | i.i.d. noise at time $t$ with distribution $F(\cdot)$. |
| $F(\cdot)$ | Cumulative distribution function of the noise $\varepsilon_t$. |
| $\Sigma$ | Covariance matrix for source and target features, which is equivalent to $\Sigma^{(0)}, \Sigma^{(k)}$ in homogeneous covariate setting. |
| $C_{\min}, C_{\max}$ | Minimum and maximum eigenvalues of second-moment matrix $\Sigma$. |
| $\widehat{\mathring{g}}_m^{(0)}$ | Debiased estimator for the target after aggregation+debias at episode $m$. |
| $\delta^{(0)}$ | True (population) debiasing correction in RKHS. |
| $\widehat{\delta}_m$ | Debiasing correction term computed at episode $m$. |
| $\lambda_{tf}$ | Regularization parameter used for the debiasing step. |
| $\lambda_{ag}$ | Regularization parameter used in aggregation. |

| Notation | Definition |
|---|---|
| $\beta^{(0)}$ | Coefficient vector for the linear mean utility in the target market. |
| $\beta^{(k)}$ | Coefficient vector for the linear utility in source market $k$. |
| $\widehat{\beta}$ | Estimated coefficient for target market. |
| $s_0$ | Task-similarity magnitude in linear utility case. |
| $g^{(0)}$ | RKHS mean-utility function of the target market. |
| $g^{(k)}$ | RKHS mean-utility function of source market $k$. |
| $H$ | Task-similarity magnitude in RKHS case. |
| $\mathcal{H}_k$ | RKHS associated with kernel $K$. |
| $K(\cdot,\cdot)$ | Positive semi-definite kernel function inducing $\mathcal{H}_k$. |
| $N(\lambda)$ | Effective dimension. |
| $\alpha$ | Parameter controlling eigenvalue decay. |
| $\beta$ | Smoothness parameter for RKHS function. |

# B O2O$_{\text{off}}$: Offline-to-Online Cross-Market Transfer Pricing

In this section, we present our algorithmic framework, theoretical results and empirical experiments for O2O$_{\text{off}}$ (Offline-to-Online) scenario.

## B.1 Offline-to-Online (O2O$_{\text{off}}$) Algorithm

Prior to deployment we form an aggregate estimator $\widehat{g}^{(\text{ag})}$ from *all* source data. At the start of episode $m = 1$ this estimator is debiased with the first batch of target observations to obtain $\widehat{g}_1^{(0)}$; subsequent episodes rely exclusively on target data. Hence O2O$_{\text{off}}$ yields a one-shot reduction of cold-start regret, but its asymptotic rate in $T$ matches that of single-market learning. The complete procedure is summarize in Algorithm 4.

## B.2 Theoretical Guarantee for CM-TDP-O2O$_{\text{off}}$ under Linear Utility

The following theorem bounds the regret of our O2O$_{\text{off}}$ Policy under linear utility model.

**Theorem 12** (Regret Upper Bound for O2O$_{\text{off}}$ under Linear Utility). *Consider linear utility model (5) with Assumptions 1 (revenue regularity), 2 (covariate property), 3 (parameter space), and 4 (market similarity) holding true, the cumulative regret of Algorithm 4 admits the following bound:*

$$Regret(T; \pi) = \mathcal{O}\left(d \log d \log T + (s_0 - d) \log d \log n_{\mathcal{K}}\right). \tag{12}$$

*where $n_{\mathcal{K}}$ and $T$ denote the number of source data and time horizon, respectively.*

Theorem 12 reveals fundamental differences in how static source data influences regret compared to Online-to-Online transfer. The bound in (12) decomposes into two interpretable components: the first term captures the intrinsic complexity of learning the $d$-dimensional target market parameters in the total $T$ periods, while the second term quantifies the net effect during transfer learning phase. Given that $s_0 < d$ by problem construction, the second term always provides a *regret reduction* proportional to $(d - s_0) \log n_{\mathcal{K}}$. This reduction grows logarithmically with the total source sample size $n_{\mathcal{K}}$, which is consistent with our numerical experiments in Figure 3.

## B.3 Theoretical Guarantee for CM-TDP-O2O$_{\text{off}}$ under Nonparametric Utility

The following theorem bounds the regret of our O2O$_{\text{off}}$ Policy under RKHS utility model.

**Theorem 13** (Regret Upper Bound for O2O$_{\text{off}}$ under RKHS Utility). *Consider RKHS utility model (9) with Assumptions 1 (revenue regularity), 2 (covariate property), 7 (parameter space), 8 (market*

---

**Algorithm 4:** CM-TDP-O2O$_{\text{off}}$

---

**Input:** Offline source market data $\{(p_t^{(k)}, \boldsymbol{x}_t^{(k)}, y_t^{(k)})\}_{t \in \mathcal{H}^{(k)}}$ for $k \in [K]$; feature matrix $\{\boldsymbol{x}_t^{(0)}\}_{t \in \mathbb{N}}$ for the target market

```
/* ******* Phase 1:  Update with transfer learning *******                  */
```

1 Call Algorithm 2 or 3 to calculate the initial aggregated estimate $\widehat{\hat{g}}^{(\text{ag})}$ using entire source market data $\{(\boldsymbol{p}_t^{(k)}, \boldsymbol{X}_t^{(k)}, \boldsymbol{y}_t^{(k)})\}_{t \in \mathcal{H}^{(k)}}$ for $k \in [K]$

2 Apply the price $\widehat{p}_1^{(0)} := h(\widehat{\hat{g}}^{(\text{ag})}(\boldsymbol{x}_1^{(0)}))$ and collect data $(\widehat{p}_1^{(0)}, \boldsymbol{x}_1^{(0)}, y_1^{(0)})$.

3 **for** *each episode* $m = 2, \ldots, m_0$ **do**

4     Set the length of the $m$-th episode: $\ell_m := 2^{m-1}$

5     Call Algorithm 2 or 3 to calculate the debiasing estimate $\widehat{\delta}_m$ using target market data $\{(p_t^{(0)}, \boldsymbol{x}_t^{(0)}, y_t^{(0)})\}_{t \in [2^{m-2}, 2^{m-1}-1]}$ and aggregated estimate $\widehat{\hat{g}}^{(\text{ag})}$.

6     Set
$$\widehat{\hat{g}}_m^{(0)} := \widehat{\hat{g}}^{(\text{ag})} + \widehat{\delta}_m.$$

7     For each time $t$, apply price $\widehat{p}_t^{(0)} := h(\widehat{\hat{g}}_m^{(0)}(\boldsymbol{x}_t^{(0)}))$ and collect data $(\widehat{p}_t^{(0)}, \boldsymbol{x}_t^{(0)}, y_t^{(0)})$.

```
/* ******* Phase 2:  Update without transfer learning *******               */
```

8 **for** *each* $m \geq m_0 + 1$ **do**

9     Set the length of the $m$-th episode: $\ell_m := 2^{m-1}$

10     Call Algorithm 2 or 3 to calculate $\widehat{\hat{g}}_m^{(0)}$ using target market data $\{(p_t^{(0)}, \boldsymbol{x}_t^{(0)}, y_t^{(0)})\}_{t \in [2^{m-2}, 2^{m-1}-1]}$.

11     For each time $t$, apply price and collect data.

**Output:** Offered price $\widehat{p}_t^{(0)}, t \geq 1$

---

*similarity), and 9 (complexity) holding true, the cumulative regret of Algorithm 4 admits the following bound:*

$$Regret(T; \pi) = \mathcal{O}\left( \widetilde{c}n_{\mathcal{K}}^{\frac{1}{2\alpha\beta+1}} + H^{\frac{2}{2\alpha+1}} (\widetilde{c}n_{\mathcal{K}})^{\frac{1}{2\alpha+1}} + T^{\frac{1}{2\alpha\beta+1}} - (\widetilde{c}n_{\mathcal{K}})^{\frac{1}{2\alpha\beta+1}} \right) \tag{13}$$

*where $n_{\mathcal{K}}$ denotes the number of source data, $\beta$ characterizes the smoothness of the aggregated utility function $g^{(ag)} = \Sigma^\beta g$ with $\beta \in (0, 1]$, and $\alpha > 1/2$ controls the effective dimension via the eigenvalue decay $N(\lambda) \lesssim \lambda^{-1/(2\alpha)}$.*

The derived result reveals an important trade-off in the regret decomposition. The first component $\mathcal{O}\left( \widetilde{c}n_{\mathcal{K}}^{\frac{1}{2\alpha\beta+1}} + H^{\frac{2}{2\alpha+1}} (\widetilde{c}n_{\mathcal{K}})^{\frac{1}{2\alpha+1}} \right)$ captures regret resulting from the transfer learning phase. This component grows with $\widetilde{c}n_{\mathcal{K}}$ because more source data leads to a longer transfer phase duration, which consequently increases the accumulated regret during this phase. However, this initial cost is offset by a more significant benefit in the second component: $\mathcal{O}\left( T^{\frac{1}{2\alpha\beta+1}} - (\widetilde{c}n_{\mathcal{K}})^{\frac{1}{2\alpha\beta+1}} \right)$, which demonstrates that the extended transfer phase enables substantially more effective single-market learning, and thus reduces the overall regret.

We defer the complete technical proof of Theorems 12 and 13 to Appendix F and I, respectively.

## B.4   Numerical Experiments

The experimental setup for O2O$_{\text{off}}$ maintains the same core structure as the O2O$_{\text{on}}$ setting, except for data collection protocol. Rather than observing synchronized data streams from $K$ active source markets, we begin with a fixed historical dataset of size $n_{\mathcal{K}} \in \{50, 100, 200, 500\}$. All other experimental parameters, including demand model specification, evaluation metrics, and comparison baselines, remain consistent with the O2O$_{\text{on}}$ described in Section 7.

All experiments were conducted on an Ubuntu 20.04 server equipped with an AMD Ryzen 9 5950X CPU (16 cores, 32 threads), 125 GiB RAM, and an NVIDIA RTX 3090 GPU (24 GiB VRAM). The primary storage was a 1.8 TB NVMe SSD.

**Simulation Results**   Figures 3 and 4 present the empirical cumulative regret for linear and RKHS utility models, respectively. As is consistent with our theoretical analysis in Theorems 12 and 13, we observe a *jump-start* benefit in the early stage, and a significantly lower overall regret compared to the single-market baseline.

The results again demonstrate that all transfer learning variants consistently outperform the no-transfer baseline across all tested conditions, with larger $n_{\mathcal{K}}$ values showing faster regret reduction, validating the effectiveness of knowledge transfer from source markets.

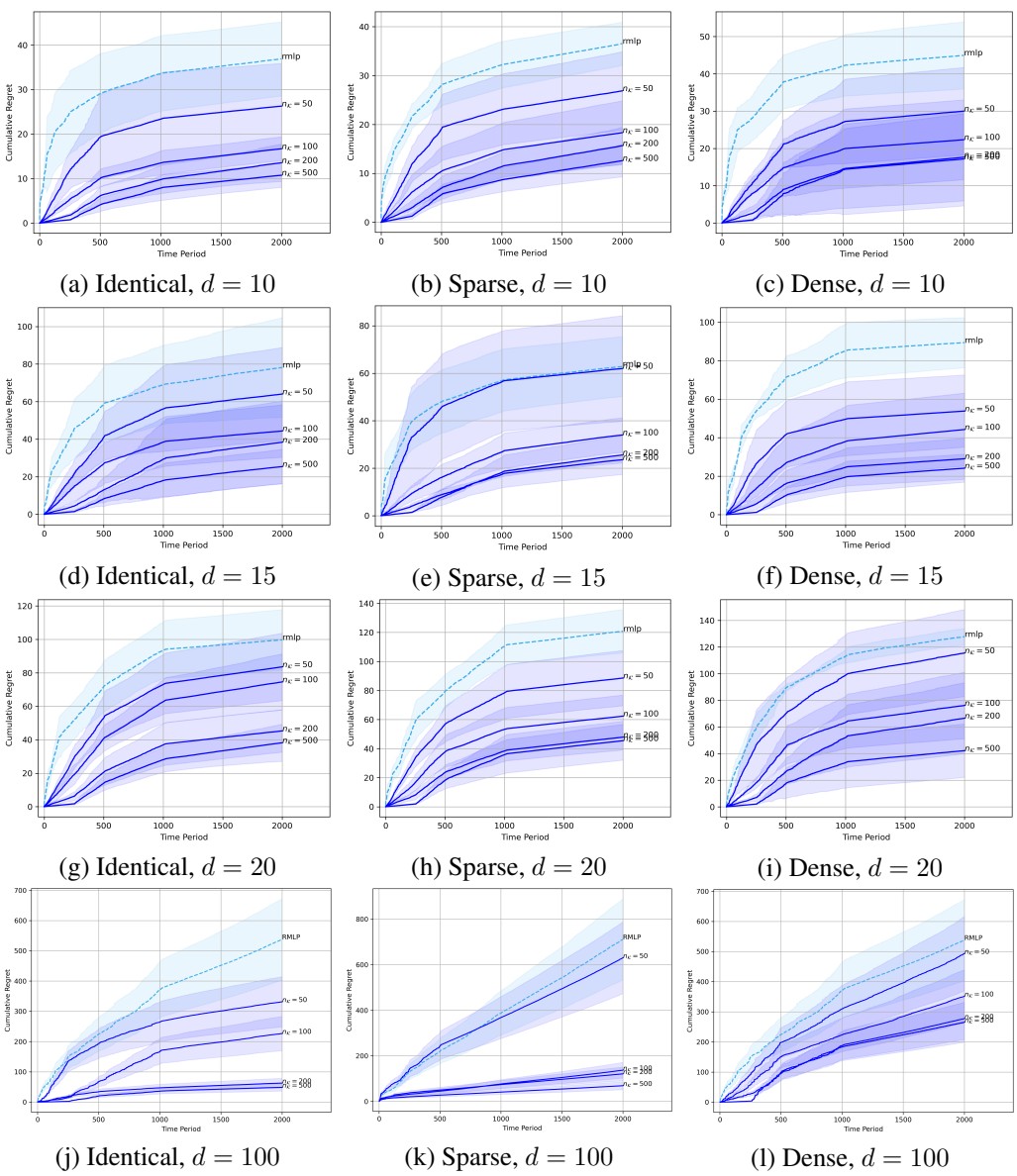

Figure 3: Cumulative regret across experimental conditions in O2O$_{\text{off}}$ with linear utility model.

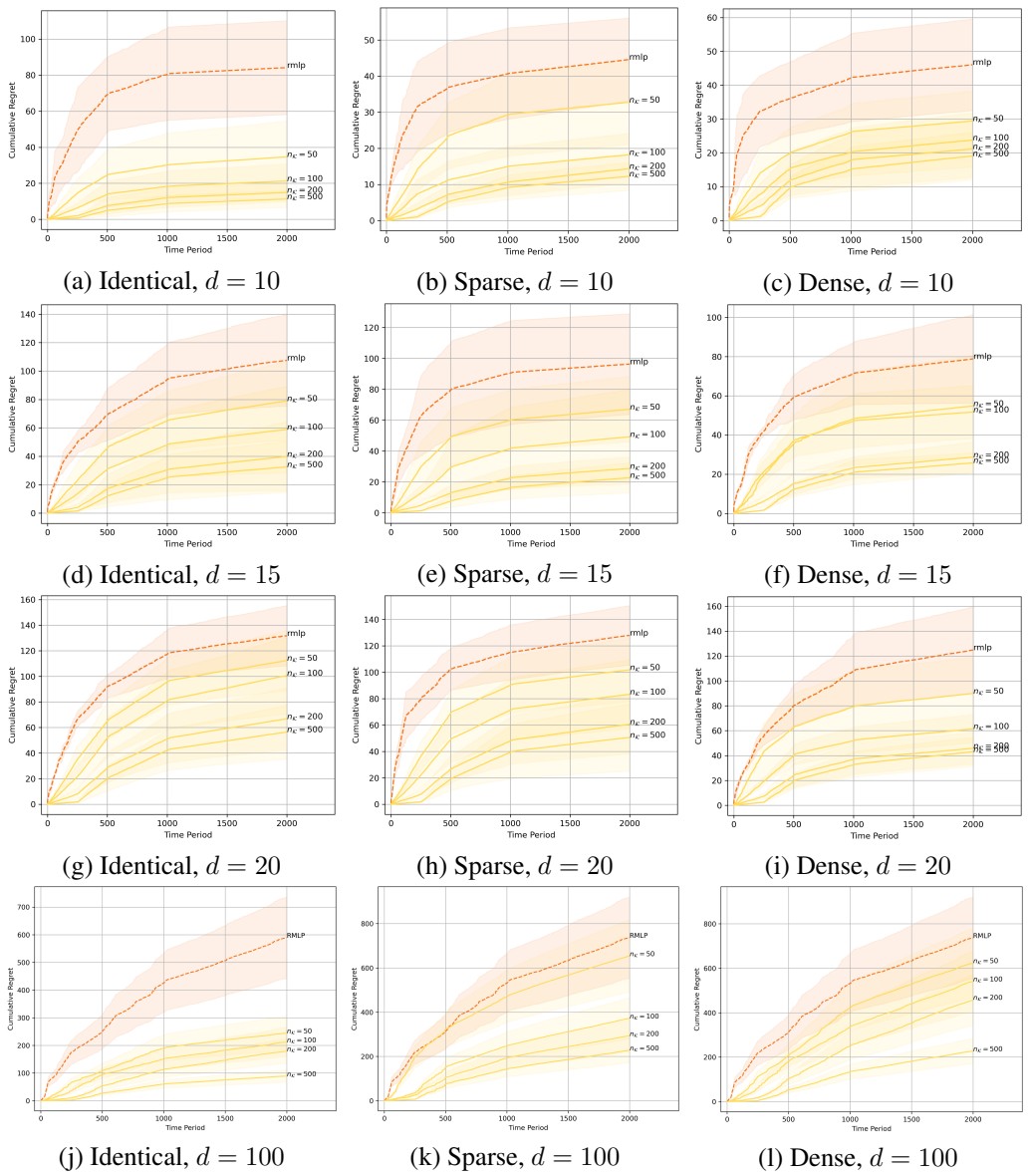

Figure 4: Cumulative regret across experimental conditions in O2O$_{off}$ transfer with RKHS utility model.

## C   Computational Complexity Analysis

In this section, we summarize the computational costs of CM-TDP across different settings.

**Linear Utility Model.**    In O2O$_{on}$, each episode consists of two stages. In the aggregation stage, unregularized maximum likelihood estimation on the aggregated source data has a complexity of $\mathcal{O}(d^2\ell_{m-1}K \cdot N)$, where $d$ is the feature dimension, $\ell_{m-1}$ is the number of past samples, $K$ is the number of source markets, and $N$ is the number of optimization iterations. In the debiasing stage, Lasso regression on the target data requires $\mathcal{O}(d^2\ell_{m-1} \cdot N)$. Over $T$ episodes, the cumulative regret analysis involves $\mathcal{O}(d^2TK \cdot N)$ operations.

For O2O$_{off}$, Phase 1 (transfer) requires $\mathcal{O}(d^2n_{\mathcal{K}} \cdot N)$ for MLE on the aggregated source market data, plus $\mathcal{O}(d^2\ell_{m-1} \cdot N)$ per episode for bias correction. Phase 2 (no transfer) reduces to standard linear MLE with complexity $\mathcal{O}(d^2\ell_{m-1} \cdot N)$. The resulting cumulative complexity across $T$ episodes is

$\mathcal{O}(d^2(n_{\mathcal{K}} + T) \cdot N)$. The dependence on $d$ depends on the optimization method: Newton's method (used in our implementation) achieves faster convergence (smaller $N$) but maintains $d^2$ dependence, whereas gradient descent reduces the per-iteration cost to $\mathcal{O}(d)$ at the expense of a larger iteration count $N$.

**RKHS Utility Model.** Exact kernel methods scale as $\mathcal{O}(n^3)$ in the number of aggregated samples $n$. In O2O$_{\text{on}}$, episode $m$ uses $n_m \simeq K \cdot 2^{m-1}$ source samples and $2^{m-1}$ target samples, so a naive solver scales as $\mathcal{O}(n_m^3)$. In practice, Nyström or sketching reduces runtime to near-linear in the effective dimension $\mathcal{N}(\lambda)$.

# D  Additional Experimental Results

## D.1  Extended Regret Plots for O2O$_{\text{on}}$

In the main text, we reported regret trajectories for O2O$_{\text{on}}$ with dimensions $d = 10, 100$. Here, we provide additional results for intermediate-dimensional settings ($d = 15, 20$)in Figures 5 and 6. As shown in the plots, CM-TDP consistently outperforms the no-transfer baseline across different dimensionalities, maintaining lower cumulative regret and faster convergence. These results confirm that the effectiveness of CM-TDP is not restricted to specific dimensions.

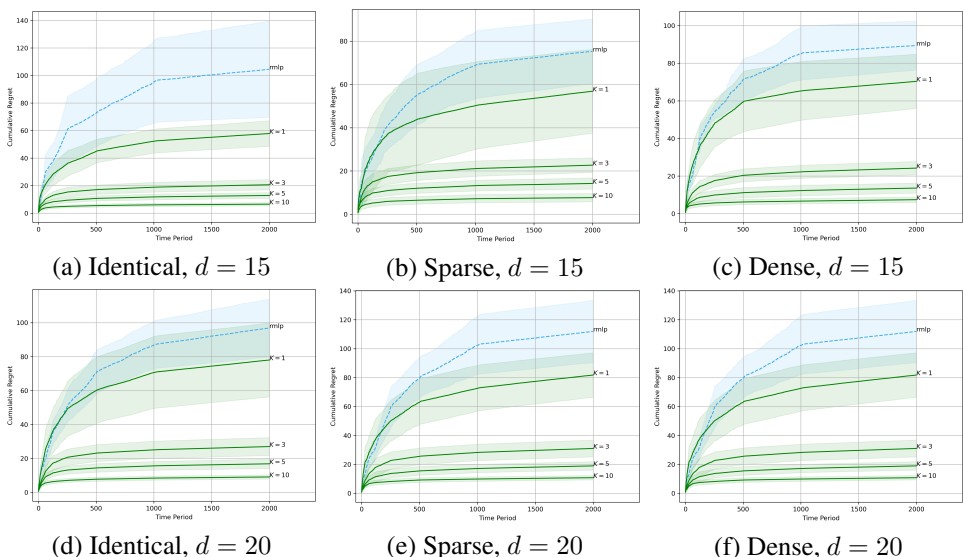

(a) Identical, $d = 15$     (b) Sparse, $d = 15$     (c) Dense, $d = 15$

(d) Identical, $d = 20$     (e) Sparse, $d = 20$     (f) Dense, $d = 20$

Figure 5: Cumulative regret across experimental conditions in O2O$_{\text{on}}$ with linear utility model.

## D.2  In-depth Analysis of Sparse-difference Markets

While our regret evaluation covers identical, sparse-difference, and dense-difference market scenarios, this part focus on the sparse-difference case. This choice reflects the primary target of our algorithm design: markets that differ sparsely in latent preferences, where transfer is both practically relevant and theoretically most distinctive. At the same time, our earlier regret plots already demonstrate that CM-TDP yields consistent improvements across identical and dense-difference settings as well, confirming that the additional deep-dive into sparse-difference scenarios is representative rather than restrictive.

### D.2.1  Benchmarking Transfer Gains

Table 2 compares CM-TDP against the canonical single-market learner in the *sparse-difference* setting, where only a small subset of coefficients differs between the target and each source market. We report three metrics averaged over dimensions $d \in \{10, 15, 20\}$: *reg* (percentage reduction in

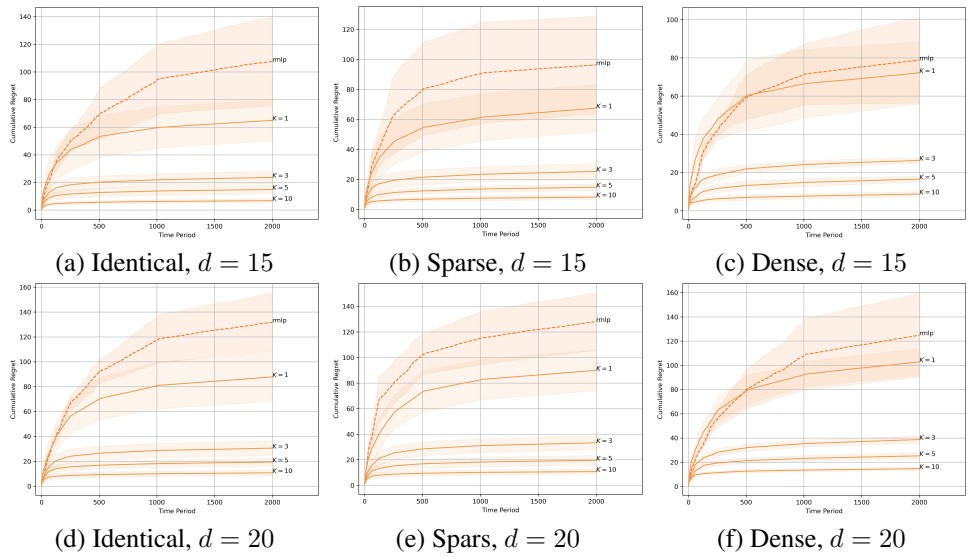

Figure 6: Cumulative regret across experimental conditions in O2O$_{on}$ with RKHS utility model.

cumulative regret, i.e., revenue lift), *std* (percentage reduction in standard error across 10 Monte-Carlo runs), and *speed* (multiplicative acceleration in reaching the single-market learner's final estimation error, i.e., $\|\widehat{\beta}^{(0)} - \beta^{(0)}\|_2$ for linear utility, and $\|\widehat{g}^{(0)} - g^{(0)}\|_K$ for RKHS utility). For O2O$_{on}$ transfer, performance is indexed by the number of live source streams $K$, whereas for O2O$_{off}$ transfer it is indexed by the historical log size $n_{\mathcal{K}}$. Across both linear and RKHS utilities, gains grow monotonically with $K$ (or $n_{\mathcal{K}}$): a single auxiliary market already cuts regret by 15–20%, while ten live sources slash regret by more than half and deliver up to a $9\times$ jump-start in learning speed. These results confirm that CM-TDP translates theoretical advantages into substantial empirical improvements even when source-target differences are sparse and high-dimensional.

Table 2: Comparison with single-market learning baseline in sparse-difference market scenario averaging over different dimensions. In the metric column, $reg$, $std$ and $speed$ means cumulative regret, standard error, and learning speed, respectively. $K$ and $n_{\mathcal{K}}$ apply to O2O$_{on}$ and O2O$_{off}$ policies, respectively.

| Model | Metric | $K = 1$ | $K = 3$ | $K = 5$ | $K = 10$ | Avg |
|---|---|---|---|---|---|---|
| O2O$_{on}$-Linear | $reg \downarrow$ | 15% | 61% | 67% | 71% | 54% |
| | $std \downarrow$ | 9% | 36% | 38% | 39% | 31% |
| | $speed \uparrow$ | $1.2\times$ | $5.9\times$ | $8.1\times$ | $9.0\times$ | $6.0\times$ |
| O2O$_{on}$-RKHS | $reg$ | 17% | 62% | 68% | 73% | 55% |
| | $std$ | 11% | 33% | 36% | 36% | 29% |
| | $speed$ | $1.3\times$ | $6.7\times$ | $8.1\times$ | $8.9\times$ | $6.2\times$ |
| | | $n_{\mathcal{K}} = 50$ | $n_{\mathcal{K}} = 100$ | $n_{\mathcal{K}} = 200$ | $n_{\mathcal{K}} = 500$ | |
| O2O$_{off}$-Linear | $reg$ | 20% | 48% | 54% | 56% | 45% |
| | $std$ | 5% | 22% | 34% | 35% | 24% |
| | $speed$ | $1.5\times$ | $3.0\times$ | $4.9\times$ | $6.2\times$ | $3.9\times$ |
| O2O$_{off}$-RKHS | $reg$ | 15% | 47% | 53% | 56% | 43% |
| | $std$ | 7% | 21% | 35% | 38% | 25% |
| | $speed$ | $1.8\times$ | $3.1\times$ | $4.4\times$ | $5.9\times$ | $3.8\times$ |

### D.2.2 Running Time Evaluation

Table 3 reports the total running time (in seconds) of CM-TDP under sparse-difference markets with horizon $T = 2000$. For $O2O_{on}$, runtime increases moderately with the number of source markets $K$ and remains manageable even at $K = 10$. Linear models are consistently faster than RKHS models, but both scale sublinearly with the dimension $d$. For $O2O_{off}$, runtime is primarily affected by the size of the offline source dataset $n_{\mathcal{K}}$, showing only gradual increases as $n_{\mathcal{K}}$ grows from 50 to 500. Overall, these results confirm that both $O2O_{on}$ and $O2O_{off}$ policies are computationally efficient, with the added flexibility of RKHS utilities incurring only a modest overhead compared to linear models.

Table 3: Total Running time (in seconds) in sparse-difference market scenario across different dimensions in $T = 2000$ periods. $K$ and $n_{\mathcal{K}}$ apply to $O2O_{on}$ and $O2O_{off}$ policies, respectively.

| Model | $d$ | No transfer | $K = 1$ | $K = 3$ | $K = 5$ | $K = 10$ |
|---|---|---|---|---|---|---|
| $O2O_{on}$-Linear | 10 | 31 | 55 | 63 | 71 | 77 |
| | 15 | 42 | 80 | 85 | 93 | 101 |
| | 20 | 48 | 101 | 109 | 121 | 138 |
| | 100 | 386 | 794 | 814 | 866 | 972 |
| $O2O_{on}$-RKHS | 10 | 43 | 77 | 84 | 99 | 112 |
| | 15 | 57 | 98 | 106 | 121 | 140 |
| | 20 | 71 | 114 | 123 | 134 | 159 |
| | 100 | 438 | 832 | 927 | 1018 | 1156 |
| | | | $n_{\mathcal{K}} = 50$ | $n_{\mathcal{K}} = 100$ | $n_{\mathcal{K}} = 200$ | $n_{\mathcal{K}} = 500$ |
| $O2O_{off}$-Linear | 10 | 31 | 43 | 53 | 58 | 64 |
| | 15 | 42 | 58 | 64 | 70 | 77 |
| | 20 | 48 | 69 | 74 | 82 | 89 |
| | 100 | 386 | 563 | 597 | 642 | 663 |
| $O2O_{off}$-RKHS | 10 | 43 | 47 | 55 | 65 | 71 |
| | 15 | 57 | 66 | 76 | 87 | 98 |
| | 20 | 71 | 80 | 92 | 101 | 123 |
| | 100 | 438 | 602 | 694 | 758 | 872 |

# E  Proof of Theorem 5

Define $H_t = \{x_1^{(0)}, x_2^{(0)}, \ldots, x_t^{(0)}, \varepsilon_1^{(0)}, \varepsilon_2^{(0)}, \ldots, \varepsilon_t^{(0)}\}$ the history set up to time $t$. We also define $\overline{H}_t = H_t \cup \{x_{t+1}^{(0)}\}$ as the set obtained after augmenting a new feature $x_{t+1}^{(0)}$, we write

$$
\begin{aligned}
\mathbb{E}(\mathrm{reg}_t | H_{t-1}) &= \mathbb{E}(p_t^{*(0)} \mathbb{1}(v_t^{(0)} \geq p_t^*) | H_{t-1}) - \mathbb{E}(p_t^{(0)} \mathbb{1}(v_t^{(0)} \geq p_t^{(0)}) | H_{t-1}) \\
&= p_t^{*(0)}(1 - F(p_t^{*(0)} - x_t^{(0)} \cdot \boldsymbol{\beta}^{(0)})) - p_t^{(0)}(1 - F(p_t^{(0)} - x_t^{(0)} \cdot \boldsymbol{\beta}^{(0)})).
\end{aligned}
\tag{14}
$$

Note that $p_t^{*(0)} \in \arg\max \mathrm{rev}_t^{(0)}(p)$ and thus $r_t'(p_t^{*(0)}) = 0$. By Taylor expansion,

$$
\mathrm{rev}_t^{(0)}(p_t^{(0)}) = \mathrm{rev}_t(p_t^{*(0)}) + \frac{1}{2} r_t''(p)(p_t^{(0)} - p_t^{*(0)})^2,
\tag{15}
$$

for some $p$ between $p_t^{(0)}$ and $p_t^{*(0)}$.

**Lemma 14** (Upper bound for price). *The price given by policy $\pi$ is upper bounded by*

$$
\widehat{p}_t^{(0)} = h(x_t^{(0)} \cdot \widehat{\boldsymbol{\beta}}^{(0)}) \leq P.
$$

**Lemma 15** (1-Lipschitz property of $h$). *Suppose Assumption 1 holds and, in addition, $\phi'$ is bounded on $[-B_u, B_u]$:*

$$
0 < L_{\phi_0} \leq \phi'(u) \leq U_{\phi_0} < \infty.
$$

*Then*

$$
\sup_{|u| \leq B_u} |h'(u)| = \sup_{|u| \leq B_u} \left| 1 - \frac{1}{\phi'(z^*(u))} \right| \leq L_h < \infty,
$$

*hence*

$$
|h(u) - h(v)| \leq L_h |u - v| \quad \text{for all } |u|, |v| \leq B_u.
$$

*The price function $h$ satisfies $h'(u) < 1$, for all values of $u \in \mathbb{R}$.*

Therefore we obtain

$$
|r_t''(p)| = |2f(p - x_t^{(0)} \cdot \boldsymbol{\beta}^{(0)}) + p f'(p - x_t^{(0)} \cdot \boldsymbol{\beta}^{(0)})| \leq 2B + PB',
$$

with $B = \max_v f(v)$, and $B' = \max_v f'(v)$, where we use the fact that $p_t^{(0)}, p_t^{*(0)} \leq P$ and consequently $p \leq P$.

Then, combining Equations 14, 15, Lemmas 14, 15 gives

$$
\begin{aligned}
\mathbb{E}(\mathrm{reg}_t | H_{t-1}) &\leq (2B + PB')(p_t^{*(0)} - p_t^{(0)})^2 \leq C(p_t^{*(0)} - p_t^{(0)})^2 \\
&= C(h(\boldsymbol{\beta}^{(0)} \cdot x_t^{(0)}) - h(\widehat{\boldsymbol{\beta}}^{(0)} \cdot x_t^{(0)}))^2 \leq C|x_t^{(0)} \cdot (\boldsymbol{\beta}^{(0)} - \widehat{\boldsymbol{\beta}}^{(0)})|^2 \\
&\overset{(a)}{\leq} C\langle \widehat{\boldsymbol{\beta}}^{(0)} - \boldsymbol{\beta}^{(0)}, \Sigma(\widehat{\boldsymbol{\beta}}^{(0)} - \boldsymbol{\beta}^{(0)})\rangle,
\end{aligned}
$$

where $(a)$ results from that $x_t^{(0)}$ is independent of $H_{t-1}$, and $\Sigma = \mathbb{E}(x_t x_t^T)$.

Since the maximum eigenvalue of $\Sigma$ is bounded by $C_{\max}$, we obtain

$$
\mathbb{E}(\mathrm{reg}_t) = \mathbb{E}(\mathbb{E}(\mathrm{reg}_t | H_{t-1})) \leq C C_{\max} L_h^2 \mathbb{E}(\|\widehat{\boldsymbol{\beta}}^{(0)} - \boldsymbol{\beta}^{(0)}\|_2^2),
\tag{16}
$$

which brings the problem down to bounding the estimation error of the proposed estimator.

**Proposition 16.** *Consider linear utility model with Assumptions 1, 2, 3 and 4 holding true. Then, there exist positive constants $c_0, c_0', c_1, c_2$ such that, for $n_0 \geq c_0 s_0 \log d \vee c_0' \frac{d}{K}$, the following holds with probability at least $1 - 2/d - 2e^{-n_0/(c_0 s_0)}$:*

$$
\|\widehat{\boldsymbol{\beta}}^{(0)} - \boldsymbol{\beta}^{(0)}\|_2^2 \leq c_1 \frac{d \log d}{n_\mathcal{K}} + c_2 \frac{s_0 \log d}{n_0}.
\tag{17}
$$

**Corollary 17.** *Under conditions of Proposition 16, the following holds true:*

$$\mathbb{E}(\|\widehat{\boldsymbol{\beta}}^{(0)} - \boldsymbol{\beta}^{(0)}\|_2^2) \le c_1 \frac{d\log d}{n_{\mathcal{K}}} + c_2 \frac{s_0 \log d}{n_0} + 4W^2\left(\frac{2}{d} + 2e^{\frac{-n_0}{c_0 s_0}}\right).$$

**Proposition 18.** *Consider linear utility model with Assumptions 1, 2, 3 and 4 holding true. There exist constants $c_3, c_4, c_5 > 0$, such that for $n_0 \ge c_3 d$, the following holds true:*

$$\mathbb{E}(\|\widehat{\boldsymbol{\beta}}^{(0)} - \boldsymbol{\beta}^{(0)}\|_2^2) \le c_4 \frac{d\log d}{n_{\mathcal{K}}} + c_5 \frac{(s_0 + 1)\log d}{n_0} + 4W^2 e^{-c_3 n_0^2}.$$

Now, since the length of episodes grows exponentially, the number of episodes by period $T$ is logarithmic in $T$. Specifically, $T$ belongs to episode $M = \lceil \log T \rceil$. Hence,

$$\text{Regret}(T; \pi) = \sum_{m=1}^{M} \text{Reg}(m\text{th Episode}).$$

We bound the total regret over each episode by considering three separate cases:

1. $2^{m-2} \le c_0 s_0 \log d \vee c_0' d$: Here, $c_0, c_0'$ are the constants in Proposition 16. In this case, episodes are not large enough to estimate the parameters accurately enough, and thus we use a naive bound. Clearly, by Lemma 14, we have $\mathbb{E}(\text{reg}_t) \le p_t^* \le P$. Hence the total regret over such episodes is at most $4Pc_0 s_0 \log d \vee 4Pc_0' d$.

2. $c_0 s_0 \log d \le 2^{m-2} < c_3 d$: Applying Corollary 17 to Equation 16,

$$
\begin{aligned}
\text{Reg}(m\text{th Episode}) &= \sum_{t=l_m}^{l_{m+1}-1} \mathbb{E}(\text{reg}_t) \le CC_{\max} \sum_{t=l_m}^{l_{m+1}-1} \mathbb{E}(\|\widehat{\boldsymbol{\beta}}_m^{(0)} - \boldsymbol{\beta}^{(0)}\|_2^2) \\
&\le CC_{\max}\left\{ c_1 l_m \frac{d\log d}{Kl_{m-1}} + c_2 l_m \frac{s_0 \log d}{l_{m-1}} + 8W^2\left(\frac{l_m}{d} + l_m e^{\frac{-l_{m-1}}{c_0 s_0}}\right)\right\} \\
&\le 2CC_{\max}\left\{ \frac{c_1}{K} d\log d + c_2 s_0 \log d + 8W^2\left(c_3 + l_{m-1} e^{\frac{-l_{m-1}}{c_0 s_0}}\right)\right\},
\end{aligned}
$$

where in the last step we used $l_m = 2l_{m-1}$ and $l_{m-1} \le c_3 d$. Therefore, in this case

$$\text{Reg}(m\text{th Episode}) \le C_1' \frac{d}{K}\log d + C_2' s_0 \log d,$$

where $C_1', C_2'$ hides various constants in the right-hand side of the above equation.

3. $c_3 d < 2^{m-2}$: Applying Proposition 18 to Equation 16,

$$
\begin{aligned}
\text{Reg}(m\text{th Episode}) &= \sum_{t=l_m}^{l_{m+1}-1} \mathbb{E}(\text{reg}_t) \le CC_{\max} \sum_{t=l_m}^{l_{m+1}-1} \mathbb{E}(\|\widehat{\boldsymbol{\beta}}_m^{(0)} - \boldsymbol{\beta}^{(0)}\|_2^2) \\
&\le CC_{\max}\left\{ c_4 l_m \frac{d\log d}{Kl_{m-1}} + c_5 l_m \frac{(s_0+1)\log d}{l_{m-1}} + 8W^2 l_{m-1} e^{\frac{-l_{m-1}}{c_3 d}}\right\} \\
&\le CC_{\max}\left\{ c_4 \frac{d}{K}\log d + c_5(s_0+1)\log d + 8W^2 l_{m-1} e^{\frac{-l_{m-1}}{c_3 d}}\right\}.
\end{aligned}
$$

Therefore, in this case

$$\text{Reg}(m\text{th Episode}) \le C_1'' \frac{d}{K}\log d + C_2'' s_0 \log d.$$

Combining the above three cases, we get

$$\text{Regret}(T; \pi) \le C_1 d\log d \cdot \log T + C_2 s_0 \log d \cdot \log n_{\mathcal{K}} = \mathcal{O}\left(\frac{d}{K}\log d \log T + s_0 \log d \log T\right),$$

which concludes the proof.

# F  Proof of Theorem 12

In offline-to-online transfer setting, we obtain the same regret inequality as in Theorem 5:

$$\mathbb{E}(\text{reg}_t) = \mathbb{E}(\mathbb{E}(\text{reg}_t|H_{t-1})) \leq CC_{\max}\mathbb{E}(\|\widehat{\boldsymbol{\beta}}^{(0)} - \boldsymbol{\beta}^{(0)}\|_2^2). \tag{18}$$

Similar to Propositions 16 and 18, we state the following two propositions on the estimation error regarding the transfer-learning phase.

**Proposition 19.** *Consider linear utility model with Assumptions 1, 2, 3 and 4 holding true. Then, there exist positive constants $c_0, c_1, c_2$ such that, for $n_0 \geq c_0 s_0 \log d$, the following holds with probability at least $1 - 2/d - 2e^{-n_0/(c_0 s_0)}$:*

$$\|\widehat{\boldsymbol{\beta}}^{(0)} - \boldsymbol{\beta}^{(0)}\|_2^2 \leq c_1 \frac{d \log d}{n_{\mathcal{K}}} + c_2 \frac{s_0 \log d}{n_0}.$$

**Corollary 20.** *Under conditions of Proposition 19, the following holds true:*

$$\mathbb{E}(\|\widehat{\boldsymbol{\beta}}^{(0)} - \boldsymbol{\beta}^{(0)}\|_2^2) \leq c_1 \frac{d \log d}{n_{\mathcal{K}}} + c_2 \frac{s_0 \log d}{n_0} + 4W^2 \left( \frac{2}{d} + 2e^{\frac{-n_0}{c_0 s_0}} \right).$$

The following proposition gives a tighter bound for the estimation error as $n_0$ gets larger.

**Proposition 21.** *Consider linear utility model with Assumptions 1, 2, 3 and 4 holding true. There exist constants $c_3, c_4, c_5 > 0$, such that for $n_0 \geq c_3 d$, the following holds true:*

$$\mathbb{E}(\|\widehat{\boldsymbol{\beta}}^{(0)} - \boldsymbol{\beta}^{(0)}\|_2^2) \leq c_4 \frac{d \log d}{n_{\mathcal{K}}} + c_5 \frac{(s_0 + 1) \log d}{n_0} + 4W^2 e^{-c_3 n_0^2}.$$

The next proposition states the estimation error in the without-transfer-learning phase of Algorithm 4.

**Proposition 22.** *Consider linear utility model with Assumptions 1, 2, 3 and 4 holding true. Then, there exist a positive constant $c_7$ such that, for $n_0 \geq \widetilde{c} n_{\mathcal{K}}$, the following holds with probability at least $1 - 1/d$:*

$$\|\widehat{\boldsymbol{\beta}}^{(0)} - \boldsymbol{\beta}^{(0)}\|_2^2 \leq c_7 \frac{d \log d}{n_0},$$

*where $\widetilde{c} \approx (c_7 d - c_2 s_0)/c_1 d$.*

The adaptation parameter $\widetilde{c}$ is dynamically determined through a comparative analysis of the estimation error bounds in Propositions 21 (transfer-enabled) and 22 (target-only), where we strategically disable transfer learning updates when the volume of target data is large enough to ensure statistically optimal estimation performance.

**Corollary 23.** *Under assumptions of Proposition 22, the following holds true:*

$$\mathbb{E}(\|\widehat{\boldsymbol{\beta}}^{(0)} - \boldsymbol{\beta}^{(0)}\|_2^2) \leq c_4 \frac{d \log d}{n_0} + \frac{4W^2}{d}.$$

We bound the total regret over each episode by considering four separate cases:

1. $2^{m-2} \leq c_0 s_0 \log d$: Here, $c_0$ is the constant in the statement of Proposition 19. In this case, episodes are not large enough to estimate the parameters accurately enough, and thus we use a naive bound. Again, by Lemma 14, we have $\mathbb{E}(\text{reg}_t) \leq p_t^* \leq P$. Hence the total regret over such episodes is at most $4Pc_0 s_0 \log d$.

2. $c_0 s_0 \log d \leq 2^{m-2} \leq c_3 d$: Here, $c_0$ is the constant in the statement of Proposition 21. Applying Corollary 20 to Equation 18 in episode $m$,

$$\text{Reg}(m\text{th Episode}) = \sum_{t=l_m}^{l_{m+1}-1} \mathbb{E}(\text{reg}_t) \leq CC_{\max} \sum_{t=l_m}^{l_{m+1}-1} \mathbb{E}(\|\widehat{\boldsymbol{\beta}}_m^{(0)} - \boldsymbol{\beta}^{(0)}\|_2^2)$$

$$\leq CC_{\max} \left\{ c_1 d \log d \frac{l_m}{n_{\mathcal{K}}} + 2c_2 s_0 \log d + 16W^2 \left( 2c_3 + l_{m-1} e^{\frac{-l_{m-1}}{c_0 s_0}} \right) \right\},$$

where in the last step we used $l_m = 2l_{m-1}$ and $l_m = 2^{m-1} \leq 2c_3 d$.

3. $c_3 d \log d \leq 2^{m-2} \leq \widetilde{c} n_\mathcal{K}$: Applying Corollary 20 to Equation 18,

$$\text{Reg}(m\text{th Episode}) = \sum_{t=l_m}^{l_{m+1}-1} \mathbb{E}(\text{reg}_t) \leq CC_{\max} \sum_{t=l_m}^{l_{m+1}-1} \mathbb{E}(\|\widehat{\boldsymbol{\beta}}_m^{(0)} - \boldsymbol{\beta}^{(0)}\|_2^2)$$

$$\leq CC_{\max} \left\{ c_1 d \log d \frac{l_m}{n_\mathcal{K}} + 2c_2 s_0 \log d + 16W^2 \left( \frac{\widetilde{c} n_\mathcal{K}}{d} + l_{m-1} e^{\frac{-l_{m-1}}{c_0 s_0}} \right) \right\},$$

Combing case 2 and 3, sum over $c_1 d \log d \frac{l_m}{n_\mathcal{K}}$ over $l_m \in [2c_0 s_0 \log d, 2\widetilde{c} n_\mathcal{K}]$ and ignore the constant term, we obtain

$$\text{Regret}(c_0 s_0 \log d \to n_\mathcal{K}; \pi) \leq c_1 d \log d \frac{1}{n_\mathcal{K}} \sum_m 2^{m-1} + C_2' s_0 \log d \log n_\mathcal{K}$$

$$= c_1 d \log d \frac{\widetilde{c} n_\mathcal{K} - c_0 s_0 \log d}{n_\mathcal{K}} + C_2' s_0 \log d \log n_\mathcal{K}$$

$$\approx C_2' s_0 \log d \log n_\mathcal{K}$$

4. $2^{m-2} > \widetilde{c} n_\mathcal{K}$: applying Proposition 22 to episode $k$, we obtain

$$\text{Reg}(m\text{th Episode}) = \sum_{t=l_m}^{l_{m+1}-1} \mathbb{E}(\text{reg}_t) \leq CC_{\max} \sum_{t=l_m}^{l_{m+1}-1} \mathbb{E}(\|\widehat{\boldsymbol{\beta}}_m - \boldsymbol{\beta}^{(0)}\|_2^2)$$

$$\leq CC_{\max} \left\{ c_4 \frac{d \log d}{l_{m-1}} l_m + 8W^2 \left( \frac{l_{m-1}}{d} + 2l_{m-1} e^{\frac{-l_{m-1}}{c_3 d}} \right) \right\}$$

$$\leq CC_{\max} \left\{ 2c_4 d \log d + 8W^2 \left( \frac{\widetilde{c} n_\mathcal{K}}{d} + 2l_{m-1} e^{\frac{-l_{m-1}}{c_3 d}} \right) \right\},$$

and thus

$$\text{Reg}(m\text{th Episode}) \leq C_3' d \log d.$$

Combining the above four cases, we get

$$\text{Regret}(T; \pi) \leq C_1 d \log d \cdot (\log T - \log n_\mathcal{K}) + C_2 s_0 \log d \cdot \log n_\mathcal{K}$$
$$= \mathcal{O}\left( d \log d \cdot \log T + (s_0 - d) \log d \cdot \log n_\mathcal{K} \right).$$

which concludes the proof.

## G  Proof of Theorem 6

Define the instance class $\mathcal{I}(F, \Sigma, W, s_0, K)$ as above and, w.l.o.g., restrict prices to $[0, \bar{p}(F, W)]$ where $\bar{p}(F, W) := \sup_{|u| \leq W} h(u)$. Let

$$z^*(u) := \phi^{-1}(-u), \qquad B_z := \sup_{|u| \leq W} |z^*(u)|,$$

$$\Phi_{\min} := \inf_{|u| \leq W} \phi'(z^*(u)), \qquad f_{\min} := \inf_{|z| \leq B_z} f(z), \qquad f_{\max} := \sup_{z \in [-W, \bar{p}(F,W)+W]} f(z),$$

$$\epsilon := \inf_{z \in [-W, \bar{p}(F,W)+W]} \min\{F(z), 1 - F(z)\},$$

$$\kappa := f_{\min} \Phi_{\min}, \qquad m_h := \inf_{|u| \leq W} \left( 1 - \frac{1}{\phi'(z^*(u))} \right).$$

For Logistic and Gaussian $F$ these constants are explicit; see Corollary 24 below.

We proceed in three steps: (i) *curvature* of the revenue around the oracle price; (ii) a *single-sample KL* upper bound that is uniform over the allowed price interval; (iii) a Fano *packing* argument for two hard sub-families whose risks add up.

**(i) Revenue curvature and price sensitivity.** Fix $u = x^\top \beta$. Let $r(p; u) := p\,[1 - F(p - u)]$. The first-order condition gives the oracle price $p^*(u) = h(u) = u + z^*(u)$, where $z^*(u)$ solves $1 - F(z^*) = p^* f(z^*)$, equivalently $\phi(z^*) = -u$. A direct calculation shows

$$\frac{\partial^2}{\partial p^2} r(p^*(u); u) \;=\; -f(z^*(u))\,\phi'(z^*(u)) \;=\; -\kappa(u), \qquad \kappa(u) := f(z^*(u))\phi'(z^*(u)). \quad (19)$$

Hence $r(\cdot; u)$ is *uniformly strongly concave* around $p^*(u)$ with curvature at least $\kappa := \inf_{|u| \leq W} \kappa(u) = f_{\min}\Phi_{\min} > 0$.[1]

Next, differentiate $h(u) = u + \phi^{-1}(-u)$ to get

$$h'(u) \;=\; 1 - \frac{1}{\phi'(z^*(u))} \qquad \Rightarrow \qquad m_h := \inf_{|u| \leq W} h'(u) = 1 - \frac{1}{\Phi_{\min}} > 0. \quad (20)$$

By strong concavity, for any $\Delta u$ small enough (chosen below) and $\Delta p := h(u + \Delta u) - h(u)$,

$$r\big(h(u); u\big) - r\big(h(u + \Delta u); u\big) \;\geq\; \frac{\kappa}{2}\,\Delta p^2 \;\geq\; \frac{\kappa\, m_h^2}{2}\,(\Delta u)^2. \quad (21)$$

We ensure the uniform validity of (21) by choosing the pack radii (below) so that $|\Delta p|$ stays within a fixed neighborhood where (19) and the lower bounds defining $\kappa$ and $m_h$ apply; this is straightforward since $h'$ is bounded on $[-W, W]$.

**(ii) A uniform single-sample KL bound.** Fix any market $k$, round $t$, context $x$, and price $p \in [0, \bar{p}(F, W)]$. Under parameter $\beta$, the success probability is

$$q(\beta) \;=\; 1 - F\big(p - x^\top \beta\big).$$

For any $\beta, \beta'$, Taylor's theorem for the Bernoulli KL yields

$$\mathrm{KL}\big(\mathrm{Bern}\big(q(\beta)\big) \,\big\|\, \mathrm{Bern}\big(q(\beta')\big)\big) \;\leq\; \frac{\big(q(\beta) - q(\beta')\big)^2}{2\,\epsilon\,(1 - \epsilon)}, \quad (22)$$

because both $q(\beta)$ and $q(\beta')$ lie in $[\epsilon, 1 - \epsilon]$ by the price truncation and the bounded ranges of $p$ and $u$.[2] Using the mean-value bound $|F(z) - F(z')| \leq f_{\max}|z - z'|$ on the same interval gives

$$\mathrm{KL}\big(\mathrm{Bern}\big(q(\beta)\big) \,\big\|\, \mathrm{Bern}\big(q(\beta')\big)\big) \leq \frac{f_{\max}^2}{2\,\epsilon(1 - \epsilon)}\,\big(x^\top(\beta - \beta')\big)^2. \quad (23)$$

Taking expectation over $x \sim P_x$ and using $\mathbb{E}[(x^\top \Delta)^2] \leq C_{\max}\|\Delta\|_2^2$ yields

$$\mathbb{E}\big[\mathrm{KL}(\cdot\|\cdot)\big] \;\leq\; C_{\mathrm{KL}}(F, \Sigma, W)\,\|\beta - \beta'\|_2^2, \qquad C_{\mathrm{KL}}(F, \Sigma, W) := \frac{C_{\max}\,f_{\max}^2}{2\,\epsilon(1 - \epsilon)}. \quad (24)$$

Summing over markets gives a factor $K$ when all markets differ (family A below), and a factor $1$ when only the target differs (family B).

**(iii) Packing and per-round error.** We use two hard sub-families and calibrate their radii so that the cumulative KL up to time $t - 1$ is a small fraction of the packing entropy. All estimates below hold conditionally on the realized (possibly adaptive) prices because (24) is uniform in $p \in [0, \bar{p}]$.

*(A) Aggregation.* Let $V \subset \{-1, +1\}^d$ be a Varshamov–Gilbert set with $|V| \geq 2^{d/8}$ and Hamming distance at least $d/8$. For $v \in V$ define the instance by $\beta^{(k)} = \theta_v := \mu\, v$ for all $k \in \{0\} \cup [K]$. Then $\|\theta_v\|_1 = \mu d \leq W$ provided $\mu \leq W/d$. For any distinct $v, v'$,

$$\frac{d}{2}\,\mu^2 \;\leq\; \|\theta_v - \theta_{v'}\|_2^2 \;\leq\; 4d\,\mu^2.$$

Up to time $t - 1$, the pathwise KL between the two induced joint laws (over all $K$ markets) is bounded by

$$\mathrm{KL}_{1:t-1} \;\leq\; K\,(t - 1)\,C_{\mathrm{KL}} \cdot 4d\,\mu^2.$$

---

[1] Since $\phi' \geq L_{\phi_0} > 0$ on $[-B_u, B_u]$ and $|z^*(u)| \leq B_z$ for $|u| \leq W$, $\Phi_{\min} \geq L_{\phi_0}$. Moreover, $f_{\min} > 0$ on $[-B_z, B_z]$ for Logistic/Gaussian $F$.

[2] On $z = p - x^\top \beta \in [-W, \bar{p}(F, W) + W]$ we have $F(z) \in [\epsilon, 1 - \epsilon]$, so $q = 1 - F(z) \in [\epsilon, 1 - \epsilon]$.

Choose
$$\mu^2 \;=\; \frac{\log |V|}{64\,K\,(t-1)\,C_{\mathrm{KL}}\,d} \;\leq\; \frac{\log 2}{512\,K\,(t-1)\,C_{\mathrm{KL}}},$$
and also $\mu \leq W/d$ (which is automatic for all $t$ large enough; for small $t$ it only improves the bound). Then Fano's inequality gives a constant probability (say $\geq 1/2$) of misidentifying the pack element, hence
$$\mathbb{E}\big[\|\widehat{\theta}_{t-1} - \theta\|_2^2\big] \;\geq\; \frac{1}{8}\,\mu^2 d.$$
By $\mathbb{E}[(x^\top e)^2] \geq C_{\min}\,\mathbb{E}\|e\|_2^2$, (i), and (21), the *target-market* instant regret at time $t$ is
$$\mathbb{E}[\mathrm{reg}_t] \;\geq\; \frac{\kappa\,m_h^2}{2}\,\mathbb{E}\big[(x_t^{(0)\top}(\widehat{\theta}_{t-1}-\theta))^2\big] \;\geq\; \frac{\kappa\,m_h^2}{2}\,C_{\min}\cdot\frac{\mu^2 d}{8} \;\geq\; \frac{C_{\min}\,\kappa\,m_h^2}{128\,C_{\mathrm{KL}}}\cdot\frac{d}{K\,t}.$$
Summing $t = 2,\ldots,T$ yields
$$\sum_{t=1}^{T}\mathbb{E}[\mathrm{reg}_t] \;\geq\; \frac{C_{\min}\,\kappa\,m_h^2}{128\,C_{\mathrm{KL}}}\cdot\frac{d}{K}\,\log T.$$

*(B) Debiasing.* Let $S \subset [d]$ with $|S| = s_0$ and let $\mathcal{W} \subset \{-1,+1,0\}^d$ be a Varshamov–Gilbert family of $s_0$-sparse sign vectors with pairwise Hamming distance at least $s_0/8$ and cardinality $|\mathcal{W}| \geq \binom{d}{s_0}^{1/8} 2^{s_0/8}$ so that $\log |\mathcal{W}| \geq \frac{s_0}{8}\log\frac{ed}{s_0}$. Let $\theta_{\mathrm{c}} \in \mathbb{R}^d$ be any fixed vector with $\|\theta_{\mathrm{c}}\|_1 \leq W/2$ and define for each $w \in \mathcal{W}$:
$$\beta^{(k)} = \theta_{\mathrm{c}} \quad (k \in [K]), \qquad \beta^{(0)} = \theta_{\mathrm{c}} + \Delta_w, \quad \Delta_w := \mu'\,w,$$
with $\|\theta_{\mathrm{c}}\|_1 + \|\Delta_w\|_1 \leq W$ ensured by taking $\mu' \leq W/(2s_0)$. Then for $w \neq w'$,
$$\frac{s_0}{2}\,\mu'^2 \;\leq\; \|\Delta_w - \Delta_{w'}\|_2^2 \;\leq\; 4s_0\,\mu'^2.$$
Only the *target* market carries information about $w$, hence up to time $t-1$
$$\mathrm{KL}_{1:t-1} \;\leq\; (t-1)\,C_{\mathrm{KL}}\cdot 4s_0\,\mu'^2.$$
Choose
$$\mu'^2 \;=\; \frac{\log |\mathcal{W}|}{64\,(t-1)\,C_{\mathrm{KL}}\,s_0} \;\leq\; \frac{\log\frac{ed}{s_0}}{512\,(t-1)\,C_{\mathrm{KL}}}.$$
Fano again yields $\mathbb{E}\|\widehat{\Delta}_{t-1} - \Delta\|_2^2 \geq \frac{1}{8}\,\mu'^2 s_0$, hence the target instant regret obeys
$$\mathbb{E}[\mathrm{reg}_t] \;\geq\; \frac{C_{\min}\,\kappa\,m_h^2}{128\,C_{\mathrm{KL}}}\cdot\frac{s_0\log\frac{ed}{s_0}}{t}.$$
Summing $t$ gives the second term in (8).

Combining (A) and (B) gives the stated result.

**Corollary 24** (Explicit constants for Logistic and Gaussian). *Let* $B_z = \sup_{|u|\leq W}|z^*(u)|$ *with* $z^*(u) = \phi^{-1}(-u)$ *and* $\bar{p}(F,W) = \sup_{|u|\leq W} h(u)$.
***Logistic noise***
$F(z) = \frac{1}{1+e^{-z}}$, $f(z) = F(z)(1-F(z))$. *We have*
$$\phi'(z) = \frac{1}{F(z)}, \quad h'(u) = 1 - \frac{1}{\phi'(z^*(u))} = 1 - F(z^*(u)), \quad \kappa(u) = f(z^*(u))\phi'(z^*(u)) = 1 - F(z^*(u)).$$
*Hence*
$$\Phi_{\min} = \frac{1}{F(B_z)}, \qquad m_h = \inf_{|u|\leq W} h'(u) = 1 - F(B_z), \qquad \kappa = \inf_{|u|\leq W}\kappa(u) = 1 - F(B_z).$$
*Moreover,* $f_{\max} = \frac{1}{4}$ *and*
$$\epsilon = \min_{z\in[-W,\,\bar{p}(F,W)+W]}\min\{F(z), 1-F(z)\} = 1 - F\big(W + \bar{p}(F,W)\big) \;\geq\; 1 - F\big(2W + B_z\big).$$

*Therefore,*

$$C_{\mathrm{KL}}(F, \Sigma, W) = \frac{C_{\max}}{32\,\epsilon(1-\epsilon)}, \qquad \frac{C_{\min}\,\kappa\,m_h^2}{C_{\mathrm{KL}}} \;\geq\; \frac{32\,C_{\min}}{C_{\max}}\,\epsilon(1-\epsilon)\,\big(1-F(B_z)\big)^3.$$

### Gaussian noise
$F(z) = \Phi(z)$, $f(z) = \varphi(z) = \frac{1}{\sqrt{2\pi}}e^{-z^2/2}$. *Let $R(z) := \frac{1-\Phi(z)}{\varphi(z)}$ be Mills' ratio. Then*

$$\phi'(z) = 2 - z\,R(z), \qquad \Phi_{\min} \;=\; \inf_{|z|\leq B_z}\big(2 - z\,R(z)\big) \;\geq\; 2 - B_z\,R(B_z).$$

*Using the inequality $R(z) \leq \frac{1}{z+\frac{1}{z}}$ for $z > 0$ gives $B_z R(B_z) \leq \frac{B_z^2}{B_z^2+1}$, hence*

$$\Phi_{\min} \;\geq\; 1 + \frac{1}{B_z^2+1}, \qquad m_h \;=\; 1 - \frac{1}{\Phi_{\min}} \;\geq\; \frac{1}{B_z^2+2}.$$

*Also $f_{\min} = \varphi(B_z) = \frac{1}{\sqrt{2\pi}}e^{-B_z^2/2}$, $f_{\max} = \varphi(0) = \frac{1}{\sqrt{2\pi}}$, and*

$$\epsilon \;=\; \min_{z\in[-W,\,\bar{p}(F,W)+W]} \min\{\Phi(z), 1-\Phi(z)\} \;=\; 1 - \Phi\big(W+\bar{p}(F,W)\big) \;\geq\; 1 - \Phi\big(2W + B_z\big).$$

*Therefore*

$$\kappa = f_{\min}\Phi_{\min} \;\geq\; \frac{1}{\sqrt{2\pi}}e^{-B_z^2/2}\Big(1+\frac{1}{B_z^2+1}\Big), \qquad C_{\mathrm{KL}}(F, \Sigma, W) = \frac{C_{\max}}{4\pi\,\epsilon(1-\epsilon)},$$

*and*

$$\frac{C_{\min}\,\kappa\,m_h^2}{C_{\mathrm{KL}}} \;\geq\; \frac{2\pi\,C_{\min}}{C_{\max}}\,\epsilon(1-\epsilon)\,\frac{1}{\sqrt{2\pi}}e^{-B_z^2/2}\Big(1+\frac{1}{B_z^2+1}\Big)\cdot\frac{1}{(B_z^2+2)^2}.$$

*In both cases $B_z$ and $\bar{p}(F,W)$ are finite since $\phi$ is strictly increasing and $|u| \leq W$; they are explicit functions of $(F,W)$ via $z^*(u) = \phi^{-1}(-u)$ and $h(u) = u + z^*(u)$.*

# H   Proof of Theorem 10

Following the same analysis in Theorem 12, we have

$$\begin{aligned}
\mathbb{E}(\mathrm{reg}_t \mid H_{t-1}) &= \mathrm{rev}_t^{(0)}(p_t^{*(0)}) - \mathrm{rev}_t^{(0)}(p_t^{(0)}) + \tfrac{1}{2}r_t''(p)\big(p_t^{(0)} - p_t^{*(0)}\big)^2 \\
&\leq C\big(p_t^{*(0)} - p_t^{(0)}\big)^2 \;=\; C\big(h(g^{(0)}(x_t^{(0)})) - h(\widehat{g}^{(m)}(x_t^{(0)}))\big)^2 \\
&\leq C\,L_h^2\big(g^{(0)}(x_t^{(0)}) - \widehat{g}^{(m)}(x_t^{(0)})\big)^2,
\end{aligned}$$

where $C$ absorbs the bound on $|r_t''(p)|$ over $[0, P]$ and $L_h$ is the Lipschitz constant of $h$ on the working interval (by Lemma 15 and the price truncation).

Using Lemma 15 gives

$$\begin{aligned}
\mathbb{E}(\mathrm{reg}_t) = \mathbb{E}\big(\mathbb{E}(\mathrm{reg}_t \mid H_{t-1})\big) &\leq C\,\mathbb{E}\Big[(g^{(0)}(x_t^{(0)}) - \widehat{g}^{(m)}(x_t^{(0)}))^2\Big] \\
&= C\,\mathbb{E}\big\|g^{(0)} - \widehat{g}^{(m)}\big\|_{L_2(P_x)}^2 \;\leq\; C\,\kappa^2\,\mathbb{E}\big\|g^{(0)} - \widehat{g}^{(m)}\big\|_{\mathcal{H}_k}^2,
\end{aligned} \tag{25}$$

where $\kappa$ is the kernel bound in Assumption 7.

**Proposition 25** (Aggregation Error). *Under Assumptions 2 and 7, choosing $\lambda_{ag} \asymp n_{\mathcal{K}}^{-\frac{2\alpha}{2\alpha\beta+1}}$, the aggregation estimation error of Algorithm 3 satisfies*

$$\mathbb{E}\Big(\|g^{(ag)} - \widehat{g}^{(ag)}\|_{\mathcal{H}_k}^2\Big) \;\leq\; C_1\big(R^2 + \sigma^2\big)\,n_{\mathcal{K}}^{-\frac{2\alpha\beta}{2\alpha\beta+1}}, \tag{26}$$

*where $R$ is the constant in Assumption 7 and $\sigma$ is the standard deviation of the market noise in (1).*

*Proof.* Let $\Sigma := \mathbb{E}_{\mathcal{K}}[K(x, \cdot) \otimes K(x, \cdot)]$ denote the kernel integral operator and $N(\lambda) := \mathrm{Tr}\big(\Sigma(\Sigma + \lambda I)^{-1}\big)$ the effective dimension. Consider the regularized empirical risk

$$\widehat{\mathcal{L}}_\lambda(g) = \frac{1}{n_{\mathcal{K}}} \sum_{i=1}^{n_{\mathcal{K}}} \ell\big(g; p_i, x_i, y_i\big) + \lambda \|g\|_{\mathcal{H}_k}^2,$$

with the Bernoulli log-loss $\ell(g; p, x, y) := -\big[1(y = 1) \log(1 - F(p - g(x))) + 1(y = 0) \log(F(p - g(x)))\big]$ used throughout Algorithm 3. Let

$$g_\lambda^{(ag)} = \arg\min_{g \in \mathcal{H}_k} \mathbb{E}\widehat{\mathcal{L}}_\lambda(g)$$

be the population minimizer. A standard RERM decomposition for smooth, strongly convex losses yields

$$\mathbb{E}\big\|\widehat{g}^{(ag)} - g_\lambda^{(ag)}\big\|_{L_2(P_x)}^2 \lesssim \frac{N(\lambda)}{n_{\mathcal{K}}} \implies \mathbb{E}\big\|\widehat{g}^{(ag)} - g_\lambda^{(ag)}\big\|_{\mathcal{H}_k}^2 \lesssim \frac{N(\lambda)}{n_{\mathcal{K}}},$$

using $\|f\|_{L_2(P_x)}^2 \leq \kappa^2 \|f\|_{\mathcal{H}_k}^2$. For the approximation error, under the source condition $g^{(ag)} \in \mathrm{Range}(\Sigma^\beta)$ (Assumption 9(ii)) we have

$$\|g_\lambda^{(ag)} - g^{(ag)}\|_{\mathcal{H}_k}^2 \lesssim \lambda^{2\beta} R^2.$$

Assumption 9(i) gives $N(\lambda) \asymp \lambda^{-1/(2\alpha)}$. Balancing $N(\lambda)/n_{\mathcal{K}}$ and $\lambda^{2\beta}$ gives $\lambda_{ag} \asymp n_{\mathcal{K}}^{-2\alpha/(2\alpha\beta+1)}$ and the stated rate. □

**Proposition 26** (Bias Correction Error). *Under Assumptions 2, 7 and 8, choosing $\lambda_{tf} \asymp (n_0 H^2)^{-2\alpha/(2\alpha+1)}$, the debiasing error of Algorithm 3 satisfies*

$$\mathbb{E}\Big(\big\|\delta^{(0)} - \widehat{\delta}\big\|_{L_2(P_x)}^2\Big) \leq C_2 \, n_0^{-2\alpha/(2\alpha+1)} \, H^{2/(2\alpha+1)}, \tag{27}$$

*where $H$ is the task-similarity parameter in Assumption 8.*

*Proof.* According to Chai et al. [7], the regularized estimator can be written as

$$\widehat{\delta}^{(0)} = \big(\widehat{\Sigma}^{(0)} + \lambda_{tf} I\big)^{-1}\big(\widehat{g}^{(0)} + \lambda_{tf}\widehat{\delta}^{(k)}\big).$$

Hence

$$\widehat{\delta}^{(0)} - \delta^{(0)} = \underbrace{\big(\widehat{\Sigma}^{(0)} + \lambda_{tf} I\big)^{-1}\big(\widehat{g}^{(0)} - \widehat{\Sigma}^{(0)}\delta^{(0)}\big)}_{\text{Variance}} + \underbrace{\lambda_{tf}\big(\widehat{\Sigma}^{(0)} + \lambda_{tf} I\big)^{-1}\big(\widehat{\delta}^{(k)} - \delta^{(0)}\big)}_{\text{Bias}}.$$

For the variance term, writing $A_\lambda := \Sigma^{1/2}(\widehat{\Sigma}^{(0)} + \lambda I)^{-1}$ implies

$$\mathbb{E}\big\|\text{Variance}\big\|_{L_2(P_x)}^2 \lesssim \frac{N_0(\lambda_{tf})}{n_0}, \quad \text{where} \quad N_0(\lambda) \asymp \lambda^{-1/(2\alpha)}.$$

For the bias term, by spectral calculus,

$$\big\|\lambda(\widehat{\Sigma}^{(0)} + \lambda I)^{-1}u\big\|_{L_2(P_x)}^2 = \sum_j \frac{\lambda^2 \mu_j}{(\mu_j + \lambda)^2} \langle u, \phi_j\rangle^2 \leq \frac{\lambda}{4} \|u\|_{\mathcal{H}_k}^2,$$

so Assumption 8 gives $\mathbb{E}\big\|\text{Bias}\big\|_{L_2(P_x)}^2 \lesssim \lambda_{tf} H^2$. Balancing $N_0(\lambda)/n_0$ with $\lambda H^2$ yields $\lambda_{tf} \asymp (n_0 H^2)^{-2\alpha/(2\alpha+1)}$ and the stated bound. □

We now bound the total regret over each episode. Using Equation (25), we obtain

$$\mathrm{Reg}(m\text{th Episode}) = \sum_{t=l_m}^{l_{m+1}-1} \mathbb{E}(\mathrm{reg}_t) \leq C \sum_{t=l_m}^{l_{m+1}-1} \mathbb{E}\big(\|\widehat{g}_m - g^{(0)}\|_{L_2(P_x)}^2\big)$$

$$\leq C \, 2^{m-1}\Big\{(K \, 2^{m-1})^{-\frac{2\alpha\beta}{2\alpha\beta+1}} + (2^{m-1})^{-\frac{2\alpha}{2\alpha+1}} H^{\frac{2}{2\alpha+1}}\Big\}.$$

Summing over $M = \lceil \log T \rceil$ episodes gives

$$\text{Regret}(T; \pi) \ \leq \ C \sum_{m=1}^{M} 2^{m-1} \left\{ (K\, 2^{m-2})^{-\gamma_1} + (2^{m-2})^{-\gamma_2} \, H^{\frac{2}{2\alpha+1}} \right\},$$

where $\gamma_1 = \frac{2\alpha\beta}{2\alpha\beta+1}$ and $\gamma_2 = \frac{2\alpha}{2\alpha+1}$. Let

$$\text{Term 1} = K^{-\gamma_1} \sum_{m=1}^{M} 2^{m-1} (2^{m-2})^{-\gamma_1} = K^{-\gamma_1} 2^{2\gamma_1 - 1} \sum_{m=1}^{M} \left( 2^{1-\gamma_1} \right)^m \ \leq \ C_1 K^{-\gamma_1} T^{1-\gamma_1},$$

$$\text{Term 2} = H^{\frac{2}{2\alpha+1}} \sum_{m=1}^{M} 2^{m-1} (2^{m-2})^{-\gamma_2} = H^{\frac{2}{2\alpha+1}} 2^{2\gamma_2 - 1} \sum_{m=1}^{M} \left( 2^{1-\gamma_2} \right)^m \ \leq \ C_2 H^{\frac{2}{2\alpha+1}} T^{1-\gamma_2}.$$

Combining both terms gives the overall regret bound:

$$\text{Regret}(T; \pi) = \mathcal{O}\left( K^{-\frac{2\alpha\beta}{2\alpha\beta+1}} \, T^{\frac{1}{2\alpha\beta+1}} \ + \ H^{\frac{2}{2\alpha+1}} \, T^{\frac{1}{2\alpha+1}} \right).$$

# I  Proof of Theorem 13

Similar to our analysis in Section K, Algorithm 4 enters Phase 2 when the volume of target-market data is large enough to provide a more accurate estimate than transfer learning. The boundary condition can be written as

$$n_{\mathcal{K}} \ < \ n_0 \left( 1 - \frac{C_2}{C_1} \, n_0^{\frac{2\alpha(\beta-1)}{(2\alpha+1)(2\alpha\beta+1)}} \, H^{\frac{2}{2\alpha+1}} \right)^{-\frac{2\alpha\beta+1}{2\alpha\beta}},$$

which, for simplicity, we denote as

$$n_0 \ > \ \widetilde{c} \, n_{\mathcal{K}}.$$

We bound the total regret over each episode by considering two cases. Throughout, we use the per-round conversion

$$\mathbb{E}(\text{reg}_t) \ \leq \ C \, \mathbb{E} \big\| g^{(0)} - \widehat{g}^{(m)} \big\|_{L_2(P_x)}^2$$

from (25), so that episode $m$ contributes a factor $2^{m-1}$ in front of the corresponding $L_2(P_x)$ error bound.

- **Case 1:** $2^{m-2} \leq \widetilde{c} \, n_{\mathcal{K}}$. In this transfer-active regime, we directly apply Theorem 10. Combining Propositions 25 and 26 and then using (25) yields

$$\text{Reg}(m\text{th Episode}) = \sum_{t=\ell_m}^{\ell_{m+1}-1} \mathbb{E}(\text{reg}_t) \ \leq \ C_1 \sum_{t=\ell_m}^{\ell_{m+1}-1} \mathbb{E}\big( \|\widehat{g}_m - g^{(0)}\|_{L_2(P_x)}^2 \big)$$

$$\leq \ C_1 \, 2^{m-1} \left\{ n_{\mathcal{K}}^{-\frac{2\alpha\beta}{2\alpha\beta+1}} \ + \ (2^{m-2})^{-\frac{2\alpha}{2\alpha+1}} \, H^{\frac{2}{2\alpha+1}} \right\}.$$

  Summing over $M' = \lfloor \log(2\widetilde{c}\, n_{\mathcal{K}}) \rfloor$ episodes,

$$\text{Regret}(\widetilde{c}\, n_{\mathcal{K}}; \pi) \ \leq \ C_1 \sum_{m=1}^{\lfloor \log(2\widetilde{c}\, n_{\mathcal{K}}) \rfloor} 2^{m-1} \left\{ n_{\mathcal{K}}^{-\frac{2\alpha\beta}{2\alpha\beta+1}} \ + \ (2^{m-2})^{-\frac{2\alpha}{2\alpha+1}} \, H^{\frac{2}{2\alpha+1}} \right\}.$$

- **Case 2:** $2^{m-2} > \widetilde{c}\, n_{\mathcal{K}}$. In this target-only regime (Phase 2), we obtain

$$\text{Reg}(m\text{th Episode}) = \sum_{t=\ell_m}^{\ell_{m+1}-1} \mathbb{E}(\text{reg}_t) \ \leq \ C_2 \sum_{t=\ell_m}^{\ell_{m+1}-1} \mathbb{E}\big( \|\widehat{g}_m - g^{(0)}\|_{L_2(P_x)}^2 \big)$$

$$\leq \ C_2 \, 2^{m-1} \left\{ (2^{m-2})^{-\frac{2\alpha\beta}{2\alpha\beta+1}} \right\}.$$

  Summing from $m = \lceil \log(2\widetilde{c}\, n_{\mathcal{K}}) \rceil$ to $m = \lceil \log T \rceil$ episodes gives

$$\text{Regret}(\widetilde{c}\, n_{\mathcal{K}} \to T; \pi) \ \leq \ C_2 \sum_{m=\lceil \log(2\widetilde{c}\, n_{\mathcal{K}}) \rceil}^{\lceil \log T \rceil} 2^{m-1} \, (2^{m-2})^{-\frac{2\alpha\beta}{2\alpha\beta+1}}.$$

Combining the above two cases, the total regret is bounded by

$$\text{Regret}(T; \pi) \leq C_1 \sum_{m=1}^{\lfloor \log(2\widetilde{c}\, n_{\mathcal{K}}) \rfloor} 2^{m-1} \left\{ n_{\mathcal{K}}^{-\frac{2\alpha\beta}{2\alpha\beta+1}} + (2^{m-2})^{-\frac{2\alpha}{2\alpha+1}} H^{\frac{2}{2\alpha+1}} \right\}$$
$$+ C_2 \sum_{m=\lceil \log(2\widetilde{c}\, n_{\mathcal{K}}) \rceil}^{\lceil \log T \rceil} 2^{m-1} (2^{m-2})^{-\frac{2\alpha\beta}{2\alpha\beta+1}}. \tag{28}$$

We now decompose the first sum into two parts.

*Part 1.* Using $\sum_{m=1}^{M'} 2^{m-1} = 2^{M'} - 1 \lesssim 2\widetilde{c}\, n_{\mathcal{K}}$,

$$\text{Part 1} = C_1\, n_{\mathcal{K}}^{-\frac{2\alpha\beta}{2\alpha\beta+1}} \sum_{m=1}^{\lfloor \log(2\widetilde{c}\, n_{\mathcal{K}}) \rfloor} 2^{m-1} \lesssim C_1 \cdot 2\widetilde{c} \cdot n_{\mathcal{K}}^{\frac{1}{2\alpha\beta+1}}.$$

*Part 2.* Since $2^{m-1}(2^{m-2})^{-\frac{2\alpha}{2\alpha+1}} = 2^{\frac{2\alpha-1}{2\alpha+1}} \cdot 2^{\frac{m}{2\alpha+1}}$, we have

$$\text{Part 2} = C_1\, H^{\frac{2}{2\alpha+1}} \sum_{m=1}^{\lfloor \log(2\widetilde{c}\, n_{\mathcal{K}}) \rfloor} 2^{m-1}(2^{m-2})^{-\frac{2\alpha}{2\alpha+1}}$$
$$= C_1\, H^{\frac{2}{2\alpha+1}} \cdot 2^{\frac{2\alpha-1}{2\alpha+1}} \sum_{m=1}^{\lfloor \log(2\widetilde{c}\, n_{\mathcal{K}}) \rfloor} 2^{\frac{m}{2\alpha+1}} \lesssim C_1\, H^{\frac{2}{2\alpha+1}} \cdot (2\widetilde{c}\, n_{\mathcal{K}})^{\frac{1}{2\alpha+1}}.$$

For the second sum in (28), note that

$$2^{m-1}(2^{m-2})^{-\frac{2\alpha\beta}{2\alpha\beta+1}} = 2^{(m-1)-(m-2)\frac{2\alpha\beta}{2\alpha\beta+1}} = 2^{\frac{m-1}{2\alpha\beta+1}} \times (\text{constant}),$$

so the summation behaves like a geometric series with ratio $2^{1/(2\alpha\beta+1)} > 1$. Therefore

$$\sum_{m=\lceil \log(2\widetilde{c}\, n_{\mathcal{K}}) \rceil}^{\lceil \log T \rceil} 2^{m-1}(2^{m-2})^{-\frac{2\alpha\beta}{2\alpha\beta+1}} \lesssim T^{\frac{1}{2\alpha\beta+1}} - (\widetilde{c}\, n_{\mathcal{K}})^{\frac{1}{2\alpha\beta+1}}. \tag{29}$$

Combining Part 1, Part 2, and (29), we obtain

$$\text{Regret}(T; \pi) \lesssim \widetilde{c}\, n_{\mathcal{K}}^{\frac{1}{2\alpha\beta+1}} + H^{\frac{2}{2\alpha+1}} (\widetilde{c}\, n_{\mathcal{K}})^{\frac{1}{2\alpha+1}} + T^{\frac{1}{2\alpha\beta+1}} - (\widetilde{c}\, n_{\mathcal{K}})^{\frac{1}{2\alpha\beta+1}},$$

which concludes the proof.

## J  Proof of Theorem 11

We convert regret to an $L^2(P_x)$ estimation error, upper bound the total information (KL) any adaptive policy can extract under binary feedback, and then invoke Fano with a tensor-product packing built from (i) a *transferable* common block and (ii) a *non-transferable* residual block.

**Lemma 27** (Local quadratic revenue drop). *Under Assumption 1 and the local regularity above, there exists $c_* > 0$ such that for any $x \in \mathcal{X}$ and any estimate $\widehat{g}$ with $g(x), \widehat{g}(x) \in \mathcal{U}_0$,*

$$\text{rev}\big(h(\,g(x)\,); g(x)\big) - \text{rev}\big(h(\widehat{g}(x)\,); g(x)\big) \geq c_*\big(\widehat{g}(x) - g(x)\big)^2, \qquad c_* := \frac{m_{\text{rev}}\, m_h^2}{2}.$$

*Consequently,*

$$\mathbb{E}\big[\text{Reg}(T; \pi)\big] \geq c_*\, T \cdot \mathbb{E}\big[\|\widehat{g} - g\|_{L^2(P_x)}^2\big], \tag{30}$$

*where $\widehat{g}$ is the utility function implicitly induced by the policy $\pi$ through its posted prices $p_t = h(\widehat{g}(x_t^{(0)}))$ during the episode.*

*Proof.* By strong concavity at $p^*(u) = h(u)$, for any $u \in \mathcal{U}_0$ and $p$ close enough to $p^*(u)$ we have $\mathrm{rev}(p^*(u); u) - \mathrm{rev}(p; u) \geq \frac{m_{\mathrm{rev}}}{2}(p - p^*(u))^2$. Set $u = g(x)$ and $p = h(\widehat{g}(x))$. By bi-Lipschitzness of $h$ on $\mathcal{U}_0$, $|p - p^*(u)| = |h(\widehat{g}(x)) - h(g(x))| \geq m_h |\widehat{g}(x) - g(x)|$, which yields the pointwise inequality with $c_* = \frac{m_{\mathrm{rev}} m_h^2}{2}$. Summing over $t$ and taking expectations gives (30). $\square$

**Lemma 28** (Bernoulli KL smoothness). *Let $q(p, u) := 1 - F(p - u)$ and consider Bernoulli distributions with means $q(p, u)$ and $q(p, u')$. Assume $F$ has a continuous density $f$ on $[-B_\varepsilon, B_\varepsilon]$ and extends smoothly to the boundary with $f(\pm B_\varepsilon) = 0$ (or take any log-concave $F$ on $\mathbb{R}$ and restrict to a compact interval of utilities). Then there exists a finite constant*

$$C_{\mathrm{KL}} := \sup_{\delta \in \mathbb{R}} \frac{f(\delta)^2}{q(\delta)\,[1 - q(\delta)]} \; < \; \infty,$$

*such that for all $p \in \mathbb{R}$, $x \in \mathcal{X}$ and $u, u' \in \mathbb{R}$,*

$$\mathrm{KL}\Big(\mathrm{Bern}\big(q(p, u)\big) \,\big\|\, \mathrm{Bern}\big(q(p, u')\big)\Big) \; \leq \; C_{\mathrm{KL}}\,\big(u - u'\big)^2. \tag{31}$$

*Consequently, for any (possibly adaptive) policy $\pi$ interacting with the target and $K$ source markets over $T$ rounds,*

$$\mathrm{KL}\big(\mathbb{P}_{g^{(0)}, \ldots, g^K} \,\big\|\, \mathbb{P}_{g'^{(0)}, \ldots, g'^K}\big) \; \leq \; C_{\mathrm{KL}}\, T\Big(\|g^{(0)} - g'^{(0)}\|_{L^2(P_x)}^2 + \sum_{k=1}^K \|g^K - g'^K\|_{L^2(P_x)}^2\Big). \tag{32}$$

*Proof.* By the mean value theorem, $|q(p, u) - q(p, u')| = |F(p - u') - F(p - u)| \leq \sup_\delta f(\delta)\,|u - u'|$. For Bernoulli variables with means $a, b \in (0, 1)$ we have the standard bound $\mathrm{KL}(\mathrm{Bern}(a) \| \mathrm{Bern}(b)) \leq \frac{(a-b)^2}{b(1-b)}$. Combining yields (31) with $C_{\mathrm{KL}} = \sup_\delta \frac{f(\delta)^2}{q(\delta)(1 - q(\delta))}$, which is finite under the stated regularity (continuity plus compactness ensures the supremum is attained; typical families such as probit/logistic also satisfy $C_{\mathrm{KL}} \leq 1/4$). Summing the one-step inequality over time and markets and applying the chain rule for KL under adaptivity gives (32). $\square$

**Proposition 29** (Packing for the aggregation block). *Define $\mathcal{G}_A(R) := \{g = \Sigma^\beta \rho : \|\rho\|_{L^2} \leq R\}$, where $\Sigma$ is the kernel integral operator with eigensystem $(\mu_j, \varphi_j)_{j \geq 1}$ and $\beta \in (0, 1]$ (Assumption 9). There exists $c_A > 0$ such that for all $0 < \delta < R$,*

$$\log \mathcal{M}\Big(\delta;\, \mathcal{G}_A(R),\, \|\cdot\|_{L^2(P_x)}\Big) \; \geq \; c_A \left(\frac{R}{\delta}\right)^{\frac{1}{\alpha\beta}}.$$

**Proposition 30** (Packing for the debiasing block). *Let $\mathcal{G}_B(H) := \{g \in \mathcal{H}_K : \|g\|_{\mathcal{H}_K} \leq H\}$ with a bounded kernel (Assumption 7). There exists $c_B > 0$ such that for all $0 < \delta < H$,*

$$\log \mathcal{M}\Big(\delta;\, \mathcal{G}_B(H),\, \|\cdot\|_{L^2(P_x)}\Big) \; \geq \; c_B \left(\frac{H}{\delta}\right)^{\frac{1}{\alpha}}.$$

*Proof.* Let $(\mu_j, \varphi_j)$ be the eigensystem of $\Sigma$. Assumption 9 (i) states $N(\lambda) = \mathrm{Tr}\big(\Sigma(\Sigma + \lambda I)^{-1}\big) \lesssim \lambda^{-1/(2\alpha)}$. Working on a *spectral slice* $\{j : \mu_j \in (\lambda/2, \lambda]\}$, the number of coordinates in the slice satisfies $m(\lambda) \asymp N(\lambda/2) - N(\lambda) \gtrsim \lambda^{-1/(2\alpha)}$ for sufficiently small $\lambda$ (selecting a subclass saturating the effective-dimension rate is admissible for minimax lower bounds).

**(A) Aggregation block $\mathcal{G}_A(R)$.** Fix a slice $J_A(\lambda) := \{j : \mu_j \in (\lambda/2, \lambda]\}$ with cardinality $m_A(\lambda) \gtrsim \lambda^{-1/(2\alpha)}$. For $\theta \in \{\pm 1\}^{m_A}$ define

$$\rho_\theta := \frac{a}{\sqrt{m_A}} \sum_{j \in J_A} \theta_j\, \varphi_j, \qquad g_{A,\theta} := \Sigma^\beta \rho_\theta = \frac{a}{\sqrt{m_A}} \sum_{j \in J_A} \mu_j^\beta \theta_j\, \varphi_j,$$

with amplitude $a \leq R/2$ to ensure $\|\rho_\theta\|_{L^2} \leq a \leq R/2$. By the Varshamov–Gilbert bound there exists $\mathcal{C}_A \subset \{\pm 1\}^{m_A}$ with $|\mathcal{C}_A| \geq 2^{m_A/8}$ and Hamming distances at least $m_A/8$. Hence, for $\theta \neq \theta'$ in $\mathcal{C}_A$,

$$\|g_{A,\theta} - g_{A,\theta'}\|_{L^2}^2 = \frac{a^2}{m_A} \sum_{j \in J_A} \mu_j^{2\beta} (\theta_j - \theta_j')^2 \gtrsim \frac{a^2}{m_A} \cdot \left(\frac{m_A}{8}\right) \cdot \lambda^{2\beta} \asymp a^2\, \lambda^{2\beta}.$$

Thus the $L^2$-separation is at least $2\delta_A(\lambda)$ with $\delta_A(\lambda) \asymp a\,\lambda^\beta$, while the packing size obeys $\log M_A(\lambda) \geq c\,m_A(\lambda) \gtrsim \lambda^{-1/(2\alpha)}$. Feasibility in RKHS is also satisfied:

$$\|g_{A,\theta}\|_{\mathcal{H}_K}^2 = \frac{a^2}{m_A} \sum_{j \in J_A} \frac{\mu_j^{2\beta}}{\mu_j} \lesssim a^2\,\lambda^{2\beta-1}.$$

Choosing $\lambda$ small enough and $a \leq \min\{R/2, c_0\,\lambda^{(1-2\beta)/2}\}$ ensures $\|g_{A,\theta}\|_{\mathcal{H}_K} \leq R/2$. Therefore,

$$\log \mathcal{M}\big(2\delta_A;\ \mathcal{G}_A(R),\ \|\cdot\|_{L^2}\big) \gtrsim \lambda^{-1/(2\alpha)} \quad \text{with} \quad \delta_A \asymp a\,\lambda^\beta,\ a \leq R/2.$$

Eliminating $\lambda$ gives $\log \mathcal{M}(\delta_A) \gtrsim (a/\delta_A)^{1/(\alpha\beta)}$, and taking $a = R/2$ yields the stated bound.

**(B) Debiasing block $\mathcal{G}_B(H)$.** Choose a slice $J_B(\lambda)$ disjoint from $J_A(\lambda)$ with $m_B(\lambda) \gtrsim \lambda^{-1/(2\alpha)}$. For $\zeta \in \{\pm 1\}^{m_B}$ define

$$g_{B,\zeta} := \frac{b}{\sqrt{m_B}} \sum_{j \in J_B} \sqrt{\mu_j}\,\zeta_j\,\varphi_j.$$

Then $\|g_{B,\zeta}\|_{\mathcal{H}_K}^2 = \frac{b^2}{m_B} \sum_{j \in J_B} 1 = b^2$, so choosing $b \leq H/2$ ensures $\|g_{B,\zeta}\|_{\mathcal{H}_K} \leq H/2$. By Varshamov–Gilbert, there exists $\mathcal{C}_B$ with $|\mathcal{C}_B| \geq 2^{m_B/8}$ and Hamming distances at least $m_B/8$. For $\zeta \neq \zeta'$ in $\mathcal{C}_B$,

$$\|g_{B,\zeta} - g_{B,\zeta'}\|_{L^2}^2 = \frac{b^2}{m_B} \sum_{j \in J_B} \mu_j\,(\zeta_j - \zeta_j')^2 \gtrsim \frac{b^2}{m_B} \cdot \left(\frac{m_B}{8}\right) \cdot \lambda \asymp b^2\,\lambda.$$

Thus the separation is at least $2\delta_B(\lambda)$ with $\delta_B(\lambda) \asymp b\,\lambda^{1/2}$ and $\log M_B(\lambda) \gtrsim \lambda^{-1/(2\alpha)}$. Eliminating $\lambda$ yields $\log \mathcal{M}(\delta_B) \gtrsim (b/\delta_B)^{1/\alpha}$; setting $b = H/2$ gives the claim.

**Pointwise control.** Because $\|K_x\|_{\mathcal{H}_K} \leq \kappa$ (Assumption 7), we have $|g(x)| \leq \kappa\|g\|_{\mathcal{H}_K}$. Scaling the amplitudes above by a universal constant (absorbed into $c_A, c_B$) ensures $g_A(x), g_B(x) \in \mathcal{U}_0$ for all $x$, which is used in Lemma 27. $\qquad\square$

We now build *two* packing families on *disjoint* eigenspaces:

$$\mathcal{F}_A \subset \mathcal{G}_A(R), \qquad \mathcal{F}_B \subset \mathcal{G}_B(H),$$

so that for any $g_A, g_A' \in \mathcal{F}_A$ and $g_B, g_B' \in \mathcal{F}_B$,

$$\|(g_A + g_B) - (g_A' + g_B')\|_{L^2}^2 = \|g_A - g_A'\|_{L^2}^2 + \|g_B - g_B'\|_{L^2}^2,$$

and the KL bound (32) splits additively as well. We also restrict amplitudes so that $g_A(x), g_B(x) \in \mathcal{U}_0$ for all $x$, by the remark above.

**Lemma 31** (Fano for the aggregation block). *Let $\mathcal{F}_A \subset \mathcal{G}_A(R)$ be a $2\delta_A$-packing with cardinality $M_A$. For any policy $\pi$ that observes the target and $K$ source streams over $T$ rounds,*

$$\inf_{\widehat{g}} \sup_{g \in \mathcal{F}_A} \mathbb{E}\big[\|\widehat{g} - g\|_{L^2}^2\big] \geq \frac{\delta_A^2}{2}\left(1 - \frac{4\,C_{\mathrm{KL}}\,K\,T\,\delta_A^2 + \log 2}{\log M_A}\right).$$

**Lemma 32** (Fano for the debiasing block). *Let $\mathcal{F}_B \subset \mathcal{G}_B(H)$ be a $2\delta_B$-packing with cardinality $M_B$. For any policy $\pi$ (only the target data contribute here),*

$$\inf_{\widehat{g}} \sup_{g \in \mathcal{F}_B} \mathbb{E}\big[\|\widehat{g} - g\|_{L^2}^2\big] \geq \frac{\delta_B^2}{2}\left(1 - \frac{4\,C_{\mathrm{KL}}\,T\,\delta_B^2 + \log 2}{\log M_B}\right).$$

*Proof.* Apply the standard multi-hypothesis Fano inequality [27] to the packing families $\mathcal{F}_A$ and $\mathcal{F}_B$. The *average* pairwise KL over each family is bounded using (32): for the aggregation block all $KT$ Bernoulli observations contribute, yielding the factor $KT$; for the debiasing block only the $T$ target observations contribute. $\qquad\square$

Combining Lemmas 31–32 with Propositions 29–30, the calibration

$$4\,C_{\mathrm{KL}}\,K\,T\,\delta_A^2 \;\lesssim\; \log M_A \;\gtrsim\; c_A\Big(\tfrac{R}{\delta_A}\Big)^{\frac{1}{\alpha\beta}}, \qquad 4\,C_{\mathrm{KL}}\,T\,\delta_B^2 \;\lesssim\; \log M_B \;\gtrsim\; c_B\Big(\tfrac{H}{\delta_B}\Big)^{\frac{1}{\alpha}},$$

gives (after eliminating $\delta_A, \delta_B$)

$$\delta_A^2 \;\asymp\; R^{\frac{2}{2\alpha\beta+1}}\,K^{-\frac{2\alpha\beta}{2\alpha\beta+1}}\,T^{-\frac{2\alpha\beta}{2\alpha\beta+1}}, \qquad \delta_B^2 \;\asymp\; H^{\frac{2}{2\alpha+1}}\,T^{-\frac{2\alpha}{2\alpha+1}}. \tag{33}$$

Finally, define the product packing $\mathcal{F} = \{g_\nu = g_{A,\theta} + g_{B,\zeta} : g_{A,\theta} \in \mathcal{F}_A, g_{B,\zeta} \in \mathcal{F}_B\}$ so that $\log |\mathcal{F}| = \log M_A + \log M_B$ and pairwise $L^2$ distances add in quadrature. The total KL between any two elements of $\mathcal{F}$ is the sum of the block-wise KLs by (32). Applying Fano on $\mathcal{F}$ yields

$$\inf_{\widehat{g}} \sup_{g \in \mathcal{F}} \; \mathbb{E}\big[\|\widehat{g} - g\|_{L^2}^2\big] \;\gtrsim\; \delta_A^2 + \delta_B^2,$$

and combining with Lemma 27 via (30) completes the proof of the lower bound in the main text.

# K   Proof of Propositions 16 and 19

We first provide a detailed proof for Proposition 16, which follows by combining Proposition 33 and Proposition 34 using triangle inequality.

$\widehat{\boldsymbol{\beta}}^{(ag)}$ is realized using all the source samples. It's probablistic limit is $\boldsymbol{\beta}^{(ag)}$. The corresponding estimation error can be bounded by the following proposition.

**Proposition 33** (Aggregation Error). *Consider linear utility model with Assumptions 1, 2, 3 and 4 holding true. Then, there exist positive constants $c_0', c_1, c_2$ such that, for $n_{\mathcal{K}} \geq c_0'd$, the following holds with probability at least $1 - 1/d$:*

$$\|\boldsymbol{\beta}^{(ag)} - \widehat{\boldsymbol{\beta}}^{(ag)}\|_2^2 \leq c_1 \frac{d \log d}{n_{\mathcal{K}}}.$$

Moreover, the probabilistic limit $\boldsymbol{\beta}^{(ag)}$ is biased from $\boldsymbol{\beta}^{(0)} \neq \boldsymbol{\beta}^{(k)}$ in general. We then correct its bias using the primary data in target market. The estimation error of the debias term can be bounded as follows.

**Proposition 34** (Bias Correction Error). *Under conditions of Proposition 33, there exist positive constants $c_0, c_1, c_2$ such that, for $n_0 \geq c_0 s_0 \log d$, the following holds with probability at least $1 - 1/d - 2e^{-n_0/(c_0 s_0)}$:*

$$\|\widehat{\boldsymbol{\delta}} - \boldsymbol{\delta}\|_2^2 \leq c_2 \frac{s_0 \log d}{n_0}.$$

The key distinction between the Proposition 16 and Proposition 19 lies in their sample size requirements: while both require the target sample size $n_0 \geq c_0 s_0 \log d$ for valid estimation, the online-to-online setting imposes an additional constraint $n_0 \geq c_0' \frac{d}{K}$ to account for the simultaneous learning from initially limited source data across $K$ markets. This reflects the fundamental operational difference that online-to-online must handle concurrent data scarcity in both domains, whereas offline-to-online leverages pre-collected source data (implicitly assuming $n_{\mathcal{K}}$ is sufficiently large).

## K.1   Proof of Propositions 22 and 33

Let $(X^{\mathcal{K}}, Y^{\mathcal{K}})$ denote the design matrix and the response vector by row-stacking of all source data $\{\boldsymbol{x}_t^{(k)}, y_t^{(k)})\}_{t \in I^{(k)}}$ for $k \in [K]$.

By the second-order Taylor expansion around the true parameter $\boldsymbol{\beta}^{(ag)}$ we have

$$L(\widehat{\boldsymbol{\beta}}^{(ag)}) - L(\boldsymbol{\beta}^{(ag)}) = \langle \nabla L(\boldsymbol{\beta}^{(ag)}), \widehat{\boldsymbol{\beta}}^{(ag)} - \boldsymbol{\beta}^{(ag)} \rangle + \frac{1}{2} \langle \widehat{\boldsymbol{\beta}}^{(ag)} - \boldsymbol{\beta}^{(ag)}, \nabla^2 L(\widetilde{\boldsymbol{\beta}})(\widehat{\boldsymbol{\beta}}^{(ag)} - \boldsymbol{\beta}^{(ag)}) \rangle$$

for some $\widetilde{\boldsymbol{\beta}}$ on the line segment between $\boldsymbol{\beta}^{(ag)}$ and $\widehat{\boldsymbol{\beta}}^{(ag)}$. Invoking Equation 6, we have

$$\nabla L(\boldsymbol{\beta}) = \frac{1}{n_{\mathcal{K}}} \sum_{x_t \in X^{\mathcal{K}}} \xi_t(\boldsymbol{\beta})x_t, \quad \nabla^2 L(\boldsymbol{\beta}) = \frac{1}{n_{\mathcal{K}}} \sum_{x_t \in X^{\mathcal{K}}} \eta_t(\boldsymbol{\beta})x_t x_t^\top, \tag{34}$$

where $\nabla$ and $\nabla^2$ represents the gradient and the hessian *w.r.t* $\boldsymbol{\beta}$. Further,

$$\xi_t(\boldsymbol{\beta}) = -\frac{f(u_t(\boldsymbol{\beta}))}{F(u_t(\boldsymbol{\beta}))}\mathbb{I}(y_t = -1) + \frac{f(u_t(\boldsymbol{\beta}))}{1 - F(u_t(\boldsymbol{\beta}))}\mathbb{I}(y_t = +1)$$

$$= -\log' F(u_t(\boldsymbol{\beta}))\mathbb{I}(y_t = -1) - \log'(1 - F(u_t(\boldsymbol{\beta})))\mathbb{I}(y_t = +1)\,.$$

$$\eta_t(\boldsymbol{\beta}) = \left(\frac{f(u_t(\boldsymbol{\beta}))^2}{F(u_t(\boldsymbol{\beta}))^2} - \frac{f'(u_t(\boldsymbol{\beta}))}{F(u_t(\boldsymbol{\beta}))}\right)\mathbb{I}(y_t = -1) + \left(\frac{f(u_t(\boldsymbol{\beta}))^2}{(1 - F(u_t(\boldsymbol{\beta})))^2} + \frac{f'(u_t(\boldsymbol{\beta}))}{1 - F(u_t(\boldsymbol{\beta}))}\right)\mathbb{I}(y_t = +1)$$

$$= -\log'' F(u_t(\boldsymbol{\beta}))\mathbb{I}(y_t = -1) - \log''(1 - F(u_t(\boldsymbol{\beta})))\mathbb{I}(y_t = +1)\,,$$

where $u_t(\boldsymbol{\beta}) = p_t - \langle x_t, \boldsymbol{\beta}\rangle$. By lemma 14, we have

$$|u_t(\boldsymbol{\beta})| \le |p_t| + \|x_t\|_\infty \|\boldsymbol{\beta}\|_1 \le P + W \tag{35}$$

Let

$$u_F \equiv \sup_{|x| \le P+W} \left\{\max\left\{\log' F(x), -\log'(1 - F(x))\right\}\right\}$$

$$\ell_F \equiv \inf_{|x| \le P+W} \left\{\min\left\{-\log'' F(x), -\log''(1 - F(x))\right\}\right\}\,.$$

Next, we bound the gradient and hessian.

**Lemma 35.** *Let*

$$\mathcal{F} \equiv \left\{\|\nabla L(\boldsymbol{\beta}^{(ag)})\|_\infty \le 2u_F\sqrt{\frac{\log d}{n_\mathcal{K}}}\right\}\,.$$

*we have* $\mathbb{P}(\mathcal{F}) \ge 1 - 1/d$.

To bound the gradient, as $|u_t(\widetilde{\boldsymbol{\beta}})| \le P + W$, cf. Equation 35. Therefore, by definition of $\ell_F$, we have $\eta_t(\widetilde{\boldsymbol{\beta}}) \ge \ell_F$. Recalling Equation 34, we get $\nabla^2 L(\widetilde{\boldsymbol{\beta}}) \succeq \ell_F(\widetilde{X}^\top \widetilde{X})$.

By the optimality condition of $\widehat{\boldsymbol{\beta}}^{(ag)}$, we write

$$L(\widehat{\boldsymbol{\beta}}^{(ag)}) \le L(\boldsymbol{\beta}^{(ag)}).$$

Rearranging the terms and using the bound on hessian, we arrive at

$$\frac{\ell_F}{n_\mathcal{K}}\|X^\mathcal{K}(\boldsymbol{\beta}^{(ag)} - \widehat{\boldsymbol{\beta}}^{(ag)})\|^2 \le \|\nabla L(\boldsymbol{\beta}^{(ag)})\|_\infty\|\widehat{\boldsymbol{\beta}}^{(ag)} - \boldsymbol{\beta}^{(ag)}\|_1\,.$$

Choosing $\lambda \ge 4u_F\sqrt{\frac{\log d}{n_\mathcal{K}}}$, we have on set $\mathcal{F}$

$$\frac{2\ell_F}{n_\mathcal{K}}\|X^\mathcal{K}(\boldsymbol{\beta}^{(ag)} - \widehat{\boldsymbol{\beta}}^{(ag)})\|^2 \le \lambda\|\widehat{\boldsymbol{\beta}}^{(ag)} - \boldsymbol{\beta}^{(ag)}\|_1\,. \tag{36}$$

Define the event $\mathcal{B}_n$ as follows:

$$\mathcal{B}_n \equiv \left\{X \in \mathbb{R}^{n_\mathcal{K} \times d} : \sigma_{\min}(X^\top X/n_\mathcal{K}) > C_{\min}/2\right\}\,.$$

Using concentration bounds on the spectrum of random matrices with subgaussian rows ([29], Equation 5.26), there exist constants $c, c_1 > 0$ such that for $n > c_1 d$, we have $\mathbb{P}(\mathcal{B}_n) \ge 1 - e^{-cn_\mathcal{K}^2}$.

By assumption 3, the *l.h.s* of Equation 36 can be bounded by the minimum eigenvalue of its second moment matrix

$$\ell_F C_{min}\|\boldsymbol{\beta}^{(ag)} - \widehat{\boldsymbol{\beta}}^{(ag)}\|^2 \le \frac{2\ell_F}{n_\mathcal{K}}\|X^\mathcal{K}(\boldsymbol{\beta}^{(ag)} - \widehat{\boldsymbol{\beta}}^{(ag)})\|^2 \le \lambda\|\widehat{\boldsymbol{\beta}}^{(ag)} - \boldsymbol{\beta}^{(ag)}\|_1 \le \lambda\sqrt{d}\|\widehat{\boldsymbol{\beta}}^{(ag)} - \boldsymbol{\beta}^{(ag)}\|\,.$$

and therefore,

$$\|\boldsymbol{\beta}^{(ag)} - \widehat{\boldsymbol{\beta}}^{(ag)}\|^2 \le \frac{d\lambda^2}{\ell_F^2 C_{min}^2}\,.$$

## K.2 Proof of Proposition 34

By the second-order Taylor expansion, expanding around $\boldsymbol{\delta}$ we have

$$L(\widehat{\boldsymbol{\delta}} + \widehat{\boldsymbol{\beta}}^{(ag)}) - L(\boldsymbol{\delta} + \widehat{\boldsymbol{\beta}}^{(ag)}) = \langle \nabla L(\boldsymbol{\delta} + \widehat{\boldsymbol{\beta}}^{(ag)}), \widehat{\boldsymbol{\delta}} - \boldsymbol{\delta} \rangle + \frac{1}{2}\langle \widehat{\boldsymbol{\delta}} - \boldsymbol{\delta}, \nabla^2 L(\widetilde{\boldsymbol{\delta}} + \widehat{\boldsymbol{\beta}}^{(ag)})(\widehat{\boldsymbol{\delta}} - \boldsymbol{\delta})\rangle$$

for some $\widetilde{\boldsymbol{\delta}}$ on the line segment between $\boldsymbol{\delta}$ and $\widehat{\boldsymbol{\delta}}$. Again we have

$$\nabla L(\boldsymbol{\delta} + \widehat{\boldsymbol{\beta}}^{(ag)}) = \frac{1}{n_0}\sum_{t=1}^{n_0}\xi_t(\boldsymbol{\delta} + \widehat{\boldsymbol{\beta}}^{(ag)})x_t, \quad \nabla^2 L(\boldsymbol{\delta} + \widehat{\boldsymbol{\beta}}^{(ag)}) = \frac{1}{n_0}\sum_{t=1}^{n_0}\eta_t(\boldsymbol{\delta} + \widehat{\boldsymbol{\beta}}^{(ag)})x_t x_t^\top,$$

where $\nabla$ and $\nabla^2$ represents the gradient and the hessian *w.r.t.* $\boldsymbol{\delta}$.

Next, we bound the gradient and hessian. According to Lemma 35, define

$$\mathcal{F} \equiv \left\{ \|\nabla L(\boldsymbol{\delta} + \boldsymbol{\beta}^{(ag)})\|_\infty \leq 2u_F\sqrt{\frac{\log d}{n_0}} \right\}.$$

we have $\mathbb{P}(\mathcal{F}) \geq 1 - 1/d$, $\eta_t(\widetilde{\boldsymbol{\delta}} + \widehat{\boldsymbol{\delta}}^{(ag)}) \geq \ell_F$, and $\nabla^2 L(\widetilde{\boldsymbol{\delta}} + \widehat{\boldsymbol{\delta}}^{(ag)}) \succeq \ell_F(\widetilde{X}^\top \widetilde{X})$.

By the optimality condition of $\widehat{\boldsymbol{\delta}}$, we write

$$L(\widehat{\boldsymbol{\delta}} + \widehat{\boldsymbol{\beta}}^{(ag)}) + \lambda\|\widehat{\boldsymbol{\delta}}\|_1 \leq L(\boldsymbol{\delta} + \widehat{\boldsymbol{\beta}}^{(ag)}) + \lambda\|\boldsymbol{\delta}\|_1. \tag{37}$$

Using the bound on hessian and gradient, choosing $\lambda \geq 4u_F\sqrt{\frac{\log d}{n_0}}$, we have on set $\mathcal{F}$

$$\frac{2\ell_F}{n_0}\|X^{(0)}(\boldsymbol{\delta} - \widehat{\boldsymbol{\delta}})\|^2 + 2\lambda\|\widehat{\boldsymbol{\delta}}\|_1 \leq \lambda\|\widehat{\boldsymbol{\delta}} - \boldsymbol{\delta}\|_1 + 2\lambda\|\boldsymbol{\delta}\|_1. \tag{38}$$

By Assumption 4, $\boldsymbol{\delta}$ is sparse. Let $S = \text{supp}(\boldsymbol{\delta}^{(ag)})$. On the l.h.s. using triangle inequality, we have

$$\|\widehat{\boldsymbol{\delta}}^{(ag)}\|_1 = \|\widehat{\boldsymbol{\delta}}_S^{(ag)}\|_1 + \|\widehat{\boldsymbol{\delta}}_{S^c}^{(ag)}\|_1 \geq \|\widehat{\boldsymbol{\delta}}_S^{(ag)}\|_1 - \|\widehat{\boldsymbol{\delta}}_S^{(ag)} - \boldsymbol{\delta}_S^{(ag)}\|_1 + \|\widehat{\boldsymbol{\delta}}_{S^c}^{(ag)}\|_1.$$

On the *r.h.s.*, we have

$$\|\widehat{\boldsymbol{\delta}}^{(ag)} - \boldsymbol{\delta}^{(ag)}\|_1 = \|\widehat{\boldsymbol{\delta}}_S^{(ag)} - \boldsymbol{\delta}_S^{(ag)}\|_1 + \|\widehat{\boldsymbol{\delta}}_{S^c}^{(ag)}\|_1.$$

Using these two equations in Equation 38, we get

$$\frac{2\ell_F}{n_0}\|X^{(0)}(\boldsymbol{\delta}^{(ag)} - \widehat{\boldsymbol{\delta}}^{(ag)})\|^2 + \lambda\|\widehat{\boldsymbol{\delta}}_{S^c}^{(ag)}\|_1 \leq 3\lambda\|\widehat{\boldsymbol{\delta}}_S^{(ag)} - \boldsymbol{\delta}_S^{(ag)}\|_1. \tag{39}$$

We next write

$$\begin{aligned}
\frac{2\ell_F}{n_0}\|X^{(0)}(\boldsymbol{\delta}^{(ag)} - \widehat{\boldsymbol{\delta}}^{(ag)})\|^2 + \lambda\|\widehat{\boldsymbol{\delta}}^{(ag)} - \boldsymbol{\delta}^{(ag)}\| &= \frac{2\ell_F}{n_0}\|X^{(0)}(\boldsymbol{\delta}^{(ag)} - \widehat{\boldsymbol{\delta}}^{(ag)})\|^2 + \lambda\|\widehat{\boldsymbol{\delta}}_S^{(ag)} - \boldsymbol{\delta}_S^{(ag)}\| + \lambda\|\widehat{\boldsymbol{\delta}}_{S^C}^{(ag)}\| \\
&\overset{(a)}{\leq} 4\lambda\|\widehat{\boldsymbol{\delta}}_S^{(ag)} - \boldsymbol{\delta}_S^{(ag)}\|_1 \overset{(b)}{\leq} 4\lambda\sqrt{s_0}\|\widehat{\boldsymbol{\delta}}_S^{(ag)} - \boldsymbol{\delta}_S^{(ag)}\|_2 \\
&\overset{(c)}{\leq} \frac{4\lambda\sqrt{2s_0}}{\sqrt{n_0 C_{\min}}}\|X^{(0)}(\widehat{\boldsymbol{\delta}}^{(ag)} - \boldsymbol{\delta}^{(ag)})\|_2, \\
&\overset{(d)}{\leq} \frac{\ell_F}{n_0}\|X^{(0)}(\widehat{\boldsymbol{\delta}}^{(ag)} - \boldsymbol{\delta}^{(ag)})\|_2^2 + \frac{8\lambda^2 s_0}{\ell_F C_{\min}},
\end{aligned}$$

where (a) follows from Equation 39; (b) holds for Cauchy-Schwarz inequality; and (c) by RE condition ([19], Proposition 23), which holds for $\widehat{\Sigma}^{(0)} = ((X^{(0)})^\top X^{(0)})/n_0$ with $\kappa(\widehat{\Sigma}^{(0)}, s_0, 3) \geq \sqrt{C_{\min}/2}$; and (d) follows from the inequality $2|ab| \leq ca^2 + \frac{b^2}{c}$.

Rearranging the terms, we obtain

$$\frac{\ell_F}{n_0}\|X^{(0)}(\boldsymbol{\delta}^{(ag)} - \widehat{\boldsymbol{\delta}}^{(ag)})\|^2 + \lambda\|\widehat{\boldsymbol{\delta}}^{(ag)} - \boldsymbol{\delta}^{(ag)}\| \leq \frac{8s_0\lambda^2}{\ell_F C_{\min}}.$$

Applying the RE condition again to the *l.h.s*, we get

$$C_{\min}\frac{\ell_F}{2}\|\boldsymbol{\delta}^{(ag)} - \widehat{\boldsymbol{\delta}}^{(ag)}\|_2^2 \leq \frac{\ell_F}{n_0}\|X^{(0)}(\boldsymbol{\delta}^{(ag)} - \widehat{\boldsymbol{\delta}}^{(ag)})\|^2 \leq \frac{8s_0\lambda^2}{\ell_F C_{\min}}$$

and therefore,

$$\|\boldsymbol{\delta} - \widehat{\boldsymbol{\delta}}\|_2^2 \leq \frac{16s_0\lambda^2}{\ell_F^2 C_{\min}^2}.$$

## L    Proof of Propositions 18 and 21

Proposition 36 gives a tighter bound for the estimation error of the debias term $\widehat{\delta}$ as $n_0$ gets larger.

**Proposition 36** (Bias Correction Error). *Consider linear utility model with Assumptions 1, 2, 3 and 4 holding true. There exist positive constants $c_0, c_3, c_4$ such that, for $n_0 \geq c_0 d$, the following holds:*

$$\mathbb{E}(\|\widehat{\boldsymbol{\delta}} - \boldsymbol{\delta}\|_2^2) \leq c_5\frac{(s_0 + 1)\log d}{n_0} + 4W^2 e^{-c_3 n_0^2}.$$

The proof follows Proposition 12 from Javanmard and Nazerzadeh [19]. Combining Propositions 33 and 36 using triangle inequality gives the stated result.

## M    Proof of Lemmas

### M.1    Proof of Lemma 14

By Assumption 3 we have $\|\widehat{\boldsymbol{\beta}}^{(0)}\|_1 \leq W$ and $|x_t^{(0)} \cdot \widehat{\boldsymbol{\beta}}^{(0)}| \leq W$ for all $t, k$. The lemma holds beacause $h$ is a continuous function and continuous functions on a closed interval are bounded.

### M.2    Proof of Lemma 15

Recalling the definition $h(u) = u + \phi^{-1}(-u)$, we have $h'(u) = 1 - 1/\phi'(\phi^{-1}(-u))$. Since $\phi$ is strictly increasing by Assumption 1, we have $h'(u) < 1$.

### M.3    Proof of Lemma 35

According to the definition of $u_F$, we have $|\xi_t(\boldsymbol{\beta}^{(ag)})| \leq u_F$. Further, recall that the sequences $\{p_t\}_{t=1}^n$ and $\{x_t\}_{t=1}^n$ are independent of $\{\varepsilon_t\}_{t=1}^n$. Therefore, $\{u_t(\boldsymbol{\beta}^{(0)})\}_{t=1}^T$ and $\{\varepsilon_t(\boldsymbol{\beta}^{(0)})\}_{t=1}^T$ are independent and by Equation 2, we have $\mathbb{E}[\xi_t(\boldsymbol{\beta}^{(ag)})] = \mathbb{E}[\mathbb{E}[\xi_t(\boldsymbol{\beta}^{(ag)})|u_t(\boldsymbol{\beta}^{(ag)})]] = 0$, which gives $\mathbb{E}[\nabla L(\boldsymbol{\beta}^{(ag)})] = 0$.

By applying *Azuma-Hoeffding inequality* to one of $d$ coordinates of feature vectors,

$$\|\nabla L(\boldsymbol{\beta}^{(ag)}) - \mathbb{E}[\nabla L(\boldsymbol{\beta}^{(ag)})] \geq \alpha\| = \|\nabla L(\boldsymbol{\beta}^{(ag)}) \geq \alpha\| \leq \exp\{\frac{-\alpha^2}{2\sum_{i=1}^{n_\mathcal{K}}\sigma_i^2}\} \leq \frac{1}{d^2}$$

where $\alpha = 2v_F\sqrt{n_\mathcal{K}\log d}$, $|\sigma_i| < v_F$. Following a union bounding over $d$ coordinates,

$$\|\nabla L(\boldsymbol{\beta}^{(ag)})\|_\infty = \mathbb{P}\left(\bigcup_{i=1}^d A_i\right) \leq \sum_{i=1}^d \mathbb{P}(A_i) = \frac{1}{d}$$

The result follows.

### M.4    Proof of Corollary 17,  20 and  23

Here we provide the detailed proof for Corollary 17, which also works for Corollary 20 and 23.

We let $\mathcal{G}$ be the event that Equation 17 holds true. Then by Proposition 16 we have $\mathbb{P}(\mathcal{G}) \leq 1 - 2/d - 2e^{-n_0/(c_0 s_0)}$.

$$
\begin{aligned}
\mathbb{E}(\|\widehat{\boldsymbol{\beta}} - \boldsymbol{\beta}^{(0)}\|_2^2) &= \mathbb{E}[(\|\widehat{\boldsymbol{\beta}} - \boldsymbol{\beta}^{(0)}\|_2^2) \cdot \mathbb{I}_\mathcal{G}] + \mathbb{E}[(\|\widehat{\boldsymbol{\beta}} - \boldsymbol{\beta}^{(0)}\|_2^2) \cdot \mathbb{I}_{\mathcal{G}^c}] \\
&\leq c_1 \frac{d \log d}{n_\mathcal{K}} + c_2 \frac{s_0 \log d}{n_0} + 4W^2 \mathbb{P}(\mathcal{G}^c) \\
&\leq c_1 \frac{d \log d}{n_\mathcal{K}} + c_2 \frac{s_0 \log d}{n_0} + 4W^2 \left( \frac{2}{d} + 2e^{\frac{-n_0}{c_0 s_0}} \right).
\end{aligned}
$$

