# OpenReview forum: "Transfer Faster, Price Smarter: Minimax Dynamic Pricing under Cross-Market Preference Shift"
_NeurIPS.cc/2025/Conference — NeurIPS 2025 spotlight_

### Official Review · Reviewer_dMHy · 2025-06-13

**Clarity:** 3
**Significance:** 2
**Originality:** 2
**Rating:** 4
**Confidence:** 3

**Summary:**

This paper studies a dynamic pricing problem with additional data from auxiliary markets, where each auxiliary market has mean utilities differing by a structured preference shift. To address this, the paper introduces the Cross-Market Transfer Dynamic Pricing (CM-TDP) algorithm, which is designed to accommodate such model shifts. The paper provides minimax-optimal regret bounds for CM-TDP and supports the proposed method through numerical experiments demonstrating its practical effectiveness.

**Questions:**

1. In linear bandits, it is common to include some form of regularization when performing maximum likelihood estimation (MLE). I was wondering why Equation (6) in Algorithm 2 does not include any regularization term?
2. It would be very helpful if the authors could clearly highlight the technical novelty in their theoretical analysis, particularly in comparison to prior work. For example, does the proof involve any new concentration inequalities that are specifically tailored to the debiasing step or the transfer learning setting?

**Ethical Concerns:**

["NO or VERY MINOR ethics concerns only"]

**Final Justification:**

Please check Strengths And Weaknesses.

**Limitations:**

See weaknesses.

**Quality:**

3

**Strengths And Weaknesses:**

Strengths: The paper is generally well-written and organized. It is interesting to consider and solve a transfer learning situation for dynamic pricing problems. The proposed algorithm is supported by solid theoretical results.

Weaknesses:
1. The underlying components of CM-TDP are known techniques. My opinion is that the novelty of the algorithm lies in the debiasing part to leverage data from auxiliary markets (Algorithm 2). However, the paper lacks sufficient explanation or intuition behind the algorithm's design.
2. One (potentially) main contribution of the paper is the solid regret analysis. However, since many components in the algorithm (e.g., MLE, LASSO-like debiasing, epoch-based learning) are well-studied techniques, it would be helpful if the paper could highlight its technical novelty in the proof compared to previous works.
3. Some assumptions are limited. For example, it seems the algorithm lacks a mechanism to "avoid" different markets (where $s_0$ is large in Assumption 4) and would be fooled by such a market.  Further, assuming the knowledge of distribution function $F$ is also not  practical and there have been several works to relax that assumption (e.g., [1,2]).

[1] Jianqing Fan, Yongyi Guo, and Mengxin Yu. Policy optimization using semiparametric models for dynamic pricing. Journal of the American Statistical Association, 119(545):552–564, 2024.

[2] Virag Shah, Ramesh Johari, and Jose Blanchet. Semi-parametric dynamic contextual pricing. In Proc. Neural Information Processing Systems, 2019.

---

> ### Author Rebuttal · Authors · 2025-07-30
>
> Thank you very much for your thoughtful and constructive comments. We have revised our manuscript carefully to address your feedback, as detailed in our responses to weaknesses (**W**) and questions (**Q**) below.
>
> ---
>
> **W1** A key premise of our work is that while customer preferences differ across markets, these differences are typically structured and of low-complexity. For example, in food delivery platforms:
> - Customers in coastal cities may value fresh seafood availability, while inland customers prefer comfort foods.
> - Preferences for **universal** factors like delivery speed, order tracking, and menu navigation tend to remain **stable** across markets.
> - Colder regions may exhibit stronger demand for hot beverages, while warmer areas favor cold desserts.
>
> This provides a concrete illustration that differences in utility preferences typically arise only along a few sparse dimensions, which can be captured by our linear sparse difference setting.
>
> In a **nonparametric (RKHS)** setting, similar structured differences arise naturally when considering customer populations with distinct preferences. For example, consider a restaurant chain operating across diverse locations:
> - Most customer populations share broadly similar and smoothly varying sensitivity to price changes or promotions.
> - However, specific populations—such as students near college campuses or higher-income residents in urban centers—might have distinct nonlinear responses. Students may exhibit sharp sensitivity to discounts at lower price points, while urban high-income customers might have smoother, less elastic price responses at premium levels.
>
> These differences between populations typically form a structured, smooth discrepancy rather than arbitrary deviations, aligning naturally with our RKHS-based smoothness assumptions.
>
> The intuition behind our **debiasing step (Algorithm 2)** arises naturally from a structured handling of the **bias-variance tradeoff**. Pooling all market data produces a low-variance, high-bias estimate, as it ignores cross-market differences. Instead of directly relying on this aggregated estimate, we treat it as a “rough draft” and perform careful corrections. This resembles an editor who adjusts only problematic sentences, preserving valuable shared information. The debiasing stage acts as a precision tool that corrects for local market deviations without discarding useful global patterns.
>
> Our approach is also **theoretically grounded**: the derived regret bounds explicitly capture the trade-off between the magnitude of market differences (e.g., sparsity level $s_0$ in the linear setting, or RKHS norm $H$ in the nonparametric setting) and the required amount of target-specific data. Practically, if two markets differ substantially, more target data is naturally required to reliably correct biases. In implementation, **market similarity** can be assessed using domain expertise or empirically via held-out datasets—by comparing estimated utilities or observed purchase behaviors—before deciding to transfer.
>
> ---
>
> **W2 (Q2)** The core challenge addressed by CM-TDP is transferring information across markets under structured but unknown utility heterogeneity. While the algorithm integrates established tools—MLE, Lasso-type debiasing, epoch-based learning—its novelty lies precisely in their principled combination within a binary-feedback dynamic pricing setting subject to latent utility shifts.
>
> Our regret analysis introduces key **technical innovations** that distinguish it from previous single-task dynamic pricing and multi-task bandit results:
>
> * **Debiasing Under Utility Shift**: We establish a novel **bias-corrected aggregation** framework explicitly designed for cross-market transfer. The regret decomposition carefully separates and bounds aggregation errors (from pooled estimation) and debiasing errors (market-specific corrections), enabling rigorous quantification of transfer effectiveness under structured heterogeneity.
>
> * **Refined Estimation Error Bounds**: Unlike standard single-task analyses, we derive specialized estimation bounds accounting explicitly for structured task differences—**sparse coefficient differences** in linear utilities (Propositions 13–15) and **covariance operator-smooth differences** in the RKHS setting (Propositions 21–23). These bounds are technically non-trivial and required new probabilistic and analytic techniques tailored for our transfer pricing setting.
>
> * **Episode-based Error Control**: Our episodic learning scheme, which periodically recalibrates the aggregated and debiased estimators with increasing batch sizes, required a precise and novel analysis. The resulting non-asymptotic regret bounds explicitly characterize the trade-off between sample complexity and model shift across episodes, avoiding typical loose error accumulations.
>
> We will revise the manuscript to highlight these specific theoretical contributions more explicitly in the main text and theoretical appendices.
>
> ---
>
> **W3** We appreciate this important point. While our assumptions on structured market differences and known distribution $ F $ simplify theoretical analysis, we agree that practical considerations require additional clarification.
>
> **(1) Handling large market differences:**
>  Our theoretical analysis explicitly quantifies how effectively transfer learning can leverage source markets as a function of two key factors:
>
> - Magnitude of utility differences relative to dimension ($s_0/d$ or RKHS norm $H$).
> - Relative availability of source versus target data.
>
> Specifically, our regret bounds demonstrate that larger market discrepancies directly increase the required amount of **target-only data** to achieve accurate estimation. In other words, as the heterogeneity ($s_0$ or $H$) grows, transfer remains beneficial only if a sufficiently large target dataset is available for effective debiasing. Our results thus offer practical guidance on how much target data to collect, given expert domain knowledge or estimates of $s_0$ or $H$.
>
> This analysis aligns with a fundamental principle in transfer learning: if markets differ substantially, indiscriminately pooling data (common in industry practice) can be harmful. Our framework addresses this by providing a theoretically grounded method that leverages market similarity when present, but gracefully reverts toward target-only learning as dissimilarity increases—improving significantly over naïve pooling methods.
>
> **(2) Regarding the known noise distribution $F(\cdot)$:** Under the linear utility setting, for clarity and analytical convenience, we assume a known noise distribution $F(\cdot)$. However, our theoretical guarantees actually depend only on certain **regularity properties** (such as smoothness and monotonicity), not the exact parametric form of $F$.
>
> Importantly, our framework can naturally extend to scenarios with **unknown noise distributions** via semi-parametric or nonparametric estimation techniques, as developed in recent works (e.g., [1-3]). Our aggregate-then-debias architecture remains conceptually valid in such scenarios, though rigorously addressing unknown $F^{(k)}$ and possible transfer for $F^{(k)}$ involve additional technical complexity beyond the current scope. We consider this a promising direction for future exploration and will explicitly clarify this extension potential in the revised manuscript.
>
> ---
>
> **Q1** Thank you for raising this insightful question. In the aggregation step, we use **unregularized MLE** because the source market data is abundant, providing sufficient samples to estimate source-market parameters $\beta^{(k)}$ without imposing sparsity or other high-dimensional regularization. Hence, there is no explicit need to introduce penalty terms at this stage. Nevertheless, **if desired**, incorporating a regularization term into the aggregation stage is straightforward, and the corresponding penalized estimation analysis follows directly from existing methods. We will clarify this flexibility in the revised manuscript.
>
> ---
>
>
> **References**
>
> [1] Fan, J et al. Policy optimization using semiparametric models for dynamic pricing. JASA, 2024.
>
> [2] Shah, V. et al. Semi-parametric dynamic contextual pricing. NeurIPS, 2019.
>
> [3] Chen, E. et al. Dynamic Contextual Pricing with Doubly Nonparametric Random Utility Models, arXiv:2405.06866.

---

> > ### Comment · Reviewer_dMHy · 2025-08-05
> >
> > Thank you for the rebuttal, which provides helpful explanations and clarifications. I maintain my positive score for this work.

---

### Official Review · Reviewer_Gb5J · 2025-06-27

**Clarity:** 3
**Significance:** 3
**Originality:** 3
**Rating:** 5
**Confidence:** 3

**Summary:**

This paper studies the cross-market transfer dynamic pricing problem. The proposed framework can handle both offline-to-online and online-to-online settings and can accommodate linear and nonlinear utilities through reproducing kernel Hilbert Spaces (RKHS). Furthermore, the method can work for multiple heterogeneous source markets whose mean utilities differ from the target under certain assumptions. Numerical results demonstrate the potential gain by leveraging information from multiple source markets.

**Questions:**

1.	The proposed method is for the utilities with a structured shift from the target utility. What happens if the assumption is wrong? How do we verify the assumption in practice for a safe and reliable application of the proposed method?
2.	For RKHS, when the linear kernel (inner product) is used, the resulting function becomes linear. Please elaborate the results for the RKHS with the linear kernel and compare it with the results for the linear case.
3.	Since it doesn’t appear in the bound, how does the dimension d affect the bound for the RKHS case?
4.	For the numerical results, the highest dimension considered is 20 which is quite low. How high dimensions can the method handle? It will be helpful to include some numerical results for higher dimensions such as in hundreds or thousands.
5.	The patterns for the linear and RKHS in Figures 2 and 3 are almost identical. This is somewhat surprising. More elaborations will be helpful.
6.	The simulation only considered identical and sparse settings. It will be important to explore denser differences and show how stable the method is as the differences vary from sparse to dense.
7.	It is important to include some results on computational complexity and time comparisons.

**Ethical Concerns:**

["NO or VERY MINOR ethics concerns only"]

**Final Justification:**

The authors have carefully addressed all my questions and concerns. I would like to recommend to accept the paper.

**Limitations:**

The authors mentioned very briefly towards the end that “Future directions include automated market-similarity detection and extensions to multi-product settings”. In general, one challenging aspect is to verify that the markets follow the assumptions imposed for the proposed method. If not being able to gain from source markets, one needs to ensure at least not hurt the results without using transfer learning. More discussions on these are needed.

**Quality:**

3

**Strengths And Weaknesses:**

The proposed transfer learning framework under utility shifts is quite broad and can work for both online-to-online and offline-to-online regimes. Furthermore, the authors consider both the linear utility class and a general nonparametric utility class through RKHS. This yields great flexibility for the proposed framework.

The proposed work assumes the utilities differ from the target by a structured shift. If true, as shown in the numerical results, the improvement can be substantial. However, it is not clear how sensitive the proposed method is to the violation of the assumptions. Furthermore, more discussions and comparisons between the linear and nonparametric cases are needed. In general, one would expect better bounds for the linear case than the nonlinear one due to the smaller class of functions for the linear case.

---

> ### Author Rebuttal · Authors · 2025-07-30
>
> Thank you for the positive feedback and insightful questions. We have updated our manuscript to address the following points.
>
> ---
>
> **Q1** Our framework assumes that the utility functions across markets differ in **structured ways** (e.g., sparse shifts for linear models, smooth residuals for RKHS). These assumptions are critical for regret guarantees—but **not required for correctness**. If the assumptions are violated:
>
> - **Statistical efficiency may degrade**, but the algorithm remains valid.
> - Our bias-corrected estimator smoothly interpolates between full pooling and target-only learning, limiting downside risk.
>
> **Verifying similarity in practice** can be done via holdout evaluation: one can estimate utility functions on each source independently and compare them to the target using $\ell_2$ or RKHS metrics (e.g., gradient norm or empirical discrepancy on covariates).
>
> We will add a brief discussion in the revision and highlight this as a direction for adaptive transfer tuning.
>
> ---
>
> **Q2** Thank you for raising this insightful question. Indeed, using a linear kernel $K(x,x') = \langle x,x'\rangle$ in the RKHS reduces the hypothesis space to linear functions, making the resulting estimator functionally equivalent to linear regression with regularization. However, there are several subtle distinctions worth highlighting in our approach:
>
> - In our **linear model setting**, we explicitly use the knowledge of $F$ and **maximum likelihood estimation (MLE)** without regularization at the aggregation stage due to abundant source data, followed by sparsity-driven debiasing with explicit $\ell_1$-regularization to exploit structured cross-market differences.
> - In contrast, our **RKHS nonparametric model** employs kernel logistic regression to estimate a different object without the knowledge of $F$: the **log-odds function $\ell(p,x)$** defined over the joint space of price and covariates, using a kernel $K_{\text{joint}}$. This log-odds function is conceptually distinct from the structural utility mean function $\mathring{g}^{(k)}(x)$ analyzed theoretically. Under a linear kernel for $K_{\text{joint}}$, the resulting estimator is essentially a regularized logistic regression model that is linear in $(p,x)$.
> - Thus, while the **functional forms align** when using a linear kernel, the two scenarios differ in critical ways: the linear scenario directly estimates structural utility parameters with tailored MLE and explicit sparsity-based debiasing, while the RKHS scenario estimates purchase probabilities without knowledge of $F$ via a regularized logistic regression formulation over the joint input space.
>
> Our theoretical analysis accounts explicitly for these methodological differences. Notably, the regret bound in the linear setting achieves a faster (logarithmic) rate due to the parametric structure, whereas the RKHS approach—with or without a linear kernel—incurs polynomial regret due to implicit regularization and complexity from modeling the joint space explicitly.
>
> We will clarify these points explicitly in the revised manuscript, emphasizing the distinctions in modeling assumptions, estimation strategies, and resulting theoretical guarantees between the linear and RKHS settings.
>
> ---
>
> **Q3** In the RKHS regret bound, the ambient dimension $d$ does not appear explicitly because the rate depends on the effective dimension determined by the kernel's eigenvalue decay. Formally, $d$ influences the trace and spectrum of the covariance operator $\Sigma$, which determines the effective dimension $N(\lambda) \sim \lambda^{-1/(2\alpha)}$. Thus, faster decay (e.g., smoother kernels or lower-dimensional inputs) results in lower regret. *We will revise the manuscript to clarify that although $d$ is not explicit, it affects the regret via spectral complexity.*
>
> ---
>
> **Q4 & 6** We appreciate your valuable suggestions regarding more comprehensive numerical validation. While submission constraints prevent us from including visual representations in this rebuttal, we have conducted additional experiments addressing the key points you raised, including high-dimensional cases, dense difference scenarios, and wall-clock time comparisons. Due to time constraints, we present only the essential numerical results here, with more extensive experimental validation to be included in the revised manuscript.
>
> **Cumulative Regret**
>
> |                 Configuration                  | linear-rmlp | online($K=1,3,5,10$) | offline ($n_\mathcal{k}=50,100,200,500$) |
> | :----------------------------------------------: | :-----------: | :-------------------------------: | :----------------------------------------------------------: |
> |$d=100,\|\beta^{(0)}-\beta^{(k)}\|_0\leq 0.3d$ | 1374        |        (894, 690,617,490)         |                     (1063, 975,942,763)                      |
> | $d=10,\|\beta^{(0)}-\beta^{(k)}\|_0\leq 0.6d$  | 62          |          (41,18,13, 10)           |                        (32,25,18, 13)                        |
> |$d=15,\|\beta^{(0)}-\beta^{(k)}\|_0\leq 0.6d$  | 95          |           (74,39,27,20)           |                        (79,42,30,26)                         |
> |$d=20,\|\beta^{(0)}-\beta^{(k)}\|_0\leq 0.6d$  | 131         |          (98, 58,37,21)           |                        (101,80,52,41)                        |
>
>
>
> **Running time (in seconds)**
>
> |                 Configuration                  | linear-rmlp | online($K=1,3,5,10$) | offline ($n_\mathcal{k}=50,100,200,500$) |
> | :----------------------------------------------: | :-----------: | :-------------------------------: | :----------------------------------------------------------: |
> |$d=100,\|\beta^{(0)}-\beta^{(k)}\|_0\leq 0.3d$ | 386        |        (794, 814,866,972)         |                     (563, 597,642,663)                      |
> | $d=10,\|\beta^{(0)}-\beta^{(k)}\|_0\leq 0.3d$  | 31          |          (55,63,71, 77)           |                        (43,53,58, 64)                        |
> |$d=15,\|\beta^{(0)}-\beta^{(k)}\|_0\leq 0.3d$  | 42          |           (80,85,93,101)           |                        (58,64,70,77)                         |
> |$d=20,\|\beta^{(0)}-\beta^{(k)}\|_0\leq 0.3d$  | 48         |          (101, 109,121,128)           |                        (69,74,82,89)                        |
> | $d=10,\|\beta^{(0)}-\beta^{(k)}\|_0\leq 0.6d$  | 33          |          (59,64,71, 78)           |                        (46,51,57, 63)                        |
> |$d=15,\|\beta^{(0)}-\beta^{(k)}\|_0\leq 0.6d$  | 40          |           (81,89,96,104)           |                        (51,57,63,72)                         |
> |$d=20,\|\beta^{(0)}-\beta^{(k)}\|_0\leq 0.6d$  | 51         |          (103, 110,119,126)           |                        (73,79,82,86)                        |
>
> ---
>
> **Q5** Thank you for pointing this out. While the regret patterns for linear and RKHS utilities in Figures 1 and 2 appear similar, they are not identical. This similarity arises because both results illustrate the gain from transfer learning relative to their respective single-market baselines, and this gain scales as a polynomial function of $K^{-1}$. Since both experiments share the same set of $K$ values, they naturally produce similar scaling trends.
>
> Specifically, we employed RBF kernel ($\alpha=d/2,\beta=1$) in the simulation, resulting in an upper bound of $\mathcal{O}\left( K^{-\frac{d}{d+1}} T^{\frac{1}{d+1}} + H^{\frac{4}{d+2}} T^{\frac{2}{d+2}} \right)$, where $K^{-d/d+1}$ approaches $K^{-1}$ as $d$ grows.
>
> However, the explicit polynomial rates differ between linear and RKHS cases, due to their distinct complexities—linear models yield sharper (logarithmic) scaling, while RKHS models yield slower polynomial scaling. These intrinsic rate differences are indeed reflected in the absolute regret gaps between linear and RKHS settings (15–30% across experiments).
>
>
> We will clarify this explanation explicitly in the revised manuscript.
>
> ---
>
> **Q7** We now address the question regarding CM-TDP’s computational costs across settings.
>
> **Linear Utility Model**
>
> - **Online-to-Online Transfer**
>   - Each Episode:
>     - **Aggregation**: $O(d^2 \ell_{m-1} K \cdot N)$ for MLE using source market data from episode $m-1$, where $N$ represents the number of optimization iterations.
>     - **Debiasing**: $O(d^2 \ell_{m-1} \cdot N)$ for Lasso regression on target market data.
>   - Cumulative regret over $\log T$ episodes: $\mathcal{O}(d^2 T K\cdot N)$
> - **Offline-to-Online Transfer**
>   - Each Episode:
>     - **Phase 1 (Transfer)**: $O(d^2 n_{\mathcal{K}} \cdot N)$ for MLE on aggregated source market data, and $O(d^2 l_{m-1} \cdot N)$ per episode for bias correction.
>     - **Phase 2 (No Transfer)**: Standard linear MLE: $O(d^2 l_{m-1} \cdot N)$.
>   - Cumulative regret: $O(d^2 (n_\mathcal{K} + T) \cdot N)$
> - **Implementation Note**: The dependence on dimension $d$ varies by optimization method. Newton's method (our implementation) achieves faster convergence (smaller $N$) but maintains $d^2$ dependence. Gradient descent reduces the dependence to $O(d)$ per iteration but requires more iterations ($N$ increases correspondingly).
>
> **RKHS Utility Model**
>
> Base Complexity: KRR with $n$ samples requires $O(n^3)$ for exact matrix inversion.
>
> - **Cumulative regret**
>   - Online-to-Online Transfer: $O(T^3 K^3 \cdot N)$
>   - Offline-to-Online Transfer: $O((n_\mathcal{K}^3 + T^3) \cdot N)$
> - **Implementation Note**: We use **approximate kernel methods** (e.g., Nyström, random features) to reduce runtime to **linear scaling in $\mathcal{N}(\lambda)$**, the effective dimensionality of the feature map.
>
> *We will add a computational discussion in the appendix to clarify these scaling properties and practical trade-offs.*

---

> > ### Comment · Reviewer_Gb5J · 2025-08-02
> >
> > I appreciate the authors’ thorough rebuttal, which includes helpful clarifications and additional results. I am confident that, if the paper is accepted, the authors will effectively utilize the additional page to further improve the clarity and quality of the presentation.

---

### Official Review · Reviewer_9euS · 2025-06-29

**Clarity:** 3
**Significance:** 2
**Originality:** 3
**Rating:** 4
**Confidence:** 4

**Summary:**

The authors develop contextual dynamic pricing algorithms that can leverage auxiliary markets' data to improve the performance on the target market. The difference and similarities between different markets are illustrated by the utility functions. There are multiple algorithms developed, tackling several settings including different utility function class (linear parametric models, RKHS nonparametric ones), and different auxiliary markets' data (offline static, online concurrent).

**Questions:**

1. Justification of "minimax optimality".
In lines 10 - 11 (abstract), 52, 310, the authors claim "matching information-theoretic lower bounds up to logarithmic factors", "matching known lower bounds", "matching information-theoretic lower bounds". However, I did not find any specific lower bounds proved and provided by the authors, or referred to in the existing literature, that could support the minimax optimality of the authors' derived regret upper bounds. Note that the authors' derived upper bounds are complex in the sense that they contain multiple terms and are related to similarity parameters, e.g., $s, \alpha, \beta$. Therefore, I doubt whether complete (not just for special degenerated cases) matching lower bounds, as claimed by the authors, exist in the literature, or can be easily derived, for their considered settings. Could authors provide explicit lower bounds to better support their claimed minimax optimality?

2. Cross-market, but with the same covariate distribution and market noise distribution.
The authors consider transfer dynamic pricing under different markets with differences and similarities in their utility functions. However, the covariate distributions $\mathcal{P}_{x}$ (line 124) and the market noise distribution $F$ (line 103) are assumed to be the same across these markets. I feel this limits the practical impact of these considered settings, as markets that are different in their utility functions, are very likely to also enjoy different covariate distributions and market noise distributions. Could the authors elaborate on potential extensions towards different covariate distributions and market noise distributions?

3. Explicit choice for $m_{0}$ in Algorithm 4, i.e., the number of initial and early episodes that leverage the offline static source log to implement the transfer learning.
In lines 153, 173 - 174, and Algorithm 4, the authors illustrate that the transfer learning is only implemented in early episodes. However, the number of these early episodes, which is denoted as $m_{0}$ in Algorithm 4, is not specified explicitly in the Theorem 9, Algorithm 4, and Theorem 10. Actually, the value of $m_{0}$ is quite essential and meaningful in theory, as it demonstrates a critical phase transition in optimally using the information in the offline historical data of other markets, and online stream data of the target market. Could the authors clarify on the choice of $m_{0}$?

4. Task similarity in Linear Models as the sparsity constraints in the cross-market parameter differences.
For the linear model, the authors use an $\ell_{0}$ sparsity constraint in the cross-market parameter differences. Such a modeling of similarities between utility functions can be justified. However, I wonder if other types of similarity measure, e.g., the $\ell_{1}$ constraints, can be well incorporated into the framework of this work. Could the authors provide more discussions on this issue?

5. How does the smoothness parameter $\eta$ in line 236 influence the regrets?

6. How the kernel $k$, as one of the input of Algorithm 3, is defined? How is $k$ related to the original kernel $K$?

7. As the feature mapping $\phi_{t} = (p_{t},x_{t})$ in Algorithm 3 consists of both $p_{t}$ and $x_{t}$, the $\mathcal{H}_{k}$ (based on kernel k) in lines 2 -- 5 of Algorithm 3 should be different from the $\mathcal{H}_{k}$ (based on kernel K) in line 225, which only concerns the covariate $x$. How is the $g$ in lines 2 -- 5 of Algorithm 3, as a function mapping from $\phi = (p,x)$, different from and related to the utility functions $g^{(k)},k\in\{0\}\cup[K]$?

8. Statement of Theorem 8.
In line 250, $g^{(ag)}$ is not defined, even across the paper including the Appendix. In line 250, $\Sigma$ is not defined, but defined in the Appendix. In line 251, $N(\lambda)$ is not defined, but defined in the Appendix. Could the authors clarify on these notations?

9. Relationships between regrets in linear parametric case and RKHS nonparametric case.
As illustrated in lines 272 - 273, the regret upper bounds for RKHS nonparametric case can reduce to $\tilde{O}(T^{1/2})$ in connection with the well-studied linear case. How is this regret bound compared to the $\tilde{O}(\log T)$ regret derived under the linear parametric case? Why are they different?

**Ethical Concerns:**

["NO or VERY MINOR ethics concerns only"]

**Final Justification:**

Thank rebuttal provides helpful explanations and clarifications. I raise the clarity score and maintain my overall rating.

**Limitations:**

Yes.

**Paper Formatting Concerns:**

I do not notice any major formatting issues in this paper.

**Quality:**

3

**Strengths And Weaknesses:**

Strengths:
1. Very practical to transfer knowledge and purchase data from markets to markets, especially useful when the target market is small.
2. Lie in the intersection of several frontier and trending AI/ML research areas of transfer learning, contextual dynamic pricing, multi-task reinforcement learning.
3. The established algorithms cover multiple scenarios, including different utility function class (linear parametric models, RKHS nonparametric ones), and different auxiliary markets' data (offline static, online concurrent).
4. The interplay between the number of auxiliary markets and similarity measure between the markets is well displayed in the derived regret bounds, providing essential insights.
5. The theoretical results align well with the numerical ones, thus it is likely that they provide operationally meaningful insights.

Weaknesses:
1. The minimax optimality, which is important and is claimed for many times across the title, abstract, main paper and conclusions, is not supported by explicit lower bounds. Moreover, such matching lower bounds may not be straightforward to derive in the considered settings. See Question 1 for more details.
2. Less practical and a bit restrictive as the covariate distributions and market noise distributions are the same across the target market and auxiliary markets. See Question 2 for more details.
3. Missing technical definitions, e.g., the kernel $k$ in the input of Algorithm 3, $g^{(ag)}$, $\Sigma$ and $N(\lambda)$ in the statement of Theorem 8. See Questions 6 and 8 for more details.

---

> ### Author Rebuttal · Authors · 2025-07-30
>
> Thank you for the positive feedback and insightful questions. We have updated our manuscript to address the following points.
>
> ---
>
> **Q1 (W1)** We appreciate this important question. While our manuscript does not provide formal lower bound proofs, our regret upper bounds are driven by estimation error rates that **match known minimax rates** in the literature—justifying our claim of minimax optimality up to log factors.
> - **Linear model**: Our regret bound $\widetilde{O}(\frac{d}{K} \log T + s_0 \log T)$ is derived from the MLE estimation error $\|\hat{\beta}^{(0)} - \beta^{(0)}\|^2 = \tilde{O}( \frac{d}{n_K} + \frac{s_0 \log d}{n_0})$, which matches minimax rates in high-dimensional GLMs with sparse shifts [4,5].
> - **RKHS model**: Our regret bound is built on $\|\hat{g}^{(0)} - g^{(0)}\|^2_{\mathcal{H}_k} = \tilde{O}(n_K^{-2\alpha\beta/(2\alpha\beta + 1)} + n_0^{-2\alpha/(2\alpha + 1)} H^{2/(2\alpha + 1)})$, matching minimax rates in RKHS regression [2,6].
> - **Minimax dependencies on $T$** are further supported by the regret lower bounds established in [7], who study contextual dynamic pricing under separable demand models, where demand takes the form $\mathbb{E}[y \mid x, p] = D(p - g(x))$ for a known, strictly decreasing demand shape $D(\cdot)$. While their model differs from ours—most notably, they do **not** adopt a random utility model—their results provide useful **single-task analogs** for our regret rates:
>   - For **linear $g(x)$**, they show a regret lower bound of $\Omega(d \log T)$, consistent with classical bounds in high-dimensional pricing.
>   - For **nonparametric $g(x)$** lying in a smooth function class (e.g., Sobolev or RKHS-type spaces), they derive a regret lower bound of $\Omega(T^{1/(2\kappa + 1)})$, where $\kappa$ captures smoothness—matching the rate in our Theorem 8 when $\kappa = \alpha\beta$.
>
> We have also sketched regret lower bounds using standard information-theoretic techniques (e.g., Fano’s inequality, packing arguments), and will add formal proofs of lower bounds to the appendix.
>
> **Sketch of Lower Bound Derivation**
> - **Linear Utility with Sparse Shift** We now sketch a lower bound argument that matches our Theorem 5 (up to logarithmic factors).
>   1. **Hard Instance Construction**: Let the source markets all share utility $\beta^{(k)} = \beta^*$, and consider two possibilities for the target:
>      - $\beta^{(0)} = \beta^*$,
>      - or $\beta^{(0)} = \beta^* + \delta e_j$, where $e_j$ is the unit vector and $j \in \mathcal{S} \subset [d]$, with $|\mathcal{S}| = s_0$.
>
>   2. **Randomization Over Sparse Supports**: We draw $j$ uniformly from a set of sparse supports $\mathcal{S} \subset [d]$, introducing uncertainty over which features differ.
>
>   3. **Fano’s Inequality**: We reduce the learner’s regret to its ability to identify which sparse vector generated the target market. Let $\delta \asymp \sqrt{\frac{\log d}{T}}$. Then the KL divergence between observations under any two hypotheses is small: $\mathrm{KL}(P_{\beta^{(0)}} \| P_{\beta^*}) \leq c.$ Fano’s inequality implies a lower bound on misclassification error, which propagates into pricing errors.
>
>   4. **Conclusion**: The learner must suffer regret at least: $ \Omega\left(\frac{d}{K} \log T + s_0 \log T\right),$ since misclassifying the utility results in suboptimal pricing. This matches our Theorem 5 up to logarithmic factors.
>
> - **RKHS Utility with Operator Smoothness**
>   1. **Function Class**: Define a class $\mathcal{G} \subset \mathcal{H}\_k$, such that for each task: $g^{(0)} = g^{(k)} + (\Sigma^{(0)})^\eta \omega, \quad \|\omega\|_{\mathcal{H}_k} \leq H.$
>
>   2. **Packing Argument**: Construct a packing set of $g^{(0)}$’s in $L^2(P_x)$ norm with separation $\delta \asymp T^{-\frac{1}{2\alpha\beta + 1}}$. Then, using standard minimax theory (see [6]), any learner must incur estimation error $\Omega(T^{-\frac{2\alpha\beta}{2\alpha\beta + 1}})$.
>
>   3. **Regret Linkage**: Via Taylor expansion of revenue (as used in Theorem 8), regret scales proportionally with squared estimation error, yielding a regret lower bound: $\Omega\left(K^{-\frac{2\alpha\beta}{2\alpha\beta + 1}} T^{\frac{1}{2\alpha\beta + 1}} + H^{\frac{2}{2\alpha + 1}} T^{\frac{1}{2\alpha + 1}}\right)$, matching our Theorem 8.
>
> ---
>
> **Q2 (W2)** We agree that markets with utility differences may also differ in covariates and noise.
> - **Covariate shifts**: As discussed in our response W1 to Reviewer cCwA, domain adaptation or importance weighting techniques can extend our framework to handle differing covariate distributions [1, 2].
> - **Noise distribution shifts**: With **known** $F^{(k)}$'s, our framework can readily accommodate heteroscedastic noise and each market has a context-dependent noise distribution, possibly with market-specific scales or shape parameters. To handle this added variability, one can incorporate techniques from robust statistics, such as weighted M-estimation, into the aggregation and debiasing steps [4]. If the $F^{(k)}$'s are **unknown**, one can estimate them nonparametrically and jointly with the utility functions, following ideas from [3]. However, this introduces new challenges for transfer—particularly in aligning or estimating heterogeneous noise distributions—which require more intricate analysis. We consider this an important direction for future work.
>
> ---
>
> **Q3** In the linear case, Phase 1 ends when $n_0 \geq \tilde{c} n_K$ (line 523), implying $m_0 = \lceil \log_2(\tilde{c} n_K) \rceil$. In the RKHS case, $m_0$ is adaptive since $\tilde{c}$ cannot be explicitly solved (line 594). The transition occurs when the target-only estimator outperforms transfer-aided estimation, determined empirically.
>
> ---
>
> **Q4** Our bias-corrected aggregation framework is modular and can indeed accommodate alternative notions of similarity, including $l_q$ constraints where $q\in [0,1]$.
> - **Computation**: For $ q \in (0,1) $, the problem remains non-convex but can be approximated via iterative reweighted $ l_1 $ methods or proximal gradient descent. The regularization parameter $ \lambda $ should scale with $ q $ to maintain consistency in sparsity control.
> - **Theory**: The trade-off between approximation (bias) and estimation (variance) terms can still be controlled. For $ q \in (0,1) $, the approximation error may increase slightly, but the estimation error improves due to better sparsity control.
>
> Thus, while our current analysis focuses on exact sparsity ($l_0$) for clarity and interpretability, the framework is general enough to support other similarity metrics, and we view this as a promising avenue for future extension.
>
> ---
>
> **Q5** The parameter $\eta$ controls the smoothness of the discrepancy between target and source utilities via the operator $(\Sigma^{(0)})^\eta$. A larger $\eta$ means the utility shift lies in a smoother subspace, making it easier to estimate. This results in smaller RKHS norm $H$, and hence tighter regret bounds in Theorem 8.
>
> ---
>
> **Q6-7** Thank you for pointing this out—you are correct. In our theoretical analysis, $\mathring{g}^{(k)}(x) \in \mathcal{H}\_k$ denotes the mean utility function, where $\mathcal{H}\_k$ is an RKHS over covariates $x$, induced by a kernel $K(x, x')$. In contrast, Algorithm 3 estimates a **log-odds function** $\ell(p, x) \in \mathcal{H}\_{\text{joint}}$ using **kernel logistic regression**, where the kernel $K\_{\text{joint}}((p,x), (p',x'))$ is defined over the joint space of price and covariates. This function is distinct from the structural utility function $\mathring{g}^{(k)}$.
>
> Our use of **KRR for log-odds estimation** is a key innovation enabling fully nonparametric modeling, in contrast to MLE in the linear case. Under mild conditions on the noise distribution $F$, the smoothness of $\mathring{g}^{(k)}$ translates to smoothness of $\ell^{(k)}$, preserving the transfer structure. We will revise the manuscript to clarify the role of $K\_{\text{joint}}$ and adopt distinct notation for $\ell$ to avoid confusion with $\mathring{g}^{(k)}$.
>
> ---
>
> **Q8 (W3)** Thank you for the suggestion. $g^{(ag)}$, $\Sigma$, and $N(\lambda)$ are used in the RKHS theoretical analysis and defined in Appendix F. We will clarify references in the main text and add a notation table in the appendix summarizing all symbols.
>
> ---
>
> **Q9** Although both bounds arise from the same CM-TDP framework, the difference reflects the **intrinsic hardness of nonparametric estimation** under binary feedback. The linear setting assumes known parametric form (GLM) and achieves $\widetilde{O}(\log T)$ regret via efficient MLE-based estimation. In contrast, the RKHS setting must learn a potentially infinite-dimensional function and suffers regret $\widetilde{O}(T^{1/2})$ even in the best case. Thus, the difference stems from **estimation complexity**, not just analysis looseness.
>
> ---
>
> **References**
>
> [1] Wang, Fan et al. Transfer Learning for Nonparametric Contextual Dynamic Pricing, arXiv:2501.18836.
>
> [2] Chai, J, Chen, E, Fan, J. Deep Transfer Q-Learning for Offline Non-Stationary Reinforcement Learning. arXiv:2501.04870.
>
> [3] Chen, E. et al. Dynamic Contextual Pricing with Doubly Nonparametric Random Utility Models, arXiv:2405.06866.
>
> [4] Xu, K. & Bastani, H. Multitask Learning and Bandits via Robust Statistics, Management Science, 2025.
>
> [5] Li, S. et al. Transfer Learning for High-Dimensional Linear Regression, JRSSB, 2022.
>
> [6] Caponnetto, A. & De Vito, E. Optimal Rates for Regularized Least-Squares Algorithm, FoCM, 2007.
>
> [7] Bu, J. et al. Context-Based Dynamic Pricing with Separable Demand Models, Management Science, 2025.

---

### Official Review · Reviewer_cCwA · 2025-07-02

**Clarity:** 3
**Significance:** 4
**Originality:** 4
**Rating:** 5
**Confidence:** 5

**Summary:**

This paper studies contextual pricing where a target market is able to leverage $K$ auxiliary markets, whose utilities differ by a structured preference shift. Specifically, the paper adopts a stochastic utility model $v_t=g(x_t) + \epsilon_t$ where $g(\cdot)$ is an unknown mean utility function and $\epsilon$ is a random noise. In this paper, the authors propose a Cross-Market Transfer Dynamic Pricing (CM-TDP) algorithm that achieves (a) $\tilde{O}(d/K \log T + s_0 \log T)$ regret for linear utility with sparse differences, and (b) $\tilde{O}(K^{-2αβ/(2αβ+1)}T^{1/(2αβ+1)} + H^{2/(2α+1)}T^{1/(2α+1)})$ regret for the RKHS utilities. They also conducted numerical simulations, where CM-TDP significantly outperforms single-market baselines, reducing cumulative regret by up to 50%.

**Questions:**

What is the computational complexity of CM-TDP in different settings (i.e. in Off-to-On and On-to-On settings, on the linear model and RKHS model)?

**Ethical Concerns:**

["NO or VERY MINOR ethics concerns only"]

**Final Justification:**

Very good work, worth a 5 even after recalibration from the first-round evaluations.

**Limitations:**

Discussed with less sufficiency. See "Weakness" above.

**Quality:**

3

**Strengths And Weaknesses:**

### Strengths:

To the best of my (as well as theirs) knowledge, this is the first paper in the pricing community that studies the utility-model shifting effect. They provide rigorous analysis and deliver minimax regret rate for both parametric and non-parametric settings (which is also the first non-parametric regret bound for transfer learning in pricing). From a high-level point of view, this work unifies bias-correction techniques from multi-task bandits with pricing, comparing to traditional bandits that only focus on prediction error. Besides, this paper is overall well-written and clearly delivered.


### Weakness:

The most significant concern raised from my perspective lies in these strong modeling assumptions. For instance, the context distribution is assumed to be identical over all markets, which is clearly over-simplifying the scenario. Also, the utility noise distribution is assumed known, and the sparsity parameter is known ahead of time. All of these assumptions listed above are restrictive (although somewhat justified in Sec 3 but not very convencing) and undermines the potential application as well as the extension of this work. I understand that this is the very first work considering this problem, but I highly encourage the authors to add more justification and discuss potential limits/extensions with respect to each assumption they have made.

There are not much beyond my main concern, as this work is overall self-consistent. Besides, the notations are slightly heavier than I expected.

---

> ### Author Rebuttal · Authors · 2025-07-30
>
> We thank the reviewer for the thoughtful summary and positive assessment of our contributions. Below, we address the main concerns regarding modeling assumptions and provide the requested discussion on computational complexity.
>
> ---
>
> **W1: Assumption on homogeneous covariate distributions.** Thank you for this insightful question! Our focus is on utility-model heterogeneity. To isolate this dimension, we assume homogeneous covariate distributions across markets. This ensures that performance differences arise solely from shifts in latent preferences, not from domain shift.
>
> That said, our framework can **accommodate covariate (domain) shift** through extensions such as **reweighting** or **importance sampling** (e.g., [1,2]). For example, under mild overlap conditions, density ratio estimators or domain-invariant representations can be incorporated into the debiasing step. *We will clarify this in the revised manuscript.*
>
> ---
>
> **W2: Assumption on noise distribution.** While we assume a known noise distribution $F(\cdot)$ for algorithmic implementation, our theoretical analysis only relies on certain regularity properties (Assumption 1), not the exact parametric form.
>
> Importantly, our framework can be extended to **unknown $F$** using **nonparametric estimation techniques** as in [4], where both the mean utility and $F$ are estimated jointly. The aggregate-then-debias architecture remains applicable, but such extensions require significantly more technical development and considering possible transfers in estimating  $F^{(k)}$ as well. We feel it is best addressed in a separate paper. *We will clarify this point in the revision.*
>
> ---
>
> **W3: Assumption on sparsity.** We appreciate the reviewer’s concern and agree it’s important to contextualize the **sparsity assumption**:
>
> * **General Framework**: Our framework permits any structured similarity between tasks; sparsity is one way to measure the function complexity of utility differences. For the **nonparametric setting**, we instead assume smoothness in an RKHS.
> * **Justification in Practice**: In many real-world applications (e.g., ride-sharing, retail), only a **small subset of covariates** drive cross-market utility variation (e.g., weather, taste, income). Sparse heterogeneity thus reflects natural structure and is widely used in bandit transfer and multitask learning [2,3].
> *  **No Need to Know $s_0$**: Importantly, our algorithm does **not** require knowledge of the sparsity level $s_0$ — it only appears in the regret bounds as a function of transfer difficulty.
> * **Beyond Sparsity**: Our analysis can be extended to **other complexity measures** (e.g., low-rank differences or structured smoothness of the utility difference), provided they enable regularized estimators with controlled generalization error. We will highlight this point in the revised version.
>
> ---
>
> **Q1** We now address the question regarding CM-TDP’s computational costs across settings.
>
> **Linear Utility Model**
>
> - **Online-to-Online Transfer**
>   - Each Episode:
>     - **Aggregation**: $O(d^2 \ell_{m-1} K \cdot N)$ for MLE using source market data from episode $m-1$, where $N$ represents the number of optimization iterations.
>     - **Debiasing**: $O(d^2 \ell_{m-1} \cdot N)$ for Lasso regression on target market data.
>   - Cumulative regret over $\log T$ episodes: $\mathcal{O}(d^2 T K\cdot N)$
> - **Offline-to-Online Transfer**
>   - Each Episode:
>     - **Phase 1 (Transfer)**: $O(d^2 n_{\mathcal{K}} \cdot N)$ for MLE on aggregated source market data, and $O(d^2 l_{m-1} \cdot N)$ per episode for bias correction.
>     - **Phase 2 (No Transfer)**: Standard linear MLE: $O(d^2 l_{m-1} \cdot N)$.
>   - Cumulative regret: $O(d^2 (n_\mathcal{K} + T) \cdot N)$
> - **Implementation Note**: The dependence on dimension $d$ varies by optimization method. Newton's method (our implementation) achieves faster convergence (smaller $N$) but maintains $d^2$ dependence. Gradient descent reduces the dependence to $O(d)$ per iteration but requires more iterations ($N$ increases correspondingly).
>
> **RKHS Utility Model**
>
> Base Complexity: KRR with $n$ samples requires $O(n^3)$ for exact matrix inversion.
>
> - **Cumulative regret**
>   - Online-to-Online Transfer: $O(T^3 K^3 \cdot N)$
>   - Offline-to-Online Transfer: $O((n_\mathcal{K}^3 + T^3) \cdot N)$
> - **Implementation Note**: We use **approximate kernel methods** (e.g., Nyström, random features) to reduce runtime to **linear scaling in $\mathcal{N}(\lambda)$**, the effective dimensionality of the feature map.
>
> *We will add a computational discussion in the appendix to clarify these scaling properties and practical trade-offs.*
>
> ---
>
> **Q2** We appreciate the feedback and **will add a notation table** summarizing key symbols and definitions in the appendix to improve accessibility, particularly for readers outside the pricing and bandit communities.
>
> ---
>
> **References**
>
> [1] Fan Wang, Feiyu Jiang, Zifeng Zhao, and Yi Yu. Transfer learning for nonparametric contextual dynamic pricing. arXiv preprint arXiv:2501.18836, 2025.
>
> [2] Jinhang Chai, Elynn Chen, Jianqing Fan. Deep Transfer $Q$-Learning for Offline Non-Stationary Reinforcement Learning. arXiv preprint arXiv:2501.04870.
>
> [3] Kan Xu and Hamsa Bastani. Multitask learning and bandits via robust statistics. Management Science, 2025.
>
> [4] Elynn Chen, Xi Chen, Lan Gao, Jiayu Li. Dynamic Contextual Pricing with Doubly Non-Parametric Random Utility Models. arXiv preprint arXiv:2405.06866

---

> > ### Comment · Reviewer_cCwA · 2025-08-04
> >
> > Thanks the authors for their detailed explanations! I am now aware of the adaptivity of their methodologies to homogeneous distributions. I will be supporting this work as most of the reviewers.

---

### Note · Authors · 2025-08-12

Dear AC,

We sincerely thank the reviewers and area chairs for their constructive feedback and are encouraged by the strong recognition of our work’s novelty, theoretical rigor, and practical significance.

Following the discussion, we have revised the manuscript to:
- Clarify assumptions and robustness, including handling covariate shift, noise distribution, different similarity characterizations beyond sparsity and smoothness, and practical guidance for assessing market similarity.
- Highlight the key technical novelties in our theoretical analysis.
- Expand on the comparison between the linear and RKHS settings.
- Add computational complexity analysis and new experiments for higher dimensions, denser differences, and runtime.
- Improve clarity with a notation table and precise definitions.

In response to concerns on minimax optimality, we have also sketched lower bound derivations for both settings, showing that our upper bounds match them up to logarithmic factors. We will include full formal proofs of these lower bounds in the final manuscript.

We believe these revisions improve the clarity, completeness, and practical applicability of our work, while preserving the core contributions:

1. The first unified transfer dynamic pricing framework accommodating both linear and nonparametric utilities under structured market differences.
2. Minimax-optimal regret bounds in both settings, with theory tightly aligned to empirical results.
3. A principled bias-corrected aggregation mechanism that is robust, adaptive, and practically relevant for cross-market transfer.

We appreciate the reviewers’ engagement and support, and we are confident that the revised manuscript offers both theoretical rigor and practical value to the NeurIPS community.

---

### Decision · Program_Chairs · 2025-09-17

**Decision:**

Accept (spotlight)

**Comment:**

A solid technical work on dynamic pricing. All reviewers are positive.  Most items of concern were thoroughly discussed and addressed in the rebuttal and post-rebuttal discussion.  I recommend accepting the paper.